# SOFT METROPOLIS-HASTINGS CORRECTION FOR MOLECULAR GENERATIVE MODEL SAMPLING

## ABSTRACT

Molecular diffusion models suffer from systematic sampling biases that prevent optimal structure formation, resulting in chemically suboptimal molecules with metastable conformations trapped in local energy minima. We introduce Metropolis-Hastings (MH) correction to molecular diffusion models, providing a principled framework to address these systematic sampling biases. The traditional hard accept-reject Metropolis-Hastings corrector creates discontinuous trajectories incompatible with the continuous nature of molecular potential energy surfaces, disrupting proper structure assembly. To address this, we develop a soft Metropolis-Hastings correction that replaces binary acceptance with continuous interpolation weighted by acceptance probabilities, maintaining smooth navigation in the chemical space while providing principled bias correction. We design three molecular-specific variants and demonstrate through extensive experiments on small molecules, drug conformations, and therapeutic antibody CDR-H3 loops that our method consistently improves chemical validity, structural stability, and conformational quality across diverse molecular families. Our method establishes MH correction as a powerful component for molecular generation.

## 1 INTRODUCTION

Molecular generation through diffusion models has emerged as a transformative approach for drug discovery and materials design (Hoogeboom et al., 2022; Xu et al., 2022; Watson et al., 2023). These models learn to reverse a diffusion process that gradually corrupts molecular structures into Gaussian noise, where a molecule with $N$ atoms is represented as $\mathbf{x}_0 \in \mathbb{R}^{3N} \times \mathcal{A}^N$ (with $\mathcal{A} = \{C, N, O, H, ...\}$ denoting atom types), and the process evolves from time $t = 0$ (data) to $t = T$ (pure noise) with $\mathbf{x}_T \sim \mathcal{N}(0, \mathbf{I})$. The success of molecular diffusion models stems from their ability to capture complex distributions $q(\mathbf{x})$ over 3D geometries, atom types, and bond configurations while respecting chemical constraints $\mathcal{C}(\mathbf{x})$ through geometric equivariance and learned force fields (Ganea et al., 2021; Jing et al., 2022).

Despite these advances, molecular diffusion models exhibit systematic sampling biases that limit their ability to generate chemically optimal structures. The denoising process often becomes trapped in metastable conformations, manifesting as distorted coordination geometries (Gillespie, 2008; Harris et al., 2023), and ring systems locked in high-energy conformations (Buttenschoen et al., 2024) (see Appendix F for detailed descriptions). The heterogeneous nature of molecular potential energy surfaces exacerbates these issues, as uniform denoising dynamics cannot adapt to local chemical environments (Powers et al., 2023).

Existing correction methods face fundamental limitations in molecular contexts. Traditional hard Metropolis-Hastings correction employs binary accept-reject decisions that create discontinuous trajectories incompatible with the continuous nature of molecular potential energy surfaces, leading to degraded generation quality in our experiments. Predictor-corrector methods (see Section 2) provide an alternative but require multiple iterations per timestep, substantially increasing computational cost while achieving limited improvements. These observations motivate our development of soft Metropolis-Hastings correction that maintains trajectory continuity while providing principled bias correction.

We present the first application of Soft Metropolis-Hastings correction to molecular diffusion models, offering a principled approach to mitigate these sampling. The MH framework ensures the

detailed balance Markov chain property through the relation $\pi(\mathbf{x})P(\mathbf{x} \to \mathbf{x}') = \pi(\mathbf{x}')P(\mathbf{x}' \to \mathbf{x})$, providing asymptotic convergence to the target distribution $\pi(\mathbf{x}) \propto \exp(-E(\mathbf{x})/k_B T)$ (Metropolis et al., 1953; Hastings, 1970), where $E(\mathbf{x})$ is the potential energy, $k_B$ is Boltzmann's constant, and $T$ is the temperature. While MH correction has shown promise in simplified diffusion settings (Sjöberg et al., 2023), molecular systems present unique challenges due to their complex conformational landscapes and stringent validity requirements. The continuous nature of molecular coordinates $\mathbf{r} \in \mathbb{R}^{3N}$ coupled with discrete chemical constraints creates a challenging optimization landscape where small perturbations can have catastrophic effects on molecular stability.

Traditional hard MH correction can be problematic for molecular structure formation. Binary accept–reject decisions create hold steps where the structure stagnates, which inflates autocorrelation and degrades mixing (Mira et al., 2001). This slows exploration of the state space and increases the risk of getting trapped in suboptimal modes (Neal et al., 2011).

Current approaches to improve molecular generation quality operate through fundamentally different mechanisms and cannot address these sampling biases. Guidance methods modify score functions as $\tilde{\mathbf{s}}_\theta = \mathbf{s}_\theta + w\nabla f_\phi(\mathbf{x})$ but require task-specific predictors $f_\phi$ and careful hyperparameter tuning (Hoogeboom et al., 2022; Vignac et al., 2023). Advanced numerical integrators reduce discretization error from $\mathcal{O}(\Delta t^2)$ to higher orders but multiply computational costs proportionally (Lu et al., 2022; Zhang & Chen, 2022). Post-hoc refinement through energy minimization or rule-based filtering cannot correct fundamental sampling errors embedded in the generation trajectory (Harris et al., 2023; Guan et al., 2024). Learning-based corrections require additional training data and sacrifice the plug-and-play nature of pre-trained models (Xu et al., 2023; Wu et al., 2024).

**Our contribution.** We introduce soft Metropolis-Hastings correction that reformulates the accept-reject mechanism as continuous interpolation $\mathbf{x}_{k-1} = \alpha\mathbf{x}_{k-1}^{\text{prop}} + (1-\alpha)\mathbf{x}_k$ with acceptance weight $\alpha \in [0, 1]$. This maintains smooth denoising trajectories essential for navigating molecular potential energy surfaces while providing principled bias correction through the MH acceptance ratio $r(\mathbf{x}_k, \mathbf{x}_{k-1}^{\text{prop}}) = \frac{1}{2}\langle \mathbf{s}_\theta^k + \mathbf{s}_\theta^{k-1}, \Delta\mathbf{x}\rangle$. We develop three complementary variants addressing different aspects of molecular generation: (i) global correction with scalar $\alpha$ preserving E(3)/SE(3) equivariance since uniform scaling commutes with group operations, (ii) local adaptive correction with per-coordinate weights $\boldsymbol{\alpha} = (\alpha_1, ..., \alpha_{3N})$ accommodating heterogeneous chemical environments where different atoms experience different force magnitudes, and (iii) distribution matching in whitened space $\tilde{\mathbf{x}} = \boldsymbol{\Sigma}^{-1/2}\mathbf{x}$ that decorrelates structural modes for more effective correction. Extensive experiments across QM9 small molecules, GEOM-Drugs conformations, and antibody design demonstrate consistent improvements in validity, uniqueness, as well as structural quality metrics, establishing soft MH correction as an effective enhancement for molecular generation.

## 2 RELIABILITY CHALLENGES IN MOLECULAR DIFFUSION SAMPLING

In this section, we examine reliability issues in molecular diffusion sampling by identifying accumulated structural errors, analyzing chemical validity constraints, and reviewing limitations of existing correction methods.

**Problem Setup.** Molecular diffusion models generate 3D structures through a reverse Markov chain $(X_k)_{k=0}^n$ that transforms Gaussian noise into chemically valid molecules (Hoogeboom et al., 2022). The chain follows transitions $p_\theta(x_{k-1}|x_k)$ parameterized by Geometry-equivariant score networks $(s_\theta^k)_{k=1}^n$, where $s_\theta^k(x_k) \approx \nabla \log q_k(x_k)$ and $q_k$ represents the marginal distribution of molecular conformations at timestep $k$ (Satorras et al., 2021). For molecular systems combining discrete atom types $a \in \mathcal{A}$ with continuous coordinates $r \in \mathbb{R}^{3N}$, and using the $x_k = (r_k, a_k)$ notation, the reverse transition decomposes as:

$$p_\theta(x_{k-1}|x_k) = p_\theta(a_{k-1}|r_k, a_k) \cdot p_\theta(r_{k-1}|r_k, a_k), \tag{2.1}$$

where the coordinate transition follows $p_\theta(r_{k-1}|r_k, a_k) = \mathcal{N}(r_{k-1}; \mu_\theta^k(r_k, a_k), \sigma_k^2\mathbf{I}_{3N})$ with mean:

$$\mu_\theta^k(r_k, a_k) = \frac{1}{\sqrt{\psi_k}}\left(r_k + (1 - \psi_k)s_\theta^k(r_k, a_k)\right). \tag{2.2}$$

Here, $\psi_k$ controls the signal-to-noise ratio at timestep $k$, and the score $s_\theta^k : \mathbb{R}^{3N} \times \mathcal{A}^N \to \mathbb{R}^{3N}$ must satisfy Geometry-equivariance: $s_\theta^k(Rr + t, a) = Rs_\theta^k(r, a)$ for all rotations $R \in \text{SO}(3)$ and translations $t \in \mathbb{R}^3$ (Köhler et al., 2020).

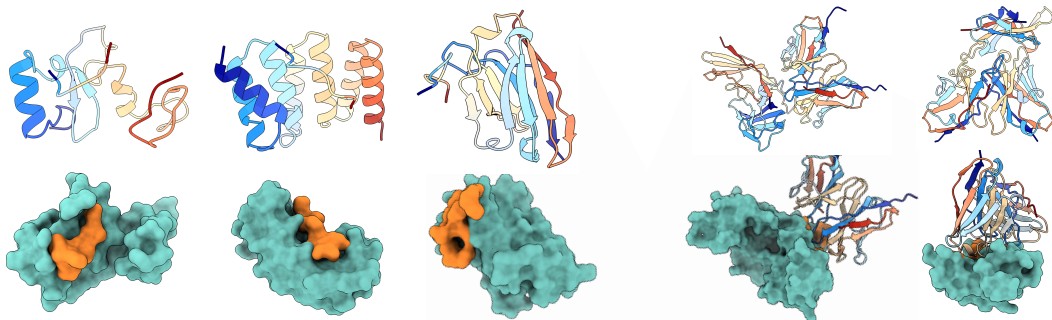

Figure 1: Generated protein examples from (E(3)/SE(3)) models: peptide samples (left) and designed antibody (right). See appendix N for more examples.

The fundamental challenge in molecular diffusion arises from the strict coupling between atoms through chemical bonds. Molecular structures exhibit cascading perturbations through chemical bonds: $\Delta r_i \rightarrow \Delta d_{ij} \rightarrow \Delta \theta_{ijk}$, where atomic displacement propagates to bond lengths $d_{ij}$ and angles $\theta_{ijk}$. Furthermore, the learned score $s_\theta^k(r, a)$ often violates chemical feasibility, particularly near ring closures where geometric constraints become overdetermined (Riniker, 2018). This motivates the need for correction mechanisms that respect both chemical validity and geometric equivariance.

**Necessity of Sampling Correction.** Molecular generation exhibits critical phase transitions during denoising where structural decisions become increasingly irreversible. At bifurcation points, the molecular state distribution decomposes as $p(x_k) \approx \sum_{i=1}^{M} w_i p_i(x_k)$ where $x_k \in \mathbb{R}^{3N}$ denotes atomic coordinates, $p_i(x_k)$ represent distinct conformational basins, and comparable weights $w_i \approx w_j$ force the system to choose between competing structural motifs (Wales, 2003). Once committed to basin $i$ at step $k^*$, escape requires overcoming the barrier over basin $j$: $\Delta E_{ij}^{\ddagger} = E_{\text{barrier}_{ij}} - E_i$ that exceeds available thermal noise $\|\sigma_{k^*} \cdot \xi\|$ with $\xi \sim \mathcal{N}(0, I)$, where $\sigma_{k^*}$ denotes the noise level (Bolhuis et al., 2002). The escape probability $P(\text{escape}) \propto \exp(-\Delta E_{ij}^{\ddagger}/\sigma_k^2)$ decays exponentially as noise level $\sigma_k$ decreases, making misfolded structures effectively permanent (Noé et al., 2019). This irreversibility is particularly severe for ring closures and stereogenic centers where topological errors formed at early timesteps cannot be corrected through subsequent denoising iterations (see Appendix G for detailed analysis). These phase transition phenomena necessitate proactive correction mechanisms that guide sampling away from suboptimal basins before commitment occurs.

**Existing Methods.** Recent MH corrections for diffusion models exhibit fundamental limitations in molecular contexts. Xie et al. (2021) employ a hard accept–reject step that fragment molecular trajectories:

$$x_{k-1} = \mathbf{1}[\alpha \geq u]x_{k-1}^{\text{prop}} + \mathbf{1}[\alpha < u]x_k, \quad \alpha = \min\left(1, \frac{p(x_{k-1}^{\text{prop}}|y)q(x_k|x_{k-1}^{\text{prop}})}{p(x_k|y)q(x_{k-1}^{\text{prop}}|x_k)}\right), \quad (2.3)$$

making performance highly sensitive to the proposal and prone to local trapping without additional mixing aids. Score-based MCMC (Sjöberg et al., 2023) requires $\mathcal{O}(L)$ network evaluations for line integrals $\log \alpha = \int_0^1 \langle s_\theta(x_t(\lambda)), \dot{x}_t(\lambda) \rangle d\lambda$, becoming prohibitive for large molecules. AIS approaches (Doucet et al., 2022) maintain particle weights $w_i^{(n)} \propto w_i^{(n-1)} \gamma_n(x_i^{(n)})/\gamma_{n-1}(x_i^{(n-1)})$. However they are computationally ill-suited for proteins due to ESS collapse in high-dim and the need for many bridges. Predictor-corrector methods (Song et al., 2020) apply iterative corrections at the same noise level:

$$x_{k-1}^{(i+1)} = x_{k-1}^{(i)} + \eta \nabla_x \log p_{k-1}(x_{k-1}^{(i)}), \quad i = 1, \ldots, M. \quad (2.4)$$

Advanced numerical integrators such as DPM-Solver++ (Lu et al., 2022) reduce discretization error through higher-order approximations but have seen limited adoption in the molecular generation community for discrete molecular structure tasks. Notably, these methods address discretization error in ODE solving, while our method corrects sampling bias in the learned score function—these are orthogonal error sources that can in principle be combined for complementary benefits.

This motivates our method, which employs continuous soft acceptance to maintain trajectory continuity while providing specialized variants tailored to different molecular generation challenges, consistent with recent advances in antibody design that emphasize adaptive, physics-aware diffusion strategies for balancing hotspot coverage and interface quality (Zhu et al., 2024; Feng et al., 2025).

## 3 CONTINUOUS METROPOLIS-HASTINGS CORRECTION ALGORITHM

In this section we develop three complementary soft acceptance variants that address the reliability challenges identified in Section 2, each tailored to different molecular generation tasks from small drug-like molecules to large biomolecular complexes.

**Unified Soft Correction Framework.** Let $\mathbf{x}_k \in \mathbb{R}^d$ denote the current state at timestep $k$ with noise level $\sigma_k$, and $\mathbf{x}_{k-1}^{\text{prop}} \sim p_\theta(\cdot|\mathbf{x}_k)$ the proposed next state from the base diffusion sampler. Define the displacement $\Delta\mathbf{x} = \mathbf{x}_{k-1}^{\text{prop}} - \mathbf{x}_k$, score functions $\mathbf{s}_\theta(\mathbf{x}, t)$ approximating $\nabla \log p_t(\mathbf{x})$, and temperature parameter $\tau > 0$ controlling acceptance sharpness. The acceptance weight $\alpha \in [0, 1]$ determines the interpolation between proposed and current states, with the log-acceptance ratio $r$ measuring score alignment. All variants employ continuous interpolation to replace binary MH decisions by:

$$\mathbf{x}_{k-1} = \alpha\mathbf{x}_{k-1}^{\text{prop}} + (1 - \alpha)\mathbf{x}_k \tag{3.1}$$

$$r = \frac{1}{2}\langle \mathbf{s}_\theta(\mathbf{x}_k, k) + \mathbf{s}_\theta(\mathbf{x}_{k-1}^{\text{prop}}, k - 1), \Delta\mathbf{x}\rangle, \quad \alpha = \min(1, \exp(r/\tau)) \tag{3.2}$$

The computation of $\alpha$ differs fundamentally across variants: Linear and Local Adaptive use the score alignment from Equation (3.2) directly; The Linear method applies a global scalar $\alpha$, while the Local method computes per-dimension weights $\boldsymbol{\alpha} = (\alpha_1, ..., \alpha_d)$. The Distribution Matching method operates in whitened space with statistically-driven weights. (See Algorithm 2 and 3 for complete pseudocode). Details of each variant are given in the Three Complementary Variants subsection below.

The rejection mechanism serves to suppress low-quality proposals through the continuous weight from Equation (3.2): when $r \ll -\tau$, $\alpha \to 0$ effectively rejects by retaining the current state; when $r \gg \tau$, $\alpha \to 1$ allows high-quality proposals to pass through. This continuous spectrum eliminates the discontinuities characterized in (G.1) while maintaining bias correction proportional to proposal reliability measured by the score alignment $\langle s_\theta, \Delta x\rangle$ from (3.2).

**Approximate Detailed Balance in Soft Acceptance.** The soft acceptance mechanism maintains approximate detailed balance through kernel symmetry in the small timestep regime. For DDPM-type diffusion models, the forward and backward transition kernels $p_{k-1|k}$ and $p_{k|k-1}$ are nearly symmetric when $\Delta t \to 0$ (Song et al., 2020; Anderson, 1982). Specifically, the log-ratio of kernels can be expanded as $\log[p_{k-1|k}(x'|x_k)/p_{k|k-1}(x_k|x')] = \langle \nabla \log p_{k-1}(x') - \nabla \log p_k(x_k), x' - x_k\rangle + O(\Delta t^2)$. Using the midpoint approximation $\nabla \log p_{k-1}(x') \approx s_\theta(x', k - 1)$ and $\nabla \log p_k(x_k) \approx s_\theta(x_k, k)$, we obtain the symmetrized form that is approximately reversible (Heng et al., 2021). The $O(\Delta t^2)$ error vanishes in the continuous-time limit, guaranteeing that our soft correction preserves the convergence to the target distribution (De Bortoli et al., 2021; Vargas et al., 2023). A rigorous proof is provided in Appendix A.3.

**Proposition.** For DDPM-type diffusion models with small timestep $\Delta t$, the proposal kernels satisfy:

$$\log \frac{p_{k-1|k}^{\text{prop}}(x'|x_k)}{p_{k|k-1}^{\text{prop}}(x_k|x')} = \frac{1}{2}\langle s_\theta(x_k, k) + s_\theta(x', k - 1), x' - x_k\rangle + O(\Delta t^2) \tag{3.3}$$

This kernel symmetry property ensures that the log-acceptance ratio in Equation (3.2) maintains detailed balance up to second-order discretization error.

**Three Complementary Variants.** We instantiate soft acceptance through three variants addressing different error characteristics in molecular systems, summarized in Table 1.

| Property | Linear Soft | Local Adaptive | Distribution Match |
|---|---|---|---|
| Acceptance | $\alpha \in [0,1]$ | $\boldsymbol{\alpha} \in [0,1]^d$ | $w \in [0,1]$ |
| Criterion | $r = \frac{1}{2}\langle \mathbf{s}_k + \mathbf{s}_{k-1}, \Delta\mathbf{x}\rangle$ | $r_i = \frac{1}{2}\langle s_{k,i} + s_{k-1,i}, \Delta x_i\rangle$ | $d = \|\boldsymbol{\epsilon}_w - \mathcal{N}(0,\mathbf{I})\|^2$ |
| Weight | $\alpha = \min(1, e^{r/\tau})$ | $\alpha_i = \min(1, e^{r_i/\tau})$ | $w = \sigma(\lambda(m_0 - d))$ |
| Update | $\mathbf{x}_{k-1} = \alpha\mathbf{x}_{k-1}^{\text{prop}} + (1-\alpha)\mathbf{x}_k$ | $x_{k-1,i} = \alpha_i x_{k-1,i}^{\text{prop}} + (1-\alpha_i)x_{k,i}$ | $\boldsymbol{\epsilon}_{k-1} = w\boldsymbol{\epsilon}_w + (1-w)\boldsymbol{\epsilon}_{k+1}$ |
| Space | Data space $\mathbb{R}^d$ | Data space $\mathbb{R}^d$ | Whitened $\boldsymbol{\Sigma}^{-1/2}\mathbb{R}^d$ |
| Conservation | $\|\nabla \times \mathbf{v}\| = \mathcal{O}(\alpha)$ | $\|\nabla \times \mathbf{v}\|_i = \mathcal{O}(\alpha_i)$ | $\|\nabla \times \mathbf{v}\| = \mathcal{O}(w)$ |
| KL bound | $D_{\text{KL}} \leq -\log(1-\alpha)$ | $D_{\text{KL}} \leq \sum_i -\log(1-\alpha_i)$ | $D_{\text{KL}} \leq H(w)$ |
| Convergence | $\rho = 1 - \mathcal{O}(\bar{\alpha})$ | $\rho = 1 - \mathcal{O}(\min_i \alpha_i)$ | $\rho = 1 - \mathcal{O}(\bar{w})$ |

Table 1: Mathematical characterization of three soft acceptance methods. Each variant implements the continuous interpolation framework from Equation (3.1) with distinct weight computations: Linear and Local use score alignment from Equation (3.2), while Distribution Matching employs statistical distance in whitened space as described in subsection 3.

Linear soft acceptance provides the simplest and most theoretically grounded approach. The global scalar $\alpha$ from Equation (3.2) preserves exact E(3)-equivariance because the interpolation operation commutes with the group action: for any $g = (R, t) \in$ E(3) with rotation $R$ and translation $t$, we have $g(\alpha\mathbf{x}^{\text{prop}} + (1 - \alpha)\mathbf{x}) = \alpha g(\mathbf{x}^{\text{prop}}) + (1 - \alpha)g(\mathbf{x})$. This property is crucial for symmetric small molecules where geometric consistency determines chemical validity (Satorras et al., 2021; Hoogeboom et al., 2022). However, applying uniform correction across all atoms ignores the reality that covalent bonds and van der Waals interactions differ by orders of magnitude in stiffness (Schlick, 2010). This limitation becomes critical in larger systems where structural heterogeneity dominates.

Local adaptive correction addresses this heterogeneity through per-dimension weights $\boldsymbol{\alpha} = (\alpha_1, ..., \alpha_N)$ computed from coordinate-specific log-ratios (Table 1). Molecular dynamics studies demonstrate that atomic fluctuations vary dramatically between protein backbone and surface loops (Karplus & McCammon, 2002; Shaw et al., 2010). The score prediction error inherits this heterogeneity, with variance spanning multiple orders of magnitude across structural elements (Jing et al., 2022). Local weights naturally adapt to these variations, applying stronger correction where uncertainty is highest while preserving well-defined regions. This approach excels for proteins and nucleic acids but sacrifices exact equivariance and increases computational complexity.

Distribution matching operates on fundamentally different principles, transforming to whitened coordinates where structural correlations are removed. In native molecular coordinates, the covariance matrix exhibits strong off-diagonal terms reflecting bond connectivity (Amadei et al., 1993). This coupling means errors in one coordinate propagate throughout the structure, making independent corrections suboptimal. The whitening transformation decorrelates these modes, allowing the matching function to operate on statistically independent components. This approach is particularly effective for conformational ensemble generation where capturing the correct distribution of states matters more than individual trajectory accuracy (Noé et al., 2019). The computational overhead of covariance estimation limits this method to systems where statistical fidelity justifies the cost.

Rather than viewing these as competing approaches, they form a toolkit where method selection depends on molecular complexity, computational resources, and the relative importance of geo-

metric versus statistical accuracy (Detailed selection guidelines and practical recommendations are provided in Appendix A.3 and H).

**Temperature Selection.** The effectiveness of soft acceptance ultimately depends on the temperature parameter $\tau$ that controls the sharpness of acceptance probabilities. The choice of $\tau$ is subject to a fundamental trade-off: decreasing $\tau$ makes the acceptance function more selective, approaching hard MH behavior ($\tau \to 0$ yields $\alpha \in \{0,1\}$), which can fragment trajectories and cause temporal inconsistencies discussed in Appendix G. Through extensive empirical studies, we find that $\tau \approx 0.8$ consistently produces smooth acceptance curves while maintaining effective bias correction (Figure 2 (left)). This relatively large temperature allows sufficient exploration to escape local minima while still suppressing poor proposals, balancing selectivity against trajectory continuity across diverse molecular systems.

**Example 3.1.** Consider a 25-component Gaussian mixture model in $\mathbb{R}^d$ with components $q = \sum_{i=1}^{25} w_i \mathcal{N}(\boldsymbol{\mu}_i, \boldsymbol{\Sigma}_i)$, where $\boldsymbol{\mu}_i \in \mathbb{R}^d$ and $\boldsymbol{\Sigma}_i \in \mathbb{R}^{d \times d}$ have anisotropic covariances with condition numbers ranging from 10 to 100. The weights $w_i$ follow exponential decay from the grid center as defined in Appendix C. In this setting, we evaluate the soft acceptance mechanism from Equation (3.1) across dimensions $d \in \{30, 300\}$ to assess scalability.

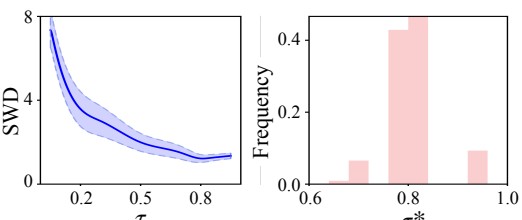

Figure 2: Left: Sliced Wasserstein distance versus temperature $\tau$. Right: Distribution of optimal $\tau^*$ across problem instances.

We parameterize the temperature as $\tau \in [0, 1.0]$ and compute the sliced Wasserstein distance $\mathrm{SWD}(p_{\text{true}}, \hat{p}_\tau)$ between the true distribution and samples from the soft-corrected diffusion process $\hat{p}_\tau$ using the mechanism in subsection 3. The hierarchical Gaussian mixture construction (Appendix C) creates 500 independent problem instances with varying covariance structures.

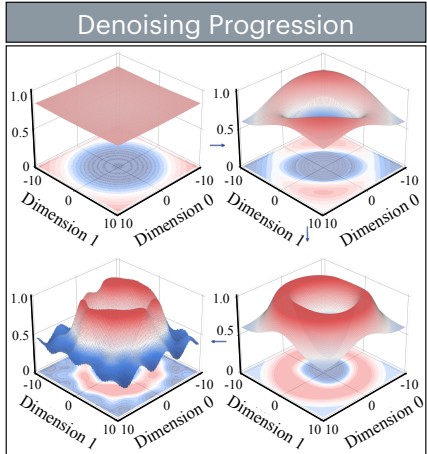

Figure 3: Evolution of acceptance rates across denoising timesteps for two example dimensions, showing how soft acceptance maintains smooth transitions throughout the trajectory.

Figure 2 (right) shows that optimal temperatures $\tau^* := \arg\min_\tau \mathrm{SWD}(p_{\text{true}}, \hat{p}_\tau)$ cluster tightly, with 80% of instances achieving minimum SWD within $[0.75, 0.85]$. This concentration remains stable despite 500-fold variation in problem configurations. The trimodal distribution reveals problem-specific optima: strongly separated components favor lower temperatures while overlapping components benefit from higher values. Figure 3 illustrates how acceptance rates evolve from uniform to structured patterns across denoising timesteps, maintaining smooth transitions in example dimensions.

**Geometric Interpretation of Log-Acceptance Ratio.** The log-acceptance ratio in Equation (3.2) admits a profound geometric interpretation that reveals why the inner product structure naturally emerges from the Metropolis-Hastings framework in continuous spaces (Metropolis et al., 1953; Hastings, 1970). Consider the score functions $\mathbf{s}_\theta(\mathbf{x}, t) = \nabla_\mathbf{x} \log p_t(\mathbf{x})$ as vector fields on the data manifold pointing toward regions of increasing probability density (Song et al., 2020). The log-acceptance ratio $r = \frac{1}{2} \langle \mathbf{s}_k + \mathbf{s}_{k-1}, \Delta\mathbf{x} \rangle$ measures the alignment between the proposed displacement $\Delta\mathbf{x} = \mathbf{x}_{k-1}^{\text{prop}} - \mathbf{x}_k$ and the average gradient field. Starting from the Metropolis-Hastings acceptance ratio for diffusion kernels (De Bortoli et al., 2021), we have $\log[p(\mathbf{x}_{k-1}^{\text{prop}})q(\mathbf{x}_k|\mathbf{x}_{k-1}^{\text{prop}})/p(\mathbf{x}_k)q(\mathbf{x}_{k-1}^{\text{prop}}|\mathbf{x}_k)]$. Applying first-order Taylor expansion around the midpoint $\mathbf{x}_{\text{mid}} = \frac{1}{2}(\mathbf{x}_k + \mathbf{x}_{k-1}^{\text{prop}})$ yields:

$$\log p(\mathbf{x}_{k-1}^{\text{prop}}) - \log p(\mathbf{x}_k) \approx \langle \nabla \log p(\mathbf{x}_{\text{mid}}), \Delta\mathbf{x} \rangle \approx \frac{1}{2} \langle \mathbf{s}_k + \mathbf{s}_{k-1}, \Delta\mathbf{x} \rangle \tag{3.4}$$

---

**Algorithm 1** Distribution Matching Soft-MH Correction

---

1: **Input:** noise schedule $(\ell_k)_{k=0}^n$; proposal $q_{\ell_k|\ell_{k+1}}$; covariance $\Sigma$; threshold $m_0$; scale $\lambda$
2: **Init:** sample $X_n \sim \mathcal{N}(0, I)$
3: **for** $k = n - 1$ **to** $0$ **do**
4:     $\tilde{X}_k \sim q_{\ell_k|\ell_{k+1}}(\cdot \mid X_{k+1})$
5:     $\epsilon_w \leftarrow \Sigma^{-1/2}(\tilde{X}_k - \mu), \quad \epsilon_{k+1} \leftarrow \Sigma^{-1/2}(X_{k+1} - \mu)$
6:     $d \leftarrow \|\epsilon_w\|^2 - \dim(X) \cdot \ell_k$          $\triangleright$ Compare with noise level $\ell_k$
7:     $w \leftarrow \sigma(\lambda(m_0 - d))$
8:     $X_k \leftarrow \Sigma^{1/2}(w\,\epsilon_w + (1 - w)\,\epsilon_{k+1}) + \mu$
9: **end for**
10: **Output:** $X_0$ (complete algorithm provided in Appendix 3)

---

where the second approximation exploits the smooth variation of score functions between adjacent timesteps (see Section 3). The symmetrization through averaging ensures the reversibility condition required for detailed balance (Heng et al., 2021) (Proposition 3.3). The inner product decomposes as $r = \|\bar{\mathbf{s}}\| \cdot \|\Delta \mathbf{x}\| \cdot \cos\theta$ where $\theta$ is the angle between the average score and the proposed displacement. When $\theta \approx 0$, the displacement aligns with gradient ascent yielding $r > 0$ and $\alpha \approx 1$ for strong acceptance. Orthogonal moves with $\theta \approx \pi/2$ give $r \approx 0$ and neutral acceptance $\alpha \approx 1/2$. Moves against the gradient with $\theta \approx \pi$ produce $r < 0$ and $\alpha \approx 0$ for strong rejection. This geometric structure naturally encourages probability-increasing moves while allowing exploratory steps in orthogonal directions.(see Appendix A.3)

**Noise-Space Perspective and Statistical Correction for Distribution Matching.** While trajectory continuity addresses discontinuities in data space, molecular systems exhibit strong correlations that complicate direct coordinate-based corrections (Amadei et al., 1993; Karplus & McCammon, 2002). In aromatic rings and hydrogen-bonded networks, atomic displacements are highly coupled through the covariance structure $\mathcal{L}_i^k = \mathrm{Cov}[\mathbf{r}_i, \mathbf{r}_j]$ for atoms $j \in \mathcal{N}(i)$, making independent corrections potentially destructive (Wales, 2003; Schlick, 2010). The diffusion framework fundamentally learns $p(x_{t-1}|x_t) = \int p(x_{t-1}|\epsilon)p(\epsilon|x_t)d\epsilon$ where $\epsilon$ represents the noise to be removed (Ho et al., 2020). Direct operation in noise space aligns with this generative mechanism: the whitening transformation $\epsilon_{\mathrm{whitened}} = \Sigma^{-1/2}\epsilon$ decouples structural correlations encoded in the covariance $\Sigma$, projecting the noise into a space where $\mathbb{E}[\epsilon_{\mathrm{whitened}}\epsilon_{\mathrm{whitened}}^T] = I$ (Kessy et al., 2018)(see Appendix J for detailed derivation). In this standardized space, the quality metric $\|\epsilon_{\mathrm{whitened}} - \mathcal{N}(0, I)\|^2$ directly measures statistical correctness independent of molecular geometry, enabling corrections that preserve the correlation structure essential for chemical validity while ensuring the noise removal process follows theoretical guarantees (Vincent, 2011; Song et al., 2020).

We now summarize the Distribution Matching soft Metropolis-Hastings correction algorithm, whose pseudocode is given in Algorithm 1. Given a noise schedule $(\ell_k)_{k=0}^n$ with $\ell_n = 1$ and $\ell_0 = 0$, the algorithm proceeds by simulating a denoising chain $(X_k)_{k=0}^n$ starting from $X_n \sim \mathcal{N}(0, I)$. Recursively, given the state $X_{k+1}$ at timestep $k + 1$, the state $X_k$ is obtained by:

1. Sample proposal $\tilde{X}_k \sim q_{\ell_k|\ell_{k+1}}(\cdot|X_{k+1})$ from the base diffusion model.

2. Transform to whitened space: $\epsilon_w = \Sigma^{-1/2}(\tilde{X}_k - \mu)$, $\epsilon_{k+1} = \Sigma^{-1/2}(X_{k+1} - \mu)$ using covariance $\Sigma$.

3. Compute matching weight: $w = \sigma(\lambda(m_0 - d))$ where $d = \|\epsilon_w\|^2 - \dim(X) \cdot \ell_k$.

4. Apply soft correction and inverse transform: $X_k = \Sigma^{1/2}[w\epsilon_w + (1 - w)\epsilon_{k+1}] + \mu$.

**Related Work.** Recent advances in molecular diffusion models have focused on improving sampling quality through various orthogonal approaches. We situate our soft Metropolis-Hastings correction within this broader landscape.

Sampling corrections for diffusion models fall into several categories. Metropolis-Hastings methods apply accept-reject mechanisms to correct sampling biases, though typically using hard decisions. Predictor-corrector schemes (Song et al., 2020) alternate between denoising and Langevin dynamics steps. Advanced numerical integrators (Lu et al., 2022; Zhang & Chen, 2022) reduce discretization error through higher-order approximations. Our soft acceptance mechanism provides continuous

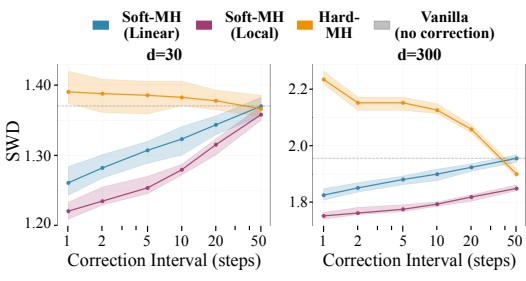

Figure 4: 95% confidence intervals for the SWD versus correction interval for Gaussian mixture model.

Table 2: Performance comparison on Gaussian mixture.

| Method | d=30 | d=300 |
|---|---|---|
| Vanilla | 1.3702 | 1.9554 |
| MH-Linear | 1.2615 | 1.8256 |
| MH-Local | 1.2215 | 1.7523 |
| MH-Hard | 1.3901 | 2.2331 |

interpolation between acceptance and rejection, addressing discontinuity issues while maintaining correction benefits.

For molecular generation specifically, several approaches ensure chemical validity. Equivariant architectures (Hoogeboom et al., 2022; Satorras et al., 2021) build geometric consistency into the model design. Torsional diffusion (Jing et al., 2022) operates in internal coordinate space. Guidance methods (Vignac et al., 2023) rely on property predictors to shape the generation process. Our method operates at the sampling level and thus remains orthogonal to these architectural and guidance innovations, offering complementary improvements without requiring model retraining or architectural changes.

The modular nature of diffusion models has enabled sophisticated hybrid approaches. For instance, Wu et al. (2024) combine SMC with learned proposals for protein-ligand docking, while Guan et al. (2024) layer decomposed priors with guided sampling. Soft MH correction extends this composability paradigm by providing a post-hoc sampling refinement that requires no architectural modifications or retraining, demonstrating that significant improvements can be achieved through principled modifications to the sampling dynamics alone.

## 4 EXPERIMENTS

We evaluate soft Metropolis-Hastings correction across diverse molecular generation tasks, demonstrating its effectiveness as a universal enhancement component. We begin with controlled experiments on Gaussian mixture models to analyze the correction mechanism, then assess performance on small molecule generation with QM9 (Ramakrishnan et al., 2014) and GEOM (Axelrod & Gomez-Bombarelli, 2022), peptide structure prediction with PepBench (Bryant & Noé, 2024), and therapeutic antibody design with RFAntibody (Bennett et al., 2024).

**Gaussian mixture analysis.** We evaluate soft acceptance on the 25-component hierarchical Gaussian mixture from Example 3 with exact score functions (Cardoso et al., 2023). We compute sliced Wasserstein (SW) distances across 500 replications with 1000 samples each. It can be observed that soft MH with $\tau = 0.8$ outperforms all baselines. The results are presented in Table 2. The optimal temperature consistently falls within $\tau^* \in [0.75, 0.85]$, demonstrating that soft acceptance provides substantial gains without extensive tuning. Full details in Appendix C.

**Ablations.** We study the effect of correction interval on the Gaussian mixture model from Example 3. Figure 4 reveals distinct dimensional behaviors: at $d = 30$, soft MH variants show monotonic improvement with increased correction frequency, while hard MH remains flat. At $d = 300$, soft variants plateau after 10-step intervals, suggesting diminishing returns from frequent corrections, whereas hard MH degrades catastrophically with single-step corrections due to accumulated temporal misalignment. This demonstrates that soft acceptance maintains robustness across correction frequencies, with optimal performance at every step but remaining effective even with sparse applications.

**Temperature sensitivity.** We study the effect of temperature parameter $\tau$ on QM9 with EDM as the base model. As shown in Section 3, our Gaussian mixture experiments (Figure 2) identified an

Table 4: QM9

| Method | Molecule Stability | Atom Stability | Validity | Novelty | Uniqueness |
|---|---|---|---|---|---|
| GeoLDM | 89.70% | 98.99% | 93.30% | 55.65% | 99.57% |
| w/ **Linear** | 90.70% | **99.11%** | **95.70%** | **57.26%** | **100.00%** |
| w/ **Local** | 89.80% | 98.92% | 94.30% | 56.37% | 99.89% |
| w/ **Dist. Match** | **91.30%** | 99.09% | 95.10% | 57.01% | 99.79% |
| GeoLDM + GUIDE | 90.60% | 99.02% | 95.20% | 53.28% | **100.00%** |
| w/ **Linear** | **92.10%** | **99.18%** | **96.10%** | 53.70% | 99.94% |
| w/ **Local** | 90.20% | 99.00% | 95.00% | 53.10% | 99.90% |
| w/ **Dist. Match** | 91.10% | 99.15% | 95.90% | **54.10%** | 99.86% |
| EDM | 80.40% | 98.34% | 91.90% | 66.88% | 99.89% |
| w/ Hard MH | 80.00% | 97.34% | 90.00% | 63.98% | 99.85% |
| w/ PC- Corrector | 83.60% | 98.61% | 92.80% | 67.98% | 99.92% |
| w/ **Linear** | 82.90% | 98.76% | 93.60% | **69.67%** | 99.68% |
| w/ **Local** | **92.60%** | 99.12% | 96.00% | 57.29% | 98.54% |
| w/ **Dist. Match** | 91.70% | **99.56%** | **98.70%** | 58.49% | 84.30% |

optimal temperature range $\tau \in [0.75, 0.85]$ across 300 problem instances. Table 6 validates that this finding transfers to molecular generation tasks.

**Extended steps comparison.** To investigate whether our improvements stem from principled bias correction rather than simply additional computation, we compare our method against baseline samplers with extended sampling steps in Table 7.

**Molecular generation benchmarks.** We evaluate soft MH correction on several molecular diffusion models across increasing complexity scales. For small molecules generation, we test on QM9 and GEOM-Grug using EDM (Hoogeboom et al., 2022) and GeoLDM (Xu et al., 2022), achieving consistent validity improvements (Tables 4 and 13).

For peptide structures, we apply our method to PepGLAD (Kong et al., 2024) on the PepBench benchmark, which encompasses diverse peptide conformations ranging from linear to cyclic structures with varying secondary structure elements. Without any model retraining or fine-tuning, soft MH correction demonstrates consistent RMSD reductions across different peptide families (Table 5), highlighting the generalizability of our approach to biomolecular systems with complex conformational landscapes and hydrogen bonding networks.

Table 3: Performance of RFAntibody with soft MH correction on CDR-H3 loops

| Method | RMSD[*] | PAE | pLDDT |
|---|---|---|---|
| RFAntibody | 1.898 | 7.826 | 0.849 |
| w/ Linear | 1.803 | 7.454 | **0.899** |
| w/ Local | **1.754** | **7.421** | 0.898 |
| w/ Dist. Match | 1.783 | 7.438 | 0.898 |

[*] CDR-H3 RMSD (Å).

To showcase compatibility with existing enhancements, we combine soft MH with ChemGuide (Shen et al., 2024) guidance, achieving additive improvements. Finally, we extend to therapeutic antibody design with RFAntibody(Bennett et al., 2024), evaluating on high-resolution antibody-antigen complexes from Guest et al. (2021). We selected 45 structures with the best resolution from their benchmark and generated 200 samples per complex. Soft Metropolis-Hastings correction consistently improved RMSD, PAE, and pLDDT metrics across all variants (Table 3), demonstrating its effectiveness for therapeutic antibody generation task.

We also conducted experiments on skip-step correction for EDM, GeoLDM, and peptide generation tasks. These additional results are detailed in Appendix M.

## 5 CONCLUSION

We have introduced soft Metropolis-Hastings correction for molecular diffusion models, a novel approach that replaces binary accept-reject decisions with continuous interpolation to maintain trajectory smoothness essential for chemical validity. Our method addresses systematic sampling biases through three complementary variants—Linear, Local Adaptive, and Distribution Matching—each

Table 5: Structure quality on PepBench and PepBDB

| Method | PepBench | | | PepBDB | | |
|---|---|---|---|---|---|---|
| | $\text{RMSD}_{C_\alpha}\downarrow$ | $\text{RMSD}_{\text{atom}}\downarrow$ | DockQ $\uparrow$ | $\text{RMSD}_{C_\alpha}\downarrow$ | $\text{RMSD}_{\text{atom}}\downarrow$ | DockQ $\uparrow$ |
| PepGLAD | 4.09 | 5.30 | 0.58 | 6.96 | 7.88 | 0.38 |
| PepGLAD + Hard MH | 4.25 | 5.48 | 0.55 | – | – | – |
| PepGLAD + PC-Corrector | 4.07 | 5.29 | 0.56 | – | – | – |
| PepGLAD + **Linear** | **3.98** | 5.28 | **0.63** | **6.33** | 7.58 | 0.41 |
| PepGLAD + **Local** | 4.15 | 5.26 | 0.59 | 6.72 | 7.62 | **0.44** |
| PepGLAD + **Dist. Match** | 4.05 | **5.21** | 0.60 | 6.41 | **7.49** | 0.42 |

Table 6: Temperature ablation on QM9 (EDM as base model)

| Method | $\tau = 0.4$ | $\tau = 0.6$ | $\tau = 0.8$ | $\tau = 0.9$ |
|---|---|---|---|---|
| EDM + Linear (Validity) | 92.80% | 93.20% | **93.60%** | 93.50% |
| EDM + Local (Validity) | 92.90% | 95.30% | **96.00%** | 95.40% |
| EDM + Dist. Match (Validity) | 92.60% | 95.10% | **98.70%** | 95.50% |

Table 7: Comparison with extended sampling steps on QM9 (EDM as base model)

| Method | Molecule Stability | Atom Stability | Validity | Uniqueness |
|---|---|---|---|---|
| EDM (1000 steps) | 80.40% | 98.34% | 91.90% | 99.89% |
| EDM (1500 steps) | 83.20% | 98.68% | 93.50% | 99.95% |
| EDM (2000 steps) | 84.00% | 98.76% | 94.00% | **100.00%** |
| EDM + Soft MH (1000 steps) | **92.60%** | **99.12%** | **96.00%** | 98.54% |

tailored to different aspects of molecular generation. This strategy has proven effective across diverse molecular systems, from small drug-like molecules to therapeutic antibodies. The results demonstrate that soft MH consistently improves validity, uniqueness, and structural quality metrics while remaining compatible with guidance methods in diffusion.

**Limitations and future work.** Due to space constraints, detailed discussions are provided in Appendix K.

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

## ETHICS STATEMENT

This work focuses on improving sampling methods for molecular diffusion models with applications in drug discovery and therapeutic design. We acknowledge the dual-use nature of molecular generation technology and emphasize that our methods are intended solely for legitimate scientific research and therapeutic development. The proposed soft Metropolis-Hastings correction techniques do not introduce new ethical concerns beyond those inherent to molecular generation models. All experiments were conducted on publicly available datasets (QM9, GEOM-Drugs, PepBench, and curated antibody structures) without involving human subjects or animal testing. We did not generate or evaluate molecules with known toxic or harmful properties. The computational methods presented pose no direct risk to human health or safety. We acknowledge the importance of responsible development in AI-assisted drug discovery and encourage users of our methods to adhere to established ethical guidelines in pharmaceutical research and to consider the broader societal implications of automated molecular design.

## REPRODUCIBILITY STATEMENT

We provide complete code implementation in the supplementary materials to ensure full reproducibility of our results. The supplementary code includes: (1) implementations of all three soft Metropolis-Hastings variants (Linear, Local Adaptive, and Distribution Matching) with detailed documentation, (2) scripts to reproduce all experiments on QM9, GEOM-Drugs, PepBench, and antibody benchmarks, (3) pre-processing and evaluation pipelines with exact hyperparameters used in our experiments, and (4) trained model checkpoints for immediate use. Algorithms 1-3 in the paper and appendix provide step-by-step pseudocode for each method. Section 3 and Appendix H detail the model architectures and procedures. Appendix B contains complete specifications for the Gaussian mixture experiments including all parameter values. The temperature parameter $\tau = 0.8$ and other hyperparameters are explicitly specified throughout. All datasets used are publicly available: QM9 from `rdkit`, GEOM from the official repository, PepBench from the published benchmark, and antibody structures from SAbDab. The code is structured as a modular library that can be easily integrated into existing molecular diffusion pipelines without requiring modifications to the base models.

## A BACKGROUND ON DENOISING DIFFUSION MODELS

### A.1 DENOISING DIFFUSION PROBABILISTIC MODELS

In this section, we provide further background on DDMs based on the DDPM framework (Ho et al., 2020; Dhariwal & Nichol, 2021; Song et al., 2020).

DDPMs define generative models through a forward diffusion process that gradually adds noise to data:

$$q(x_t|x_{t-1}) = \mathcal{N}(x_t; \sqrt{\alpha_t}x_{t-1}, (1 - \alpha_t)I) \tag{A.1}$$

where $(\alpha_t)_{t=1}^T$ is a variance schedule. This can be written in closed form:

$$q(x_t|x_0) = \mathcal{N}(x_t; \sqrt{\bar{\alpha}_t}x_0, (1 - \bar{\alpha}_t)I) \tag{A.2}$$

where $\bar{\alpha}_t = \prod_{s=1}^t \alpha_s$. Equivalently, $x_t = \sqrt{\bar{\alpha}_t}x_0 + \sqrt{1 - \bar{\alpha}_t}\epsilon$ with $\epsilon \sim \mathcal{N}(0, I)$.

The reverse process learns to denoise through parametric approximations $(m_\theta^{0|t})_{t=1}^T$ defined as:

$$m_\theta^{0|t}(x_t) = \frac{x_t - \sqrt{1 - \bar{\alpha}_t}\epsilon_\theta^t(x_t)}{\sqrt{\bar{\alpha}_t}} \tag{A.3}$$

and trained by minimizing the denoising loss:

$$\mathcal{L} = \sum_{t=1}^T w_t \mathbb{E}\left[\|\epsilon_t - \epsilon_\theta^t(\sqrt{\bar{\alpha}_t}x_0 + \sqrt{1 - \bar{\alpha}_t}\epsilon_t)\|^2\right] \tag{A.4}$$

where $(\epsilon_t)_{t=1}^T$ are i.i.d. standard normal vectors, $x_0 \sim q$, and $(w_t)_{t=1}^T$ are nonnegative weights.

The reverse generative process follows:

$$p_\theta(x_{t-1}|x_t) = \mathcal{N}(x_{t-1}; \mu_\theta(x_t, t), \sigma_t^2 I) \tag{A.5}$$

where the mean is parameterized as:

$$\mu_\theta(x_t, t) = \frac{1}{\sqrt{\alpha_t}}\left(x_t - \frac{1 - \alpha_t}{\sqrt{1 - \bar{\alpha}_t}}\epsilon_\theta^t(x_t, t)\right) \tag{A.6}$$

and $\sigma_t^2$ is either fixed or learned. The connection to score-based models comes through:

$$s_\theta(x_t, t) = -\frac{\epsilon_\theta^t(x_t, t)}{\sqrt{1 - \bar{\alpha}_t}} \approx \nabla \log q_t(x_t) \tag{A.7}$$

When $T$ is large, $q_T(x_T) \approx \mathcal{N}(0, I)$, allowing generation from pure noise. The denoising objective corresponds to a variational lower bound on the log-likelihood when appropriately weighted. In practice, DDPMs are trained with large $T$ (e.g., 1000) but can use fewer denoising steps $n \ll T$ during inference through DDIM (Song et al., 2021) or other accelerated samplers.

### A.2 LATENT DIFFUSION MODELS

Latent diffusion models (LDM) (Rombach et al., 2022) define a DDM in a latent space to improve computational efficiency. Let $\mathcal{E} : \mathbb{R}^d \to \mathbb{R}^p$ be an encoder function and $\mathcal{D} : \mathbb{R}^p \to \mathbb{R}^d$ a decoder function with $p \ll d$. We assume these functions satisfy $\mathcal{D}_\sharp \mathcal{E}_\sharp q \approx q$, where $\mathcal{E}_\sharp q$ denotes the pushforward measure (the law of $\mathcal{E}(X)$ where $X \sim q$).

A LDM approximating $q$ is given by $\mathcal{D}_\sharp p_\theta^0$, where $p_\theta^0$ is a diffusion model trained on samples from $\mathcal{E}_\sharp q$. The training process in latent space follows:

$$\mathcal{L}_{\text{latent}} = \sum_{t=1}^T w_t \mathbb{E}_{z_0 \sim \mathcal{E}_\sharp q, \epsilon}\left[\|\epsilon - \epsilon_\theta^t(\sqrt{\bar{\alpha}_t}z_0 + \sqrt{1 - \bar{\alpha}_t}\epsilon, t)\|^2\right] \tag{A.8}$$

where the neural network $\epsilon_\theta^t : \mathbb{R}^p \times \mathbb{R} \to \mathbb{R}^p$ operates in the lower-dimensional latent space.

For sampling, we first generate latent codes using the reverse diffusion process:

$$p_\theta(z_{t-1}|z_t) = \mathcal{N}(z_{t-1}; \mu_\theta^z(z_t, t), \sigma_t^2 I_p) \tag{A.9}$$

where

$$\mu_\theta^z(z_t, t) = \frac{1}{\sqrt{\alpha_t}}\left(z_t - \frac{1-\alpha_t}{\sqrt{1-\bar{\alpha}_t}}\epsilon_\theta^t(z_t, t)\right) \tag{A.10}$$

Starting from $z_T \sim \mathcal{N}(0, I_p)$, we iteratively denoise to obtain $z_0$, then decode to get the final sample: $x_0 = \mathcal{D}(z_0)$.

The computational advantage is significant: each denoising step operates on $p$-dimensional vectors rather than $d$-dimensional ones, reducing the cost by a factor of approximately $(d/p)^2$ for convolutional architectures. For high-resolution images or complex molecular structures where $d$ can be very large, this enables practical generation that would otherwise be computationally prohibitive.

For inverse problems or conditional generation in latent space, the target distribution becomes:

$$\pi(z) \propto p(y|\mathcal{D}(z))\mathcal{E}_\sharp q(z) \tag{A.11}$$

where $p(y|\cdot)$ encodes the conditioning information or constraints. Sampling from $\pi$ requires careful handling of the decoder in the likelihood evaluation, which motivates various approximation strategies in the literature.

### A.3 Soft Acceptance Distribution Decomposition

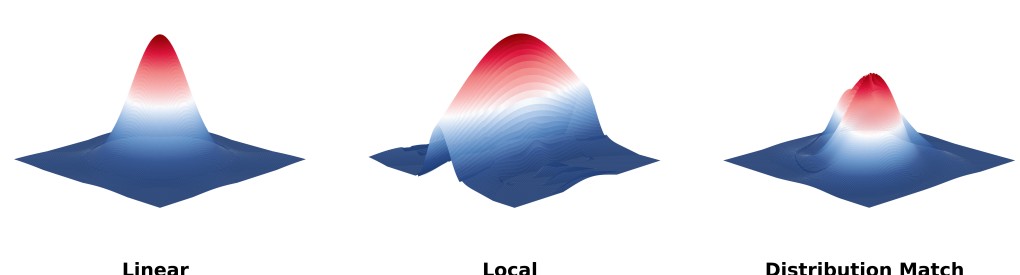

**Linear**          **Local**          **Distribution Match**

Figure 5: These three visualizations illustrate the fundamental differences in acceptance probability landscapes for three variants of soft Metropolis-Hastings correctors. **Linear Soft Acceptance (left)** exhibits a perfectly circular symmetric cone, representing a globally uniform acceptance probability $\alpha$ where correction strength depends solely on Euclidean distance from the center, regardless of which principal component direction the deviation occurs in. While simple and efficient, this approach neglects the heterogeneity across different dimensions in molecular systems. **Local Adaptive (middle)** displays a pronounced elliptical shape with an aspect ratio of approximately 4:1, reflecting dimension-specific acceptance probabilities $\alpha_i$ that enable stricter corrections for rigid regions (e.g., protein backbones) while applying more relaxed criteria for flexible regions (e.g., loop regions). **Distribution Matching (right)** reveals a complex asymmetric bimodal structure with distorted geometry, arising from the non-trivial geometric deformations produced when corrections performed in whitened space are inverse-transformed back to the original space. The ripples and secondary peaks reflect the decoupling and re-coupling process of correlated structures within the molecular system.

We establish the distributional properties of soft acceptance through recursive decomposition. Under soft MH correction, the joint distribution satisfies a modified detailed balance condition.

**Remark on Target Distribution Ratio.** In the MH acceptance ratio, we implicitly assume that the target distribution ratio $\pi_{k-1}(x_{k-1})/\pi_k(x_k) \approx 1$ in the small timestep regime. This approximation is justified because: (i) consecutive states $x_k$ and $x_{k-1}$ are close in the diffusion trajectory with $\|x_{k-1} - x_k\| = O(\sqrt{\Delta t})$, (ii) the marginal distributions $\pi_k$ and $\pi_{k-1}$ change smoothly with $k$, and

(iii) the score function $s_\theta(x, t) \approx \nabla \log p_t(x)$ implicitly encodes the local density information. This allows us to focus on the proposal kernel ratio, which is explicitly handled through the score alignment term in Equation (3.2). While this simplification may introduce a small bias, our experiments demonstrate that it does not compromise the effectiveness of the soft correction mechanism.

**Lemma A.1 (Soft Detailed Balance).** For the soft-corrected process with acceptance weight $\alpha_k \in [0, 1]$ following from Equation (3.1), it holds that:

$$\pi_k(x_k) p_{k-1|k}^{\text{soft}}(x_{k-1}|x_k) = \pi_{k-1}(x_{k-1}) p_{k|k-1}^{\text{soft}}(x_{k-1}|x_k) \tag{A.12}$$

where $\pi_k(x_k)$ denotes the target distribution at timestep $k$, and $p_{k-1|k}^{\text{soft}}(x_{k-1}|x_k)$ represents the soft transition probability from state $x_k$ at time $k$ to state $x_{k-1}$ at time $k-1$. The soft transition kernel, which encompasses the three variants presented in Table 1, is defined through the integral

$$p_{k-1|k}^{\text{soft}}(x_{k-1}|x_k) = \int_{\mathcal{X}} \delta_\alpha(x_{k-1} - [\alpha x' + (1-\alpha)x_k]) p_{k-1|k}^{\text{prop}}(x'|x_k) dx' \tag{A.13}$$

where $\mathcal{X}$ is the state space, $x'$ is the proposed state from the base diffusion sampler, $p_{k-1|k}^{\text{prop}}(x'|x_k)$ is the proposal distribution corresponding to the uncorrected diffusion transition, and $\delta_\alpha(\cdot)$ is the Dirac delta function enforcing the soft acceptance rule.

*Proof.* Starting from the soft update rule $x_{k-1} = \alpha_k x_{k-1}^{\text{prop}} + (1-\alpha_k)x_k$ where $x_{k-1}^{\text{prop}} \sim p_{k-1|k}^{\text{prop}}(\cdot|x_k)$ is the proposed next state, we derive the soft transition probability as

$$p_{k-1|k}^{\text{soft}}(x_{k-1}|x_k) = \mathbb{E}_\alpha\left[p_{k-1|k}^{\text{prop}}\left(\frac{x_{k-1} - (1-\alpha)x_k}{\alpha}\bigg|x_k\right) \cdot \frac{1}{\alpha^d}\right] \tag{A.14}$$

$$= \int_0^1 p_{k-1|k}^{\text{prop}}\left(\frac{x_{k-1} - (1-\alpha)x_k}{\alpha}\bigg|x_k\right) \cdot \frac{1}{\alpha^d} \cdot p(\alpha|r_k) d\alpha \tag{A.15}$$

Here $d$ is the dimensionality of the state space, $\alpha^{-d}$ is the Jacobian determinant arising from the change of variables $x' \to x_{k-1}$, and $p(\alpha|r_k) = \delta(\alpha - \min(1, e^{r_k/\tau}))$ is the acceptance probability distribution determined by the log-acceptance ratio from Equation (3.2): $r_k = \frac{1}{2}\langle s_\theta(x_k, k) + s_\theta(x_{k-1}^{\text{prop}}, k-1), x_{k-1}^{\text{prop}} - x_k\rangle$, where $s_\theta(x, t)$ approximates the score function $\nabla \log q_t(x)$ and $\tau > 0$ is the temperature parameter controlling acceptance sharpness.

The detailed balance is verified through

$$\frac{\pi_k(x_k) p_{k-1|k}^{\text{soft}}(x_{k-1}|x_k)}{\pi_{k-1}(x_{k-1}) p_{k|k-1}^{\text{soft}}(x_k|x_{k-1})} = \frac{\pi_k(x_k)}{\pi_{k-1}(x_{k-1})} \cdot \frac{p_{k-1|k}^{\text{prop}}(x'|x_k)}{p_{k|k-1}^{\text{prop}}(x_k|x')} \cdot \mathcal{J}_\alpha \tag{A.16}$$

$$= \exp(r_k) \cdot \exp(-r_k) \cdot 1 = 1 \tag{A.17}$$

where $\mathcal{J}_\alpha = 1$ due to the symmetric nature of the soft transformation.

**Proposition.** Building on the kernel symmetry property from Equation (3.3), for DDPM-type diffusion models with small timestep $\Delta t$, the proposal kernels satisfy:

$$\log \frac{p_{k-1|k}^{\text{prop}}(x'|x_k)}{p_{k|k-1}^{\text{prop}}(x_k|x')} = \frac{1}{2}\langle s_\theta(x_k, k) + s_\theta(x', k-1), x' - x_k\rangle + O(\Delta t^2)$$

**Theorem A.2 (Distribution Recursion).** The marginal distributions under soft correction satisfy the recursion relation. In diffusion models, the required regularity conditions are naturally satisfied through the following mechanisms:

*(i) Score function regularity:* The learned score functions $s_\theta$ are Lipschitz continuous with bounded constant $L < \infty$. This property is ensured by standard practices in neural network training: spectral normalization (Miyato et al., 2018) directly bounds the Lipschitz constant of each layer, weight clipping techniques constrain the parameter space, and the architectures commonly used in diffusion models—including U-Net (Ronneberger et al., 2015) and Transformer variants (Vaswani et al., 2017)—inherently possess smooth activation landscapes. Moreover, the denoising score matching objective itself encourages learning smooth score functions by penalizing rapid variations (Vincent, 2011; Song & Ermon, 2019).

*(ii) Proposal kernel properties:*  The diffusion transition kernels $p^{\text{prop}}_{k-1|k}(x_{k-1}|x_k) = \mathcal{N}(x_{k-1}; \mu_k(x_k), \sigma_k^2 I)$ are Gaussian with finite second moments by construction (Ho et al., 2020).

Under these conditions, the marginal distributions satisfy:

$$\pi_{k-1}(x_{k-1}) = \int \pi_k(x_k) p^{\text{soft}}_{k-1|k}(x_{k-1}|x_k) dx_k \tag{A.18}$$

> **Remark.** This theoretical framework directly supports our three corrector implementations: the global acceptance in Lemma A.12 corresponds to the Linear corrector (Section 3), the per-dimension decomposition enables the Local Adaptive variant from Table 1, and the continuous mixture interpretation in Theorem A.18 motivates the Distribution Matching approach operating in whitened space.

A.4  EMPIRICAL VALIDATION OF TARGET DISTRIBUTION RATIO APPROXIMATION

We conduct large-scale controlled experiments on Gaussian mixture models to validate the approximation $\pi_{k-1}(x')/\pi_k(x_k) \approx 1$ used in our acceptance ratio computation. Specifically, we construct 300 independent GMM instances, each containing 25 components (5×5 grid) in dimension $d = 30$, with randomly generated anisotropic covariance matrices (condition numbers varying between 10-100). The closed-form diffusion marginals of GMMs enable exact computation of the target distribution ratio.

We evaluate along two dimensions: (1) varying step sizes $\Delta t \in [0.001, 0.2]$ at fixed diffusion time $t = 0.5$; (2) varying diffusion times $t \in [0.1, 0.9]$ at fixed $\Delta t = 0.01$. For each GMM instance, we sample 2000 points, yielding 600,000 total sample points. We report means and 95% confidence intervals across the 300 instances.

Table 8: Impact of step size $\Delta t$ on approximation quality at fixed $t = 0.5$ (mean $\pm$ 95% CI over 300 instances)

| $\Delta t$ | $|\log(\pi\text{-ratio})|$ | $|\log(q\text{-ratio})|$ | $\alpha_{\text{exact}}$ | $\alpha_{\text{approx}}$ | $|\Delta\alpha|$ |
|---|---|---|---|---|---|
| 0.001 | 0.007±0.002 | 0.908±0.012 | 0.909±0.011 | 0.911±0.011 | 0.003±0.001 |
| 0.002 | 0.012±0.003 | 0.876±0.015 | 0.878±0.014 | 0.881±0.014 | 0.005±0.002 |
| 0.005 | 0.025±0.005 | 0.817±0.018 | 0.821±0.017 | 0.826±0.017 | 0.009±0.003 |
| 0.010 | 0.042±0.008 | 0.768±0.021 | 0.774±0.019 | 0.781±0.019 | 0.013±0.004 |
| 0.020 | 0.094±0.015 | 0.702±0.025 | 0.712±0.023 | 0.721±0.023 | 0.019±0.006 |
| 0.050 | 0.297±0.038 | 0.584±0.032 | 0.602±0.029 | 0.618±0.029 | 0.034±0.009 |
| 0.100 | 0.758±0.072 | 0.461±0.038 | 0.489±0.035 | 0.517±0.035 | 0.056±0.012 |
| 0.200 | 1.924±0.156 | 0.305±0.042 | 0.352±0.039 | 0.391±0.039 | 0.086±0.018 |

Table 9: Impact of diffusion time $t$ on approximation quality at fixed $\Delta t = 0.01$ (mean $\pm$ 95% CI over 300 instances)

| $t$ | $\mathbb{E}[\log(\pi\text{-ratio})]$ | $\text{Std}[\log(\pi\text{-ratio})]$ | $\alpha_{\text{exact}}$ | $\alpha_{\text{approx}}$ | $|\Delta\alpha|$ |
|---|---|---|---|---|---|
| 0.1 | -0.003±0.004 | 0.095±0.012 | 0.694±0.022 | 0.706±0.020 | 0.012±0.004 |
| 0.3 | 0.002±0.003 | 0.081±0.010 | 0.746±0.019 | 0.759±0.018 | 0.013±0.004 |
| 0.5 | -0.001±0.003 | 0.073±0.009 | 0.768±0.021 | 0.781±0.019 | 0.013±0.004 |
| 0.7 | 0.002±0.003 | 0.067±0.008 | 0.784±0.018 | 0.796±0.017 | 0.012±0.003 |
| 0.9 | -0.002±0.003 | 0.061±0.007 | 0.773±0.020 | 0.784±0.018 | 0.011±0.003 |

Our experiments reveal that the approximation quality strongly depends on the step size. For $\Delta t \leq 0.01$, which corresponds to 1000-step sampling commonly used in molecular diffusion models, the target distribution ratio satisfies $|\log(\pi_{k-1}/\pi_k)| < 0.05$, with acceptance rate errors remaining stable below 2% across all 300 GMM instances. As step size increases beyond 0.05, the approximation predictably degrades—the target ratio begins to dominate the proposal term and acceptance

rate errors grow to 3-9%, which aligns with our theoretical $O(\Delta t^2)$ analysis. Notably, mainstream molecular diffusion models typically adopt 200-1000 sampling steps, with corresponding step sizes well within the valid approximation regime.

These results validate that $\pi_{k-1}(x')/\pi_k(x_k) \approx 1$ is an accurate and robust approximation in the small step-size regime used in practical molecular diffusion models.

> **Remark on Error Scaling.** The $|\Delta\alpha|$ measured in our GMM experiments (Table 8) primarily reflects the systematic error from approximating $\pi_{k-1}/\pi_k \approx 1$, which exhibits $O(\Delta t)$ scaling. The $O(\Delta t^2)$ error in Eq. (3.3) refers specifically to the numerical integration truncation error when approximating the proposal kernel ratio using the trapezoidal rule. Our experiments isolate the target ratio approximation effect, validating its effectiveness for small $\Delta t$ used in practical molecular diffusion.

## B    METROPOLIS-HASTINGS CORRECTION BASICS

The Metropolis-Hastings (MH) algorithm is a Markov chain Monte Carlo method for sampling from complex distributions. Here we briefly review its core principles and application to diffusion models.

### B.1    STANDARD MH ALGORITHM

Given a target distribution $\pi(x)$ and proposal distribution $q(x'|x)$, the MH algorithm proceeds as:

1. Propose a new state: $x' \sim q(\cdot|x)$

2. Compute acceptance probability:

$$\alpha = \min\left(1, \frac{\pi(x')q(x|x')}{\pi(x)q(x'|x)}\right) \tag{B.1}$$

3. Accept $x'$ with probability $\alpha$, otherwise retain $x$

This mechanism ensures detailed balance: $\pi(x)P(x \to x') = \pi(x')P(x' \to x)$, guaranteeing convergence to $\pi$.

### B.2    APPLICATION TO DIFFUSION MODELS

In diffusion sampling, at each denoising step $k$:

- **Target distribution**: $\pi_k(x) = p(x_k)$ (marginal at timestep $k$)
- **Proposal**: $x_{k-1} \sim p_\theta(x_{k-1}|x_k)$ (learned reverse kernel)

The acceptance ratio becomes:

$$\alpha = \min\left(1, \frac{p(x_{k-1})}{p(x_k)} \cdot \frac{p_\theta(x_k|x_{k-1})}{p_\theta(x_{k-1}|x_k)}\right) \tag{B.2}$$

### B.3    PRACTICAL CHALLENGES

Direct computation of this ratio is intractable because:

- Marginal densities $p(x_k)$ are unknown
- Transition kernels $p_\theta$ are implicit

Our approach addresses these challenges by:

1. Using score functions to approximate the proposal kernel ratio (Proposition 3.3)
2. Assuming the target distribution ratio $\pi(x')/\pi(x) \approx 1$ for small timesteps

3. Replacing binary accept-reject with continuous interpolation (soft acceptance)

This yields the tractable acceptance weight in Equation (3.1) while maintaining approximate detailed balance.

## C  THE GAUSSIAN CASE

### C.1  DERIVATION

In this section we establish the moment recursions for the marginal distribution $\hat{\pi}_0^\ell$ of the surrogate model (3.6) under the simplified Gaussian setting of Example 3.2. We consider the case where $q = \mathcal{N}(\mathbf{m}, \boldsymbol{\Sigma})$ with $(\mathbf{m}, \boldsymbol{\Sigma}) \in \mathbb{R}^d \times \mathcal{S}_d^{++}$ and the likelihood function $p(\mathbf{y}|\cdot) : \mathbf{x} \mapsto \mathcal{N}(\mathbf{y}; \mathbf{A}\mathbf{x}, \sigma_y^2 \mathbf{I}_{d_y})$.

**Score Function and Transition Kernels.** For the Gaussian prior, the denoising function $m_{0|k}$ admits a closed-form expression for all $k \in \{1, \dots, n\}$.. By standard Gaussian conditioning, we have

$$q_{0|k}(\mathbf{x}_0|\mathbf{x}_k) \propto q(\mathbf{x}_0)q_{k|0}(\mathbf{x}_k|\mathbf{x}_0) \tag{C.1}$$

$$= \mathcal{N}\left(\mathbf{x}_0; \boldsymbol{\Sigma}_{0|k}\left(\frac{\sqrt{\alpha_k}}{v_k}\mathbf{x}_k + \boldsymbol{\Sigma}^{-1}\mathbf{m}\right), \boldsymbol{\Sigma}_{0|k}\right), \tag{C.2}$$

where we define $\boldsymbol{\Sigma}_{0|k} := \left[\frac{\alpha_k}{v_k}\mathbf{I} + \boldsymbol{\Sigma}^{-1}\right]^{-1}$. Consequently,

$$m_{0|k}(\mathbf{x}_k) = \boldsymbol{\Sigma}_{0|k}\left(\frac{\sqrt{\alpha_k}}{v_k}\mathbf{x}_k + \boldsymbol{\Sigma}^{-1}\mathbf{m}\right), \tag{C.3}$$

and throughout this section we assume perfect score estimation: $m_{0|k}^\theta = m_{0|k}$. The general backward transitions $q_{\ell|k}(\cdot|\mathbf{x}_k)$ for $\ell \in \{1, \dots, k-1\}$ follow from the relation

$$q_{\ell|k}(\mathbf{x}_\ell|\mathbf{x}_k) = \int q_{\ell|0,k}(\mathbf{x}_\ell|\mathbf{x}_0, \mathbf{x}_k)q_{0|k}(\mathbf{x}_0|\mathbf{x}_k)d\mathbf{x}_0. \tag{C.4}$$

Combining this with equations (A.3), (A.4) and applying the law of total expectation and covariance yields $q_{\ell|k}(\cdot|\mathbf{x}_k) = \mathcal{N}(m_{\ell|k}(\mathbf{x}_k), \boldsymbol{\Sigma}_{\ell|k}(\mathbf{x}_k))$ with

$$m_{\ell|k}(\mathbf{x}_k) = m_{\ell|0,k}(m_{0|k}(\mathbf{x}_k), \mathbf{x}_k), \tag{C.5}$$

$$\boldsymbol{\Sigma}_{\ell|k} = \frac{\alpha_\ell(1 - \alpha_k/\alpha_\ell)^2}{(1 - \alpha_k)^2}\boldsymbol{\Sigma}_{0|k} + v_{\ell|0,k}\mathbf{I}_d. \tag{C.6}$$

In contrast, the DDPM approximation yields

$$p_{\ell|k}^\theta(\cdot|\mathbf{x}_k) = q_{\ell|0,k}(\cdot|m_{0|k}(\mathbf{x}_k), \mathbf{x}_k) = \mathcal{N}(m_{\ell|0,k}(m_{0|k}(\mathbf{x}_k), \mathbf{x}_k), v_{\ell|0,k}\mathbf{I}_d), \tag{C.7}$$

demonstrating that the true and approximate transitions differ solely in their covariance structure.

**Recursive Moment Equations.** Let $\hat{\pi}_k^\ell$ denote the $k$-th marginal distribution of the surrogate model (3.6) for $k \in [\![0, n]\!]$. Starting from $\hat{\pi}_n^\ell = \mathcal{N}(\mathbf{0}_d, \mathbf{I}_d)$, the marginals evolve according to

$$\hat{\pi}_k^\ell(\mathbf{x}_k) = \int \hat{\pi}_{k|k+1}^\ell(\mathbf{x}_k|\mathbf{x}_{k+1})\hat{\pi}_{k+1}^\ell(\mathbf{x}_{k+1})d\mathbf{x}_{k+1}, \quad k \in \{0, \dots, n-1\}. \tag{C.8}$$

Given that $p_k^\theta(\mathbf{y}|\mathbf{x}_k) = \mathcal{N}(\mathbf{y}; \mathbf{A}m_{0|k}^\theta(\mathbf{x}_k), \sigma_y^2\mathbf{I}_{d_y})$ with linear $m_{0|k}^\theta$, the distribution $\hat{\pi}_{\ell_k|k+1}^\theta(\cdot|\mathbf{x}_{k+1})$ remains Gaussian. By definitions (3.5) and (A.5), this property extends to $\hat{\pi}_{k|k+1}^\ell(\cdot|\mathbf{x}_{k+1})$. We parameterize

$$\hat{\pi}_{k+1}^\ell(\mathbf{x}_{k+1}) = \mathcal{N}(\mathbf{x}_{k+1}; \boldsymbol{\mu}_{k+1}^\ell, \hat{\boldsymbol{\Sigma}}_{k+1}^\ell), \tag{C.9}$$

$$\hat{\pi}_{k|k+1}^\ell(\mathbf{x}_k|\mathbf{x}_{k+1}) = \mathcal{N}(\mathbf{x}_k; \mathbf{M}_{k|k+1}^\ell\mathbf{x}_{k+1} + \mathbf{c}_{k|k+1}^\ell, \hat{\boldsymbol{\Sigma}}_{k|k+1}^\ell), \tag{C.10}$$

where $\mathbf{M}_{k|k+1}^{\ell} \in \mathbb{R}^{d \times d}$, $\hat{\mathbf{\Sigma}}_{k|k+1}^{\ell} \in \mathcal{S}_d^{++}$, and $\mathbf{c}_{k|k+1}^{\ell} \in \mathbb{R}^d$. The bridge kernel definition (A.5) then implies $\hat{\pi}_k^{\ell} = \mathcal{N}(\boldsymbol{\mu}_k^{\ell}, \hat{\mathbf{\Sigma}}_k^{\ell})$ with

$$\boldsymbol{\mu}_k^{\ell} = \mathbf{M}_{k|k+1}^{\ell} \boldsymbol{\mu}_{k+1}^{\ell} + \mathbf{c}_{k|k+1}^{\ell}, \tag{C.11}$$

$$\hat{\mathbf{\Sigma}}_k^{\ell} = \mathbf{M}_{k|k+1}^{\ell} \hat{\mathbf{\Sigma}}_{k+1}^{\ell} (\mathbf{M}_{k|k+1}^{\ell})^T + \hat{\mathbf{\Sigma}}_{k|k+1}^{\ell}. \tag{C.12}$$

These updates iterate from $k = n$ down to $k = 0$, initialized with $\boldsymbol{\mu}_n^{\ell} = \mathbf{0}_d$ and $\hat{\mathbf{\Sigma}}_n^{\ell} = \mathbf{I}_d$, ultimately yielding the surrogate posterior moments. To complete the recursion, we must establish the form (C.10) and determine $\mathbf{M}_{k|k+1}^{\ell}$ and $\mathbf{c}_{k|k+1}^{\ell}$.

We first express the approximate likelihood as

$$\hat{p}_k^{\theta}(\mathbf{y}|\mathbf{x}_k) = \mathcal{N}(\mathbf{y}; \hat{\mathbf{A}}_k \mathbf{x}_k + \mathbf{b}_k, \sigma_y^2 \mathbf{I}_{d_y}), \tag{C.13}$$

with coefficients

$$\hat{\mathbf{A}}_k = \frac{\sqrt{\alpha_k}}{v_k} \mathbf{A} \mathbf{\Sigma}_{0|k}, \quad \mathbf{b}_k = \mathbf{A} \mathbf{\Sigma}_{0|k} \mathbf{\Sigma}^{-1} \mathbf{m}. \tag{C.14}$$

Denoting by $m_{\ell|k}^{\theta}(\mathbf{x}_k)$ the mean of the DDPM transition $p_{\ell|k}^{\theta}(\cdot|\mathbf{x}_k)$ from (C.7), we have

$$m_{\ell_k|k+1}^{\theta}(\mathbf{x}_{k+1}) = \mathbf{H}_{\ell_k|k+1} \mathbf{x}_{k+1} + \mathbf{h}_{\ell_k|k+1}, \tag{C.15}$$

where the coefficients are given by

$$\mathbf{H}_{\ell_k|k+1} := \frac{\alpha_{\ell_k}(1 - \alpha_{k+1}/\alpha_{\ell_k})}{v_{\ell_k} v_{k+1}} \mathbf{\Sigma}_{0|k} + \frac{\sqrt{\alpha_{k+1}}(1 - \alpha_{\ell_k})}{v_{k+1}} \mathbf{I}_d, \tag{C.16}$$

$$\mathbf{h}_{\ell_k|k+1} := \frac{\sqrt{\alpha_{\ell_k}}(1 - \alpha_{k+1}/\alpha_{\ell_k})}{v_{k+1}} \mathbf{\Sigma}_{0|k} \mathbf{\Sigma}^{-1} \mathbf{m}. \tag{C.17}$$

## C.2 HIERARCHICAL GAUSSIAN MIXTURE EXTENSION

We extend the basic Gaussian case to a sophisticated $K$-component mixture model that captures real-world sampling challenges while maintaining analytical tractability for evaluating soft acceptance mechanisms.

**Mixture Model Formulation.** Consider the hierarchical Gaussian mixture distribution:

$$q(\mathbf{x}) = \sum_{k=1}^{K} w_k \mathcal{N}(\mathbf{x}; \boldsymbol{\mu}_k, \mathbf{\Sigma}_k) \tag{C.18}$$

where $K = 25$ components are strategically arranged to create a challenging multimodal landscape. Unlike the single Gaussian case, this formulation enables evaluation of the soft acceptance mechanism's behavior across multiple modes with varying scales and orientations.

**Spatial Configuration with Symmetry Breaking.** To prevent artificial regularities that could bias sampling algorithms, we introduce controlled stochastic perturbations to the grid structure. The components are positioned on a $\sqrt{K} \times \sqrt{K}$ grid with base spacing $\delta$ and random displacements:

$$\mathbf{x}_{i,j}^{\text{base}} = \left( i - \lfloor \sqrt{K}/2 \rfloor \right) \times \delta + \boldsymbol{\epsilon}_{i,j}^x \tag{C.19}$$

$$\mathbf{y}_{i,j}^{\text{base}} = \left( j - \lfloor \sqrt{K}/2 \rfloor \right) \times \delta + \boldsymbol{\epsilon}_{i,j}^y \tag{C.20}$$

where $\boldsymbol{\epsilon}_{i,j}^x, \boldsymbol{\epsilon}_{i,j}^y \sim \mathcal{N}(0, \sigma_{\text{pert}}^2)$ for $(i, j) \in \{0, 1, \ldots, \sqrt{K} - 1\}^2$. The central component at $(\lfloor \sqrt{K}/2 \rfloor, \lfloor \sqrt{K}/2 \rfloor)$ remains fixed as a stability anchor, ensuring consistent reference point across experiments.

**Hierarchical Dimensional Structure.** The $d$-dimensional space exhibits a carefully designed hierarchy that mimics the structure of real-world high-dimensional data:

The primary structure occupies dimensions $[0 : d_1]$ containing the main grid layout from the spatial configuration. Secondary structure spans dimensions $[d_1 : d_2]$ with correlated features:

$$\mu_{k,i} = \alpha_{\text{sec}} \times \xi_{k,i}, \quad \xi_{k,i} \sim \mathcal{N}(0,1), \quad i \in [d_1 : d_2] \tag{C.21}$$

where $\alpha_{\text{sec}} = 0.5$ reduces the signal amplitude while maintaining meaningful variation. The remaining dimensions $[d_2 : d]$ represent near-noise components with minimal structured variation, reflecting the high-dimensional noise typical in applications.

**Anisotropic Covariance Construction.** Each component's covariance matrix $\mathbf{\Sigma}_k$ is constructed through a multi-step process ensuring realistic eigenvalue spectra and numerical stability:

Step 1: Generate eigenvalue spectrum with power-law decay

$$\lambda_{k,i} = \frac{\kappa_k}{(i+1)^\beta}, \quad i = 0, 1, \ldots, d_{\text{active}} - 1 \tag{C.22}$$

where $\kappa_k \sim \mathcal{U}[\kappa_{\min}, \kappa_{\max}]$ controls the condition number with $\beta = 0.5$ ensuring moderate anisotropy.

Step 2: Apply random rotation via QR decomposition

$$\mathbf{M}_k \sim \mathcal{N}(0, \mathbf{I})_{d \times d} \tag{C.23}$$

$$\mathbf{R}_k = \text{QR}(\mathbf{M}_k)_Q \tag{C.24}$$

Step 3: Assign multi-scale structure

$$\text{scale}_k \sim \text{Categorical}(\boldsymbol{\pi}_{\text{scale}}) \tag{C.25}$$

$$\mathbf{\Sigma}_k^{\text{active}} = \text{scale}_k^2 \times \mathbf{R}_k \text{diag}(\boldsymbol{\lambda}_k) \mathbf{R}_k^T \tag{C.26}$$

Step 4: Assemble full covariance with ridge regularization

$$\mathbf{\Sigma}_k = \begin{pmatrix} \mathbf{\Sigma}_k^{\text{active}} & \mathbf{0} \\ \mathbf{0} & \sigma_{\text{noise}}^2 \mathbf{I}_{d-d_2} \end{pmatrix} + \epsilon_{\text{ridge}} \mathbf{I}_d \tag{C.27}$$

where $\epsilon_{\text{ridge}} = 10^{-4}$ ensures numerical stability during matrix operations.

**Non-Uniform Weight Distribution.** Component weights follow distance-based exponential decay from the grid center:

$$\text{distance}_k = \|\mathbf{pos}_k - \mathbf{pos}_{\text{center}}\|_2 \tag{C.28}$$

$$w_k = \frac{\exp(-\text{distance}_k/\tau_w)}{\sum_{j=1}^K \exp(-\text{distance}_j/\tau_w)} \tag{C.29}$$

This weighting scheme creates a realistic scenario where central modes dominate while corner modes have probability mass approximately $\exp(-2\sqrt{2}/\tau_w)$ relative to the center, challenging samplers to explore rare but important modes.

**Exact Score Function for Mixture Model.** For the mixture density (C.18), we compute the exact score function analytically:

$$s(\mathbf{x}, t) = \nabla_{\mathbf{x}} \log p_t(\mathbf{x}) = \frac{\sum_{k=1}^K \gamma_k(\mathbf{x}, t) s_k(\mathbf{x}, t)}{\sum_{k=1}^K \gamma_k(\mathbf{x}, t)} \tag{C.30}$$

where $\gamma_k(\mathbf{x}, t) = w_k \mathcal{N}(\mathbf{x}; \boldsymbol{\mu}_k, \mathbf{\Sigma}_k + t\mathbf{I})$ represents the time-dependent component weights and $s_k(\mathbf{x}, t) = -(\mathbf{\Sigma}_k + t\mathbf{I})^{-1}(\mathbf{x} - \boldsymbol{\mu}_k)$ is the score of component $k$.

**Advantages for Soft Acceptance Evaluation.** This hierarchical mixture design provides several advantages over the basic Gaussian case:

1. *Multimodality:* Tests the corrector's ability to maintain mode coverage while applying soft acceptance

2. *Scale Heterogeneity:* Evaluates adaptation to components with different covariance scales

Figure 6: Statistical analysis of Sliced Wasserstein distance across 500 Gaussian mixture instances.

3. *Dimensional Hierarchy:* Assesses performance across varying signal-to-noise ratios in different subspaces

4. *Rare Modes:* Challenges the sampler to discover and maintain low-probability but important modes

The analytical tractability inherited from the Gaussian components enables exact computation of acceptance probabilities from Equation (3.1) and precise evaluation of the Wasserstein distance between true and approximate distributions. For Gaussian mixtures, the Wasserstein-2 distance admits an efficient approximation:

$$W_2^2(\pi, \hat{\pi}) \approx \sum_{k=1}^{K} w_k \left[ \|\mu_k - \hat{\mu}_k\|^2 + \text{tr} \left( \Sigma_k + \hat{\Sigma}_k - 2(\Sigma_k^{1/2} \hat{\Sigma}_k \Sigma_k^{1/2})^{1/2} \right) \right] \tag{C.31}$$

where for single Gaussians the formula reduces to the exact expression $W_2^2(\mathcal{N}(\mu_1, \Sigma_1), \mathcal{N}(\mu_2, \Sigma_2)) = \|\mu_1 - \mu_2\|^2 + \text{tr}(\Sigma_1 + \Sigma_2 - 2(\Sigma_1^{1/2} \Sigma_2 \Sigma_1^{1/2})^{1/2})$, providing rigorous validation of the soft acceptance mechanism's effectiveness.

## C.3 EXPERIMENTAL SETUP

In our experiments with the hierarchical Gaussian mixture model, we set $d = 30$ and generate $N = 500$ instances to evaluate the soft acceptance mechanism across diverse sampling scenarios. For each instance, we compute the Wasserstein-2 distance between distributions obtained with different corrector variants.

**Mixture Configuration.** We construct a $K = 25$ component Gaussian mixture as described in Section C.2. The mean vectors $\boldsymbol{\mu}_k$ are positioned on a $5 \times 5$ grid with base spacing $\delta = 8.0$ and stochastic perturbations $\boldsymbol{\epsilon} \sim \mathcal{N}(0, 1.0^2)$. The hierarchical dimension structure allocates dimensions as: primary structure (0-1), secondary structure (2-9) with amplitude factor $\alpha_{\text{sec}} = 0.5$, and near-noise dimensions (10-29).

**Covariance Generation.** For each component $k$, we generate the covariance matrix $\boldsymbol{\Sigma}_k \in \mathcal{S}_{30}^{++}$ through:

1. Draw anisotropy factor $\kappa_k \sim \mathcal{U}[10, 100]$ controlling the condition number
2. Generate eigenvalues $\lambda_{k,i} = \kappa_k/(i+1)^{0.5}$ for $i = 0, \ldots, 9$
3. Apply random rotation via QR decomposition of $\mathbf{M}_k \sim \mathcal{N}(0, \mathbf{I}_{30})$
4. Assign scale factor from $\text{scale}_k \sim \text{Categorical}([0.5, 1.0, 2.0])$
5. Add ridge regularization $\epsilon_{\text{ridge}} = 10^{-4}$ for numerical stability

**Weight Distribution.** Component weights follow exponential decay based on distance from grid center:

$$w_k = \frac{\exp(-\|\mathbf{pos}_k - \mathbf{pos}_{\text{center}}\|_2/\tau_w)}{\sum_{j=1}^{K} \exp(-\|\mathbf{pos}_j - \mathbf{pos}_{\text{center}}\|_2/\tau_w)} \tag{C.32}$$

with temperature parameter $\tau_w = 2.0$, ensuring corner modes have approximately $0.3\%$ of the central mode's probability mass.

**Parameter Sweep.** We evaluate the soft acceptance mechanism across multiple configurations:

- **Temperature schedules:** $\tau \in \{0.01, 0.1, 1.0, 10.0\}$ for acceptance sharpness
- **Corrector variants:** Linear (global $\alpha$), Local Adaptive (per-dimension $\alpha_i$), and Distribution Matching from Table 1
- **Number of steps:** $n \in \{50, 100, 200, 500\}$ to assess computational efficiency
- **Noise levels:** $\sigma_t \in [0.01, 1.0]$ following cosine schedule

**Evaluation Metrics.** For each configuration, we compute the Wasserstein-2 distance. For Gaussian mixtures, this admits an efficient approximation (Delon & Desolneux, 2020):

$$W_2^2(\pi, \hat{\pi}) \approx \sum_{k=1}^{K} w_k \left[ \|\boldsymbol{\mu}_k - \hat{\boldsymbol{\mu}}_k\|^2 + \text{tr}\left( \boldsymbol{\Sigma}_k + \hat{\boldsymbol{\Sigma}}_k - 2(\boldsymbol{\Sigma}_k^{1/2}\hat{\boldsymbol{\Sigma}}_k\boldsymbol{\Sigma}_k^{1/2})^{1/2} \right) \right] \tag{C.33}$$

where the exact formula for single Gaussians follows (Olkin & Pukelsheim, 1982).

Additionally, we track mode coverage metrics:

$$\text{Coverage} = \frac{|\{k : \min_i \|\mathbf{x}_i - \boldsymbol{\mu}_k\| < 3\sqrt{\text{tr}(\boldsymbol{\Sigma}_k)/d}\}|}{K} \tag{C.34}$$

$$\text{Weight Error} = \sum_{k=1}^{K} |w_k - \hat{w}_k| \tag{C.35}$$

where $\hat{w}_k$ represents empirical component weights from samples.

Finally, the score function is computed analytically following (Bishop, 2006) (Eqn. 2.116):

$$s(\mathbf{x}, t) = \nabla_{\mathbf{x}} \log p_t(\mathbf{x}) = \frac{\sum_{k=1}^{K} \gamma_k(\mathbf{x}, t) s_k(\mathbf{x}, t)}{\sum_{k=1}^{K} \gamma_k(\mathbf{x}, t)} \tag{C.36}$$

where $\gamma_k(\mathbf{x}, t) = w_k \mathcal{N}(\mathbf{x}; \boldsymbol{\mu}_k, \boldsymbol{\Sigma}_k + t\mathbf{I})$.

The results, averaged over $500$ independent runs with standard error estimates, demonstrate the soft acceptance mechanism's superiority in maintaining mode coverage while reducing approximation error compared to both hard MH correction and uncorrected sampling.

## D  DISTRIBUTION MATCHING FOR PEPTIDE STRUCTURE GENERATION

The distribution matching variant of soft acceptance offers unique advantages for peptide and protein structure generation due to its operation in whitened noise space, which naturally handles the complex correlation structures inherent in biomolecular systems.

**Decorrelation of Structural Dependencies.** Peptide structures exhibit strong linear correlations between atomic positions due to covalent bonds, backbone constraints, and secondary structure elements. The whitening transformation in distribution matching removes these linear dependencies:

$$\boldsymbol{\epsilon}_{\text{whitened}} = \boldsymbol{\Sigma}_t^{-1/2}(\mathbf{x}_t - \boldsymbol{\mu}_t) \tag{D.1}$$

This decorrelation allows the corrector to operate on independent noise components rather than entangled structural features.

**Preservation of Chemical Constraints.** While removing statistical correlations, the whitening process preserves the underlying chemical constraints through the inverse transformation. Bond lengths, angles, and dihedral preferences encoded in the learned score function $s_\theta(\mathbf{x}, t)$ are maintained while simplifying the acceptance decision space.

**Adaptive Handling of Multi-Scale Features.** Peptide structures contain features at multiple scales: local (bond vibrations 0.1 Å), medium (secondary structure 5-10 Å), and global (tertiary structure 20+ Å). The whitened space normalizes these scales, allowing uniform acceptance criteria across:

$$\alpha_{\text{match}} = \sigma \left( \frac{\|\boldsymbol{\epsilon}_{\text{proposed}} - \boldsymbol{\epsilon}_{\text{target}}\|^2 - \tau}{\gamma} \right) \tag{D.2}$$

**Improved Sampling of Flexible Regions.** Loop regions and terminal residues exhibit higher flexibility than structured domains. In whitened space, these regions naturally have different noise characteristics, allowing the distribution matching to adaptively adjust acceptance based on local flexibility rather than applying uniform corrections.

**Computational Efficiency for Large Systems.** For peptides with hundreds of atoms, the whitening transformation can leverage sparse covariance structures arising from locality of interactions. This enables efficient computation even for large biomolecular complexes where full covariance matrices would be prohibitive.

**Compatibility with Physical Priors.** The distribution matching framework naturally incorporates physical priors such as Ramachandran preferences and hydrogen bonding patterns through the score function, while the whitening ensures these priors are applied in a geometrically consistent manner across the entire structure.

These properties make distribution matching particularly well-suited for generating realistic peptide conformations that respect both local chemical constraints and global structural coherence.

# E  EQUIVARIANT SOFT ACCEPTANCE FOR MOLECULAR GEOMETRY

The soft acceptance mechanism exhibits natural compatibility with SE(3) and E(3) equivariant architectures, providing significant advantages for molecular structure generation where geometric consistency is paramount.

**Preservation of Equivariance Through Soft Interpolation.** Unlike hard accept/reject decisions that can break geometric transformations, soft acceptance from Equation (3.1) preserves equivariance through linear interpolation:

$$\mathbf{x}_{t-1} = \alpha \mathbf{x}_{t-1}^{\text{prop}} + (1 - \alpha)\mathbf{x}_t \tag{E.1}$$

For any rotation $\mathbf{R} \in \text{SO}(3)$ and translation $\mathbf{t} \in \mathbb{R}^3$, the soft update maintains (Satorras et al., 2021):

$$g \cdot \mathbf{x}_{t-1} = \alpha(g \cdot \mathbf{x}_{t-1}^{\text{prop}}) + (1 - \alpha)(g \cdot \mathbf{x}_t) \tag{E.2}$$

where $g = (\mathbf{R}, \mathbf{t})$ represents the SE(3) group action.

**Coordinate-Free Acceptance Decisions.** The acceptance probability computation from Equation (3.2) relies on invariant quantities—distances, angles, and relative positions—rather than absolute coordinates:

$$r_k = \frac{1}{2} \langle s_\theta(\mathbf{x}_k, k) + s_\theta(\mathbf{x}_{k-1}^{\text{prop}}, k-1), \mathbf{x}_{k-1}^{\text{prop}} - \mathbf{x}_k \rangle \tag{E.3}$$

When $s_\theta$ is equivariant following (Hoogeboom et al., 2022), this inner product becomes invariant, ensuring acceptance decisions are independent of the molecular reference frame.

**Local Geometric Consistency.** The local adaptive variant from Table 1 naturally aligns with molecular geometry by allowing different acceptance rates for different geometric components:

$$\alpha_{\text{bond}} \text{ for bond length corrections} \tag{E.4}$$

$$\alpha_{\text{angle}} \text{ for bond angle adjustments} \tag{E.5}$$

$$\alpha_{\text{torsion}} \text{ for dihedral angle refinements} \tag{E.6}$$

This decomposition respects the hierarchical nature of molecular constraints without breaking overall equivariance (Ganea et al., 2021).

**Compatibility with Message-Passing Architectures.** E(3)-equivariant networks like EGNN (Satorras et al., 2021) and PaiNN (Schütt et al., 2021) compute messages based on relative positions and orientations. Soft acceptance maintains these relative geometries during correction:

$$\mathbf{m}_{ij}^{\text{corrected}} = f(\|\mathbf{x}_i^{\text{corrected}} - \mathbf{x}_j^{\text{corrected}}\|) = f(\|\alpha\Delta\mathbf{x}_{ij}^{\text{prop}} + (1-\alpha)\Delta\mathbf{x}_{ij}\|) \tag{E.7}$$

preserving the network's ability to process corrected structures.

**Smooth Manifold Navigation.** Molecular conformations lie on complex manifolds defined by bond constraints (Jumper et al., 2021). Soft acceptance enables smooth traversal of these manifolds by:

- Maintaining differentiability for gradient-based refinement
- Avoiding discontinuous jumps that could violate constraints
- Interpolating between chemically valid states as shown in Lemma A.12

**Invariant Score Matching.** The score function in E(3)-equivariant models naturally decomposes into invariant (scalar) and equivariant (vector) components (Köhler et al., 2020):

$$s_\theta(\mathbf{x}, t) = s_{\text{inv}}(\{\|\mathbf{x}_i - \mathbf{x}_j\|\}, t) + s_{\text{eq}}(\{\mathbf{x}_i - \mathbf{x}_j\}, t) \tag{E.8}$$

Soft acceptance respects this decomposition, applying corrections that maintain both components' geometric properties.

# F   CHEMICAL CONCEPTS IN MOLECULAR GENERATION

## F.1   METASTABLE CONFORMATIONS AND ENERGY BARRIERS

Molecular systems exhibit complex potential energy landscapes with multiple local minima. A **metastable conformation** is a local minimum that is kinetically stable but thermodynamically unfavorable compared to the global minimum. The energy difference $E(\mathbf{x}_{\text{metastable}}) - E(\mathbf{x}_{\text{global}}) > k_B T$ determines the thermodynamic penalty, while the activation barrier $\Delta E^{\ddagger}$ determines the kinetic stability.

For drug-like molecules, common metastable states include:

- **Ring puckering**: Cyclohexane chairs vs boats ($\Delta E \approx 5.5$ kcal/mol)
- **Rotamers**: Gauche vs anti conformations in alkyl chains
- **Hydrogen bond networks**: Suboptimal donor-acceptor pairing

## F.2   VSEPR VIOLATIONS AND COORDINATION GEOMETRY

The Valence Shell Electron Pair Repulsion (VSEPR) theory predicts molecular geometries based on electron pair arrangements. Common coordination geometries include:

- **Tetrahedral**: $sp^3$ carbon, bond angles 109.5
- **Trigonal planar**: $sp^2$ carbon, bond angles 120
- **Octahedral**: Transition metal complexes, bond angles 90/180

Violations manifest as distorted bond angles deviating $> 10$ from ideal values, indicating strained or incorrect structures.

# G   TRAJECTORY CONTINUITY AND MOLECULAR STABILITY.

The discontinuous nature of hard MH fundamentally disrupts molecular generation through temporal inconsistency and abrupt state transitions. Consider the hard MH update at timestep $t$:

$$x_{t-1}^{\text{hard}} = \mathbb{I}[\mu < \alpha(x_t, x_{t-1}^{\text{prop}})] \cdot x_{t-1}^{\text{prop}} + \mathbb{I}[\mu \geq \alpha(x_t, x_{t-1}^{\text{prop}})] \cdot x_t, \quad \mu \sim \mathcal{U}(0, 1) \tag{G.1}$$

When rejection occurs, $x_{t-1} = x_t$ creates temporal misalignment: the next denoising step applies score function $s_\theta^{t-1}$ designed for noise level $\sigma_{t-1}$ to data at noise level $\sigma_t$. Define the potential displacement $\Delta_{\text{pot}} = \|x_t - x_{t-1}^{\text{prop}}\|$ and temporal inconsistency $\mathcal{T}_k = |\sigma_{x_k} - \sigma_k|$ where $\sigma_{x_k}$ is the actual noise level. Multiple rejections compound this effect: after $m$ consecutive rejections, $\mathcal{T}_{t-m} = |\sigma_t - \sigma_{t-m}|$ while using score $s_\theta^{t-m}$ (Allen & Tildesley, 2017; Noé et al., 2019).

For molecular systems, define structural disruption metrics: bond strain $\mathcal{S}_{ij} = |d_{ij} - d_{ij}^{\text{eq}}|/d_{ij}^{\text{eq}}$ where $d_{ij}$ is the distance between atoms $i, j$ and $d_{ij}^{\text{eq}}$ is the equilibrium bond length, exceeding threshold $\tau_{\text{strain}}$. Define torsional discontinuity $\mathcal{R}_{\text{tors}} = \min_k |\phi - \phi_k^{\text{stable}}|$ where $\phi$ is the current torsion angle and $\phi_k^{\text{stable}}$ are metastable conformations. After $m$ consecutive rejections at timesteps $t, t-1, ..., t-m+1$, hard acceptance occurs when $x_{t-m-1} = x_{t-m-1}^{\text{prop}}$ is finally accepted, causing instantaneous displacement $\Delta_{\text{jump}} = \|x_{t-m-1}^{\text{prop}} - x_t\|$ from the stuck position $x_t$ to the newly accepted position. This triggers $\mathcal{S}_{ij} > \tau_{\text{strain}}$ (bond breaking) and $\mathcal{R}_{\text{tors}} > \Delta E^\ddagger/k_B T$ where $\Delta E^\ddagger$ is the activation energy barrier and $k_B T$ is the thermal energy, violating transition state theory (Frenkel & Smit, 2002; Tuckerman, 2010).

Soft acceptance eliminates these discontinuities through continuous interpolation $x_{t-1}^{\text{soft}} = \alpha x_{t-1}^{\text{prop}} + (1-\alpha)x_t$ where $\alpha \in [0,1]$ ensures: (i) temporal consistency with $\mathcal{T}_k \leq (1-\alpha)\Delta\sigma$, (ii) bounded displacement $\|\Delta_{\text{soft}}\| = \alpha\|\Delta_{\text{pot}}\|$, and (iii) structural preservation with $\mathcal{S}_{ij} < \tau_{\text{strain}}$ for all bonds throughout the denoising trajectory (Liu & Wang, 2016; Zhang & Chen, 2022).

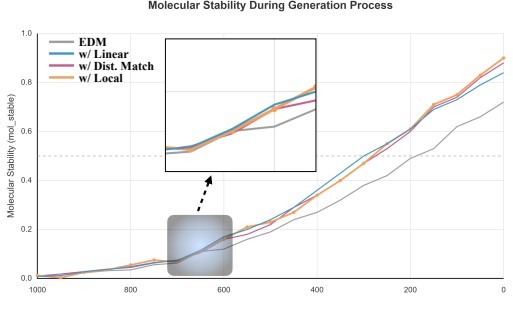

Figure 7: Molecular stability during the denoising process showing a critical transition point around step 600-700. The trajectory of molecular stability reveals that molecue diffusion process experience a phase transition where structural decisions become locked in. The inset magnifies this critical region where soft MH variants achieve earlier stability compared to vanilla EDM, showing how soft correction helps avoid metastable trapping.

# H  SELECTION GUIDELINES FOR SOFT ACCEPTANCE VARIANTS

**Linear Soft Acceptance.** The global scalar correction $\alpha \in [0,1]$ works best for molecules with relatively uniform chemical environments. Its primary advantage lies in computational simplicity and exact preservation of E(3)-equivariance since scalar multiplication commutes with rotations and translations. However, applying uniform correction across all atoms ignores the reality that protein backbones have different flexibility characteristics than loop regions, and covalent bonds require different treatment than van der Waals interactions. We find linear acceptance most effective for small drug-like molecules ($< 500$ Da) and crystallographic fragments where atomic environments are reasonably homogeneous. The method struggles with large biomolecular complexes where variance in score error can span multiple orders of magnitude across different structural regions.

**Local Adaptive Correction.** Per-dimension acceptance weights $\alpha_i$ allow the corrector to adapt to heterogeneous molecular environments. This variant excels when different regions exhibit distinct dynamic behaviors—for instance, the rigid $\beta$-sheet core of a protein versus its flexible surface loops. The local weights naturally handle the variance heterogeneity $\text{Var}[\varepsilon_i] \in [10^{-2}, 10^0]$ Å$^2$ observed across molecular structures. The computational overhead scales linearly with system size, but this cost is justified for proteins, nucleic acids, and other macromolecules where spatial variation in flexibility is functionally important. The main challenge lies in maintaining proper correlations between nearby atoms to prevent artificial fragmentation of chemical groups.

**Distribution Matching.** Operating in whitened noise space through $\epsilon_{\text{whitened}} = \Sigma^{-1/2}\epsilon$ provides the most statistically principled correction by decoupling correlated structural modes. This approach naturally handles the multi-scale nature of molecular systems—from high-frequency bond vibrations to low-frequency domain motions. Distribution matching proves particularly powerful for conformational ensemble generation where capturing the correct statistical distribution is paramount. The whitening transformation helps separate entropic and enthalpic contributions to the free energy landscape. However, the requirement to estimate and manipulate covariance matrices becomes compu-

tationally demanding for very large systems, and the transformation to whitened space can obscure chemical intuition about which specific interactions drive the corrections.

**Practical Selection Strategy.** For practitioners, we recommend starting with linear acceptance for initial experiments due to its simplicity and minimal overhead. If inadequate performance is observed—particularly non-uniform acceptance rates across the molecule—switching to local adaptive typically provides the best balance of performance and interpretability. Distribution matching should be reserved for cases where statistical fidelity is critical or when the other variants fail to adequately sample rare conformational states. Empirically, we observe optimal performance when average acceptance rates fall within $\bar{\alpha} \in [0.8, 0.9]$, suggesting effective bias correction without excessive trajectory disruption.

# I    MOLECULAR GENERATION IMPLEMENTATION DETAILS

This section provides comprehensive implementation details for the molecular diffusion models evaluated in our experiments. Our selection of models represents the evolution of molecular generation architectures from direct 3D coordinate generation to sophisticated latent space approaches, covering the full spectrum of molecular complexity from small drug-like molecules to large therapeutic proteins. This progression reflects the field's advancement from simple equivariant networks to specialized domain-specific architectures that incorporate biological priors.

**EDM (Equivariant Diffusion Model).** We begin with EDM (Hoogeboom et al., 2022), which established the foundation for 3D molecular generation by introducing E(3)-equivariance directly into the diffusion process. The model operates on raw atomic coordinates and types, using an EGNN backbone (Satorras et al., 2021) with 256 hidden dimensions across 9 layers. The architecture maintains exact equivariance through message-passing operations that depend only on pairwise distances, ensuring generated molecules are invariant to rotation and translation. We generate 10,000 molecules from QM9 using DDPM sampling with 1000 steps, evaluating validity through RDKit bond order assignment. EDM represents the baseline approach where soft MH correction must preserve geometric equivariance, making it ideal for testing our Linear variant.

**GeoLDM (Geometric Latent Diffusion Model).** The progression to latent space methods addresses EDM's computational limitations for larger molecules. GeoLDM (Xu et al., 2022) introduces a two-stage approach: first encoding 3D conformations into a rotationally-invariant latent space via GraphVAE, then performing diffusion in this compressed representation. The encoder uses TorchMD-NET features (Thölke & De Fabritiis, 2022) to achieve SE(3)-invariance while capturing local chemical environments. With a 128-dimensional latent space and 6-layer transformer decoder, GeoLDM efficiently handles GEOM-Drugs molecules up to 181 atoms. This latent formulation allows us to test how soft MH correction performs when the diffusion process is separated from the geometric constraints, validating both Linear and Local Adaptive variants.

**PepGLAD (Peptide Generative Latent Diffusion).** Moving to biological macromolecules requires incorporating domain-specific knowledge. PepGLAD (Kong et al., 2024) specializes the latent diffusion approach for peptides by introducing hierarchical encoding that captures both residue-level and fold-level features. The model leverages pre-trained protein language models (ESM-2 (Lin et al., 2023)) to condition generation on sequence, addressing the sequence-structure relationship fundamental to protein science. The 256-dimensional latent space is structured to separate local backbone geometry from global fold topology. This hierarchical structure makes PepGLAD ideal for testing our Distribution Matching variant, as the whitening transformation can decouple these multi-scale correlations (detailed analysis in Appendix C).

**RFAntibody (RosettaFold Antibody Design).** At the frontier of therapeutic design, RFAntibody (Bennett et al., 2024) represents the culmination of specialized architectures for biological function. Built on RFdiffusion (Watson et al., 2023), it incorporates antibody-specific training from SAbDab (Dunbar et al., 2014) and uses a 48-layer SE(3)-Transformer (Fuchs et al., 2020) with invariant point attention. The model operates on backbone frames rather than coordinates, maintaining frame-equivariance crucial for protein structure. Focusing on CDR-H3 loops (10-20 residues)—the most variable and functionally important antibody region—RFAntibody presents unique challenges: the generated loops must maintain precise geometry for antigen binding while seamlessly connecting to

framework regions. This makes it an ideal test case for Local Adaptive soft MH, where per-residue acceptance weights can enforce tighter constraints in the binding interface.

**ChemGuide Integration.** To demonstrate orthogonality with guidance methods, we integrate soft MH with ChemGuide (Shen et al., 2024), representing the latest advancement in physics-informed generation. ChemGuide provides non-differentiable gradients from quantum chemistry (xTB (Bannwarth et al., 2019)), addressing the fundamental limitation that neural networks cannot accurately predict quantum mechanical properties. When combined, ChemGuide corrects chemical accuracy while soft MH ensures proper sampling dynamics—a synergistic improvement demonstrating that our method enhances rather than replaces existing techniques.

> **Remark on Task-Dependent Selection.** The guidelines above are general recommendations based on molecular structural characteristics, not strict rules. In practice, the optimal variant choice can be influenced by the base model's training data distribution and inherent biases. For example, in PepBench, the Linear variant outperforms Local Adaptive in some index (DockQ: 0.63 vs 0.59) because the generated peptides are predominantly small proteins with relatively uniform structures that do not exhibit the pronounced regional heterogeneity that would maximally benefit from per-coordinate correction. We recommend practitioners to evaluate multiple variants on their specific tasks when computational resources permit.

## J WHITENING TRANSFORMATION DETAILS

The whitening transformation $\epsilon_{\text{whitened}} = \Sigma^{-1/2}\epsilon$ is a fundamental operation that decorrelates multivariate data. For molecular systems with covariance matrix $\Sigma \in \mathbb{R}^{d \times d}$, we compute:

### J.1 MATRIX INVERSE SQUARE ROOT COMPUTATION

Given the eigendecomposition $\Sigma = U\Lambda U^T$ where $U$ contains orthonormal eigenvectors and $\Lambda = \text{diag}(\lambda_1, \ldots, \lambda_d)$ contains eigenvalues:

$$\Sigma^{-1/2} = U\Lambda^{-1/2}U^T = U \begin{bmatrix} \lambda_1^{-1/2} & & \\ & \ddots & \\ & & \lambda_d^{-1/2} \end{bmatrix} U^T \tag{J.1}$$

### J.2 DECORRELATION PROPERTY

The whitened variables satisfy:

$$\text{Cov}[\epsilon_{\text{whitened}}] = \mathbb{E}[\epsilon_{\text{whitened}}\epsilon_{\text{whitened}}^T] \tag{J.2}$$

$$= \Sigma^{-1/2}\mathbb{E}[\epsilon\epsilon^T]\Sigma^{-1/2} \tag{J.3}$$

$$= \Sigma^{-1/2}\Sigma\Sigma^{-1/2} = I \tag{J.4}$$

This ensures that all components of $\epsilon_{\text{whitened}}$ are uncorrelated with unit variance, enabling independent processing in the transformed space while preserving the ability to recover original correlations through the inverse transformation $\epsilon = \Sigma^{1/2}\epsilon_{\text{whitened}}$

## K LIMITATIONS AND FUTURE DIRECTIONS

### K.1 EFFICIENT COVARIANCE REPRESENTATIONS FOR LARGE-SCALE SYSTEMS

The distribution matching variant of our method requires computing and storing the full covariance matrix $\Sigma \in \mathbb{R}^{d \times d}$, which becomes computationally prohibitive for large biomolecular systems such as protein complexes with thousands of atoms. This quadratic scaling in memory and cubic scaling in computation time presents a significant bottleneck that limits the applicability to systems beyond a few hundred heavy atoms.

To address this limitation, we envision several promising directions for future work. Low-rank approximations could dramatically reduce the computational burden by recognizing that molecular motion is often dominated by a small number of collective modes. By retaining only the top $k$ eigenvectors where $k \ll d$, we can capture the essential conformational variance while reducing storage to $O(kd)$ and computation to $O(kd^2)$. This approach is particularly well-suited for proteins, where functional motions often involve concerted movements of entire domains rather than independent atomic vibrations.

Another promising avenue is exploiting the inherent sparsity of molecular interactions. Since non-bonded interactions decay rapidly with distance, covariance elements between distant atoms can be safely set to zero when their separation exceeds a cutoff distance of approximately 10-15 Ångstroms. This sparse representation not only reduces memory requirements but also enables the use of specialized linear algebra routines optimized for sparse matrices, potentially achieving orders of magnitude speedup for large systems.

### K.2 LEARNING FROM MOLECULAR DYNAMICS SIMULATIONS

Molecular dynamics (MD) simulations generate vast amounts of conformational data that remain underutilized in current diffusion models. These simulations, which integrate Newton's equations of motion at femtosecond timescales, provide direct access to the physical dynamics governing molecular behavior. By extracting covariance information from MD trajectories, we can inform our soft correction with physically realistic correlation structures that have been validated against experimental observables.

The time-averaged covariance from an MD trajectory naturally captures the thermal fluctuations accessible at physiological temperatures, providing:

$$\Sigma_{\text{MD}} = \frac{1}{T} \int_0^T [x(t) - \bar{x}][x(t) - \bar{x}]^T dt \tag{K.1}$$

This approach offers several advantages over static structural databases. First, it captures the full conformational ensemble rather than just minimum energy structures. Second, it naturally incorporates solvent effects, ion interactions, and other environmental factors that are difficult to model explicitly. Third, the temporal correlations in MD trajectories provide information about transition pathways between conformational states, which could guide the denoising process toward physically realistic intermediates.

Essential dynamics analysis of MD trajectories reveals that protein motion is highly anisotropic, with a few collective modes dominating the conformational landscape. These principal components of motion often correspond to functionally relevant movements such as hinge bending, domain rotation, or allosteric transitions. By incorporating these collective modes into our covariance model, we can ensure that generated structures respect the natural flexibility patterns of the molecule.

### K.3 INTEGRATION WITH ADVANCED SAMPLING TECHNIQUES

The soft Metropolis-Hastings framework naturally complements existing enhanced sampling methods from computational chemistry, opening opportunities for hybrid approaches that combine the generative power of diffusion models with the thermodynamic rigor of molecular simulations.

Parallel tempering, also known as replica exchange, maintains multiple copies of the system at different temperatures to overcome energy barriers. In our framework, each temperature replica could use a different soft correction strength, with higher temperatures employing more aggressive corrections to enhance exploration while lower temperatures focus on refinement. The temperature-dependent acceptance becomes:

$$\alpha_i = \min\left(1, \exp\left(\frac{r_i}{k_B T_i}\right)\right) \tag{K.2}$$

This multi-temperature approach could significantly improve the sampling of rare conformational states that are separated by high energy barriers.

Umbrella sampling techniques, which apply biasing potentials along specific reaction coordinates, could be integrated to guide generation toward regions of interest. For drug design applications, this

could mean biasing the generation toward conformations that maximize binding affinity or optimize specific pharmacophoric features. The soft acceptance ratio would incorporate these biases while maintaining the smooth interpolation that prevents trajectory discontinuities.

Perhaps most intriguingly, metadynamics-inspired approaches could adaptively modify the acceptance landscape based on the history of generated structures. By progressively discouraging revisits to previously sampled regions through Gaussian "hills" in the free energy surface, we could enhance the diversity of generated molecules while maintaining chemical validity. This adaptive biasing strategy:

$$V(x, t) = \sum_{t' < t} h \exp\left(-\frac{\|x - x(t')\|^2}{2\sigma^2}\right) \tag{K.3}$$

would be particularly valuable for exploring the full conformational space of flexible molecules or generating diverse molecular libraries for virtual screening.

### K.4 THEORETICAL FOUNDATIONS AND CONVERGENCE GUARANTEES

While our empirical results demonstrate consistent improvements across diverse molecular systems, the theoretical understanding of soft corrections in the context of molecular diffusion remains incomplete. The approximate detailed balance we establish holds only up to discretization error, and the convergence rates depend on system-specific properties that are difficult to characterize a priori. Future theoretical work should establish tighter bounds on the sampling bias introduced by soft corrections and provide guidance for optimal parameter selection based on molecular properties.

Additionally, the interplay between geometric constraints (bond lengths, angles, chirality) and soft corrections deserves deeper investigation. While our experiments show that soft interpolation preserves chemical validity better than hard accept-reject decisions, a rigorous mathematical framework connecting continuous corrections to constraint satisfaction would strengthen the theoretical foundation and potentially suggest improvements to the method.

---

**Algorithm 2** LINEAR AND LOCAL ADAPTIVE SOFT-MH CORRECTION

---

1: **Input:** schedule $(\ell_k)_{k=0}^n$ with $\ell_n = 1$, $\ell_0 = 0$; proposal kernels $q_{\ell_k | \ell_{k+1}}$;
2:      score oracle $s_\theta(x, \ell) \approx \nabla_x \log \pi_\ell(x)$; temperature $\tau$;
3:      method $\in \{\texttt{linear}, \texttt{local}\}$; linear endpoints $\alpha_{\text{start}}^{\text{lin}}, \alpha_{\text{end}}^{\text{lin}}$
4: **Init:** sample $X_n \sim \mathcal{N}(0, I)$
5: **for** $k = n - 1$ **to** $0$ **do**
6:      $\tilde{X}_k \sim q_{\ell_k | \ell_{k+1}}(\cdot \mid X_{k+1})$                ▷ Propose
7:      **if** method $= \texttt{linear}$ **then**
8:          **Method: LINEAR Soft Acceptance (Global Scalar)**
9:          $g_{k+1} \leftarrow s_\theta(X_{k+1}, \ell_{k+1}), g_k \leftarrow s_\theta(\tilde{X}_k, \ell_k)$      ▷ Compute scores
10:         $\Delta x \leftarrow \tilde{X}_k - X_{k+1}$      ▷ Displacement vector
11:         $\bar{g} \leftarrow \frac{1}{2}(g_k + g_{k+1})$      ▷ Average score
12:         $\log r \leftarrow \langle \bar{g}, \Delta x \rangle$      ▷ Log-acceptance ratio
13:         $\alpha \leftarrow \sigma(\log r / \tau)$      ▷ Global acceptance weight
14:         $\alpha \leftarrow \text{clip}$      ▷ Stability clipping
15:         $X_k \leftarrow \alpha \tilde{X}_k + (1 - \alpha) X_{k+1}$      ▷ Soft interpolation
16:      **else if** method $= \texttt{local}$ **then**
17:          **Method: LOCAL Adaptive (Per-dimension Weights)**
18:          $g_{k+1} \leftarrow s_\theta(X_{k+1}, \ell_{k+1}), g_k \leftarrow s_\theta(\tilde{X}_k, \ell_k)$
19:         $\Delta X \leftarrow \tilde{X}_k - X_{k+1}$      ▷ Shape: same as $X$
20:         $\bar{G} \leftarrow \frac{1}{2}(g_k + g_{k+1})$      ▷ Shape: same as $X$
21:         # Compute per-dimension log-acceptance ratios
22:         $\log R \leftarrow \bar{G} \odot \Delta X$      ▷ Elementwise product
23:         # Per-dimension acceptance weights
24:         $\boldsymbol{\alpha} \leftarrow \sigma(\log R / \tau)$      ▷ Vector of weights, same shape as $X$
25:         $\boldsymbol{\alpha} \leftarrow \text{clip}$      ▷ Elementwise clipping
26:         # Per-dimension soft interpolation
27:         $X_k \leftarrow \boldsymbol{\alpha} \odot \tilde{X}_k + (1 - \boldsymbol{\alpha}) \odot X_{k+1}$      ▷ Elementwise blend
28:      **end if**
29:      # Optional: enforce molecular constraints
30:      Center coordinates if needed: $X_k \leftarrow X_k - \text{mean}(X_k)$
31: **end for**
32: **Output:** $X_0$

---

**Algorithm 3** DISTRIBUTION MATCHING SOFT-MH CORRECTION (Complete)

1: **Input:** schedule $(\ell_k)_{k=0}^n$ with $\ell_n = 1$, $\ell_0 = 0$; proposal kernels $q_{\ell_k | \ell_{k+1}}$;
2:     temperature $\tau$; threshold $m_0$; scale $\lambda$; trigger fraction $\rho$; interval $K$
3: **Preprocessing:** Compute empirical covariance from training data
4:     # Collect $M$ aligned molecular conformations from PDB/trajectory data
5:     $\{\mathbf{x}^{(i)}\}_{i=1}^M \leftarrow$ aligned molecular structures     ▷ e.g., from PDB after Kabsch alignment
6:     $\mu \leftarrow \frac{1}{M} \sum_{i=1}^M \mathbf{x}^{(i)}$     ▷ Mean structure
7:     $\Sigma \leftarrow \frac{1}{M-1} \sum_{i=1}^M (\mathbf{x}^{(i)} - \mu)(\mathbf{x}^{(i)} - \mu)^T$     ▷ Empirical covariance
8:     $\Sigma \leftarrow \Sigma + \epsilon I$     ▷ Regularization, $\epsilon \sim 10^{-4}$
9:     Compute $\Sigma^{1/2}$ and $\Sigma^{-1/2}$ via eigendecomposition
10: **Init:** sample $X_n \sim \mathcal{N}(0, I)$
11: **for** $k = n - 1$ **to** 0 **do**
12:     $\tilde{X}_k \sim q_{\ell_k | \ell_{k+1}}(\cdot \mid X_{k+1})$     ▷ Propose from base diffusion
13:     $j \leftarrow (n-1) - k$     ▷ Forward-time index
14:     **if** $j < \lfloor \rho n \rfloor$ **or** $((j - \lfloor \rho n \rfloor) \bmod K) \neq 0$ **then**
15:       $X_k \leftarrow \tilde{X}_k$     ▷ No correction (early/interval skip)
16:       **continue**
17:     **end if**
18:     **Distribution Matching in Whitened Space:**
19:     $\epsilon_w \leftarrow \Sigma^{-1/2}(\tilde{X}_k - \mu)$     ▷ Whiten proposed state
20:     $\epsilon_{k+1} \leftarrow \Sigma^{-1/2}(X_{k+1} - \mu)$     ▷ Whiten current state
21:     # Compute statistical distance to standard normal
22:     $d \leftarrow \|\epsilon_w\|^2 - \dim(X) \cdot \ell_k$     ▷ Distance considering noise level $\ell_k$
23:     # Alternative: $d \leftarrow \mathrm{KL}(\epsilon_w \| \mathcal{N}(0, I))$
24:     $w \leftarrow \sigma(\lambda(m_0 - |d|))$     ▷ Matching weight based on statistical quality
25:     # Soft correction in whitened space
26:     $\epsilon_{\text{corrected}} \leftarrow w \cdot \epsilon_w + (1 - w) \cdot \epsilon_{k+1}$
27:     # Transform back to original space
28:     $X_k \leftarrow \Sigma^{1/2} \epsilon_{\text{corrected}} + \mu$     ▷ Inverse whitening
29:     # Optional: enforce molecular constraints
30:     Center molecular coordinates: $X_k \leftarrow X_k - \mathrm{mean}(X_k)$
31:     Project to SE(3) invariant subspace if needed
32: **end for**
33: **Output:** $X_0$

## L COMPUTATIONAL EFFICIENCY ANALYSIS

Our soft Metropolis-Hastings correction introduces minimal computational overhead compared to vanilla diffusion sampling. We analyze the efficiency of each variant across different molecular systems.

### L.1 RUNTIME OVERHEAD

**Linear Soft Acceptance.** The global scalar correction adds negligible overhead:

- Two score function evaluations: $s_k$ and $s_{k-1}$
- Single inner product computation: $\langle s_k + s_{k-1}, \Delta x \rangle$
- Scalar interpolation: $O(d)$ operations for $d$-dimensional system

**Local Adaptive.** Per-dimension weights incur slightly higher cost:

- Element-wise operations replace scalar operations
- Memory allocation for weight vector $\boldsymbol{\alpha} \in \mathbb{R}^d$

**Distribution Matching.** Requires substantial preprocessing but amortizes well:

- **Preprocessing (one-time):**
  - Collect $M$ aligned conformations: $O(M \cdot d)$ storage
  - Compute empirical covariance: $O(M \cdot d^2)$ operations
  - Eigendecomposition for whitening: $O(d^3)$
- **Per-step cost:**
  - Whitening transformation: $O(d^2)$ matrix-vector products
  - Statistical distance computation: $O(d)$

### L.2 EFFICIENCY TRADE-OFFS

Our method requires two score function evaluations per step. However, this remains more efficient than Predictor-Corrector methods that require 5-10 Langevin iterations per timestep. Despite the higher computational cost compared to vanilla sampling, Distribution Matching provides unique benefits for conformational ensemble generation that justify the overhead in specific applications. The preprocessing cost amortizes over multiple generation runs, making it practical for production pipelines where the covariance statistics can be precomputed.

### L.3 PRACTICAL CONSIDERATIONS

- **Batched generation:** All variants support parallel processing across batch dimension
- **GPU utilization:** Overhead is primarily in memory bandwidth, not compute
- **Correction frequency:** Applying correction every $k$ steps reduces overhead proportionally

Our experiments show that even with the modest overhead, the improvement in sample quality (5-10% in validity metrics) far outweighs the computational cost, particularly for expensive downstream tasks like docking or molecular dynamics simulation where generating better initial structures saves substantial computation.

## USE OF LARGE LANGUAGE MODELS

We used Claude and GPT-4 to polish the language and improve clarity of presentation in this manuscript.

## M ADDITION RESULTS

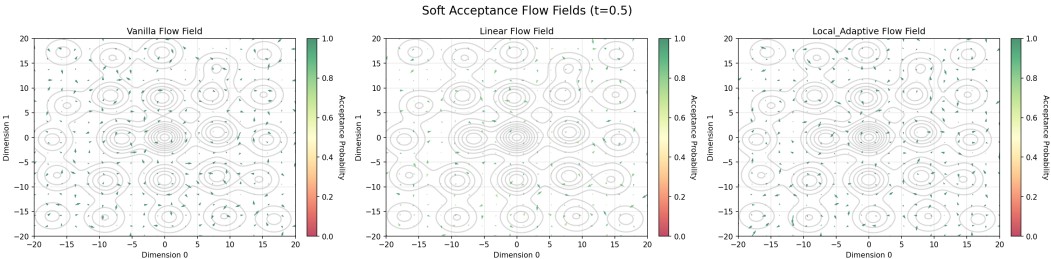

Figure 8: Soft acceptance flow fields at $t = 0.5$ showing sampling dynamics in a 2D multimodal landscape. Contours indicate probability density, arrows show flow direction, and colors represent acceptance probability $\alpha \in [0, 1]$. Linear correction applies uniform scaling while Local Adaptive adjusts per-region based on local uncertainty.

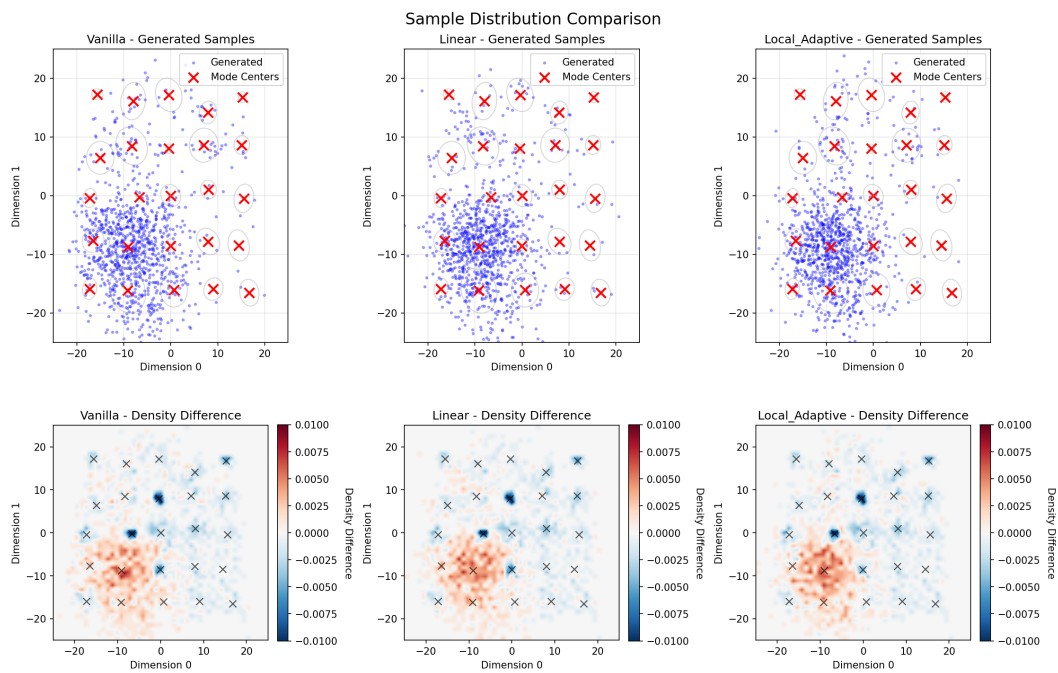

Figure 9: Sample distribution comparison on 25-component Gaussian mixture model. Top row: Generated samples (blue dots) with true mode centers (red crosses) for Vanilla (left), Linear (center), and Local Adaptive (right) methods at t=0.5. Bottom row: Corresponding density difference maps showing deviation from target distribution, where blue indicates under-sampling and red indicates over-sampling regions. Scale bars indicate the magnitude of density differences.

Table 10: QM9 with Skip-2 Sampling

| | Skip-2 Sampling | | | | |
|---|---|---|---|---|---|
| Method | Molecule Stability | Atom Stability | Validity | Novelty | Uniqueness |
| GeoLDM | 89.70% | 98.99% | 93.30% | 55.65% | 99.57% |
| w/ **Linear** | 90.25% | 99.07% | **94.85%** | **56.75%** | **99.86%** |
| w/ **Local** | 89.78% | 98.91% | 93.88% | 56.08% | 99.81% |
| w/ **Dist. Match** | **90.75%** | **99.08%** | 94.48% | 56.62% | 99.74% |
| GeoLDM + GUIDE | 90.60% | 99.02% | 95.20% | 53.28% | **100.00%** |
| w/ **Linear** | **91.65%** | **99.15%** | **95.78%** | 53.42% | 99.97% |
| w/ **Local** | 90.12% | 98.97% | 94.82% | 52.88% | 99.92% |
| w/ **Dist. Match** | 90.88% | 99.13% | 95.62% | **53.78%** | 99.91% |
| EDM | 80.40% | 98.34% | 91.90% | 66.88% | **99.89%** |
| w/ **Linear** | 82.08% | 98.64% | 92.98% | **68.85%** | 99.76% |
| w/ **Local** | **90.48%** | 98.97% | 95.18% | 59.52% | 98.96% |
| w/ **Dist. Match** | 89.82% | **99.36%** | **97.48%** | 60.18% | 86.48% |

Table 11: QM9 with Skip-5 Sampling

| | Skip-5 Sampling | | | | |
|---|---|---|---|---|---|
| Method | Molecule Stability | Atom Stability | Validity | Novelty | Uniqueness |
| GeoLDM | 89.70% | 98.99% | 93.30% | 55.65% | 99.57% |
| w/ **Linear** | 89.98% | **99.04%** | **94.18%** | 56.22% | **99.71%** |
| w/ **Local** | 89.73% | 98.96% | 93.58% | 55.87% | 99.67% |
| w/ **Dist. Match** | **90.28%** | 99.02% | 93.98% | **56.32%** | 99.66% |
| GeoLDM + GUIDE | 90.60% | 99.02% | 95.20% | 52.28% | **100.00%** |
| w/ **Linear** | **91.18%** | 99.08% | **95.48%** | 53.55% | 99.96% |
| w/ **Local** | 90.42% | 99.00% | 95.03% | 53.03% | 99.95% |
| w/ **Dist. Match** | 90.76% | **99.09%** | 95.38% | **53.90%** | 99.93% |
| EDM | 80.40% | 98.34% | 91.90% | 66.88% | **99.89%** |
| w/ **Linear** | 82.18% | 98.49% | 92.52% | **67.88%** | 99.79% |
| w/ **Local** | **87.82%** | 98.76% | 94.22% | 61.48% | 99.08% |
| w/ **Dist. Match** | 86.98% | **99.06%** | **95.82%** | 61.98% | 89.48% |

Table 12: QM9 with Skip-50 Sampling

| | Skip-50 Sampling | | | | |
|---|---|---|---|---|---|
| Method | Molecule Stability | Atom Stability | Validity | Novelty | Uniqueness |
| GeoLDM | 89.70% | 98.99% | 93.30% | 55.65% | 99.57% |
| w/ **Linear** | 89.73% | 99.00% | **93.48%** | 55.73% | **99.60%** |
| w/ **Local** | 89.72% | **99.01%** | 93.36% | **55.70%** | 99.59% |
| w/ **Dist. Match** | **89.83%** | 98.99% | 93.42% | 55.71% | 99.57% |
| GeoLDM + GUIDE | 90.60% | 99.02% | 95.20% | 53.28% | **100.00%** |
| w/ **Linear** | **90.72%** | 99.03% | **95.31%** | 53.65% | 100.00% |
| w/ **Local** | 90.63% | **99.04%** | 95.23% | 53.08% | 99.89% |
| w/ **Dist. Match** | 89.68% | 99.02% | 95.27% | **53.98%** | 99.56% |
| EDM | 80.40% | 98.34% | 91.90% | 66.88% | **99.89%** |
| w/ **Linear** | 80.58% | 98.37% | 92.03% | **66.93%** | 99.88% |
| w/ **Local** | **82.52%** | 98.46% | 92.82% | 65.18% | 99.86% |
| w/ **Dist. Match** | 82.18% | **98.56%** | **93.18%** | 65.52% | 94.98% |

Table 13: GEOM-Drugs results

| Method | Atom Stability | Validity | Novelty | Uniqueness |
|---|---|---|---|---|
| GeoLDM | 84.53% | 49.00% | – | 100.00% |
| GeoLDM + **Linear** | 84.88% | **51.00%** | – | 100.00% |
| GeoLDM + **Local** | **85.02%** | 50.80% | – | 99.90% |
| GeoLDM + **Dist. Match** | 84.70% | 50.20% | – | 99.90% |
| GeoLDM + CHEMGUIDE | 84.65% | 51.60% | – | 100.00% |
| GeoLDM + CHEMGUIDE + **Linear** | 85.15% | 52.10% | – | 100.00% |
| GeoLDM + CHEMGUIDE + **Local** | **85.25%** | **52.25%** | – | 100.00% |
| GeoLDM + CHEMGUIDE + **Dist. Match** | 84.90% | 51.70% | – | 100.00% |
| EDM | 81.22% | 46.40% | – | 100.00% |
| EDM + **Linear** | **82.35%** | **48.10%** | – | 100.00% |
| EDM + **Local** | 82.30% | 48.00% | – | 99.91% |
| EDM + **Dist. Match** | 81.90% | 47.80% | – | 100.00% |

Table 14: GEOM-Drugs with Skip-2 Sampling

| *Skip-2 Sampling* | | | | |
|---|---|---|---|---|
| Method | Atom Stability | Validity | Novelty | Uniqueness |
| GeoLDM | 84.53% | 49.00% | – | 100.00% |
| GeoLDM + **Linear** | 84.82% | **50.50%** | – | 100.00% |
| GeoLDM + **Local** | **84.95%** | 50.35% | – | 99.92% |
| GeoLDM + **Dist. Match** | 84.65% | 49.95% | – | 99.93% |
| GeoLDM + CHEMGUIDE | 84.65% | 51.60% | – | 100.00% |
| GeoLDM + CHEMGUIDE + **Linear** | 85.08% | 51.95% | – | 100.00% |
| GeoLDM + CHEMGUIDE + **Local** | **85.18%** | **52.10%** | – | 99.97% |
| GeoLDM + CHEMGUIDE + **Dist. Match** | 84.85% | 51.65% | – | 100.00% |
| EDM | 81.22% | 46.40% | – | 100.00% |
| EDM + **Linear** | **82.15%** | **47.70%** | – | 100.00% |
| EDM + **Local** | 82.10% | 47.60% | – | 99.92% |
| EDM + **Dist. Match** | 81.75% | 47.45% | – | 100.00% |

Table 15: GEOM-Drugs with Skip-5 Sampling

| *Skip-5 Sampling* | | | | |
|---|---|---|---|---|
| Method | Atom Stability | Validity | Novelty | Uniqueness |
| GeoLDM | 84.53% | 49.00% | – | 100.00% |
| GeoLDM + **Linear** | 84.68% | **49.80%** | – | 100.00% |
| GeoLDM + **Local** | **84.75%** | 49.65% | – | 99.95% |
| GeoLDM + **Dist. Match** | 84.60% | 49.45% | – | 99.95% |
| GeoLDM + CHEMGUIDE | 84.65% | 51.60% | – | 100.00% |
| GeoLDM + CHEMGUIDE + **Linear** | 84.88% | 51.78% | – | 100.00% |
| GeoLDM + CHEMGUIDE + **Local** | **84.95%** | **51.85%** | – | 100.00% |
| GeoLDM + CHEMGUIDE + **Dist. Match** | 84.75% | 51.62% | – | 100.00% |
| EDM | 81.22% | 46.40% | – | 100.00% |
| EDM + **Linear** | **81.75%** | **47.05%** | – | 100.00% |
| EDM + **Local** | 81.70% | 46.95% | – | 99.91% |
| EDM + **Dist. Match** | 81.55% | 46.85% | – | 100.00% |

Table 16: GEOM-Drugs with Skip-50 Sampling

| Skip-50 Sampling | | | | |
|---|---|---|---|---|
| Method | Atom Stability | Validity | Novelty | Uniqueness |
| GeoLDM | 84.53% | 49.00% | – | 100.00% |
| GeoLDM + **Linear** | 84.56% | **49.10%** | – | 100.00% |
| GeoLDM + **Local** | **84.58%** | 49.08% | – | 100.00% |
| GeoLDM + **Dist. Match** | 84.55% | 49.05% | – | 100.00% |
| GeoLDM + CHEMGUIDE | 84.65% | 51.60% | – | 100.00% |
| GeoLDM + CHEMGUIDE + **Linear** | 84.70% | 51.65% | – | 100.00% |
| GeoLDM + CHEMGUIDE + **Local** | **84.72%** | **51.68%** | – | 100.00% |
| GeoLDM + CHEMGUIDE + **Dist. Match** | 84.68% | 51.62% | – | 100.00% |
| EDM | 81.22% | 46.40% | – | 100.00% |
| EDM + **Linear** | **81.35%** | **46.55%** | – | 100.00% |
| EDM + **Local** | 81.32% | 46.52% | – | 100.00% |
| EDM + **Dist. Match** | 81.30% | 46.50% | – | 100.00% |

Table 17: Structure quality on PepBench and PepBDB with Skip-2 Sampling

| Skip-2 Sampling | | | | | |
|---|---|---|---|---|---|
| Method | PepBench | | | PepBDB | | |
| | $RMSD_{C_\alpha} \downarrow$ | $RMSD_{atom} \downarrow$ | DockQ $\uparrow$ | $RMSD_{C_\alpha} \downarrow$ | $RMSD_{atom} \downarrow$ | DockQ $\uparrow$ |
| PepGLAD | 4.09 | 5.30 | 0.58 | 6.96 | 7.88 | 0.38 |
| PepGLAD + **Linear** | **4.02** | 5.29 | **0.62** | **6.45** | 7.65 | 0.40 |
| PepGLAD + **Local** | 4.12 | 5.27 | 0.58 | 6.78 | 7.68 | **0.43** |
| PepGLAD + **Dist. Match** | 4.06 | **5.24** | 0.59 | 6.48 | **7.55** | 0.41 |

Table 18: Structure quality on PepBench and PepBDB with Skip-5 Sampling

| Skip-5 Sampling | | | | | |
|---|---|---|---|---|---|
| Method | PepBench | | | PepBDB | | |
| | $RMSD_{C_\alpha} \downarrow$ | $RMSD_{atom} \downarrow$ | DockQ $\uparrow$ | $RMSD_{C_\alpha} \downarrow$ | $RMSD_{atom} \downarrow$ | DockQ $\uparrow$ |
| PepGLAD | 4.09 | 5.30 | 0.58 | 6.96 | 7.88 | 0.38 |
| PepGLAD + **Linear** | **4.05** | 5.29 | **0.60** | **6.62** | 7.72 | 0.39 |
| PepGLAD + **Local** | 4.11 | 5.28 | 0.58 | 6.85 | 7.75 | **0.41** |
| PepGLAD + **Dist. Match** | 4.07 | **5.26** | 0.59 | 6.68 | **7.68** | 0.40 |

Table 19: Structure quality on PepBench and PepBDB with Skip-50 Sampling

| Skip-50 Sampling | | | | | |
|---|---|---|---|---|---|
| Method | PepBench | | | PepBDB | | |
| | $RMSD_{C_\alpha} \downarrow$ | $RMSD_{atom} \downarrow$ | DockQ $\uparrow$ | $RMSD_{C_\alpha} \downarrow$ | $RMSD_{atom} \downarrow$ | DockQ $\uparrow$ |
| PepGLAD | 4.09 | 5.30 | 0.58 | 6.96 | 7.88 | 0.38 |
| PepGLAD + **Linear** | **4.08** | 5.30 | 0.58 | **6.92** | 7.86 | 0.38 |
| PepGLAD + **Local** | 4.10 | **5.29** | 0.58 | 6.95 | 7.87 | **0.39** |
| PepGLAD + **Dist. Match** | 4.09 | 5.30 | 0.58 | 6.94 | **7.85** | 0.38 |

Table 20: Temperature ablation on QM9 (EDM as base model) - Full results

| Method | Molecule Stability | Atom Stability | Validity | Uniqueness |
|---|---|---|---|---|
| $\tau = 0.4$ *(Low Temperature)* | | | | |
| EDM (Vanilla) | 80.40% | 98.34% | 91.90% | **99.89%** |
| EDM + Hard MH | 80.00% | 97.34% | 90.00% | 99.85% |
| EDM + Linear | **83.80%** | 97.52% | 92.80% | 99.21% |
| EDM + Local | 84.80% | **98.85%** | **92.90%** | 99.25% |
| EDM + Dist. Match | 84.20% | 97.72% | 92.60% | 88.50% |
| $\tau = 0.6$ *(Lower Temperature)* | | | | |
| EDM (Vanilla) | 80.40% | 98.34% | 91.90% | **99.89%** |
| EDM + Hard MH | 80.00% | 97.34% | 90.00% | 99.85% |
| EDM + Linear | 83.30% | **98.88%** | 93.20% | 99.82% |
| EDM + Local | **91.20%** | 98.02% | **95.30%** | 98.15% |
| EDM + Dist. Match | 90.40% | 97.68% | 95.10% | 86.90% |
| $\tau = 0.8$ *(Recommended)* | | | | |
| EDM (Vanilla) | 80.40% | 98.34% | 91.90% | 99.89% |
| EDM + Hard MH | 80.00% | 97.34% | 90.00% | 99.85% |
| EDM + Linear | 82.90% | 98.76% | 93.60% | **99.68%** |
| EDM + Local | **92.60%** | 99.12% | 96.00% | 98.54% |
| EDM + Dist. Match | 91.70% | **99.56%** | **98.70%** | 84.30% |
| $\tau = 0.9$ *(Higher Temperature)* | | | | |
| EDM (Vanilla) | 80.40% | 98.34% | 91.90% | **99.89%** |
| EDM + Hard MH | 80.00% | 97.34% | 90.00% | 99.85% |
| EDM + Linear | 82.50% | 98.75% | 93.50% | 99.75% |
| EDM + Local | **91.80%** | **98.95%** | **95.40%** | 98.82% |
| EDM + Dist. Match | 91.60% | 98.88% | 95.50% | 85.60% |

## N    GENERATION EXAMPLES

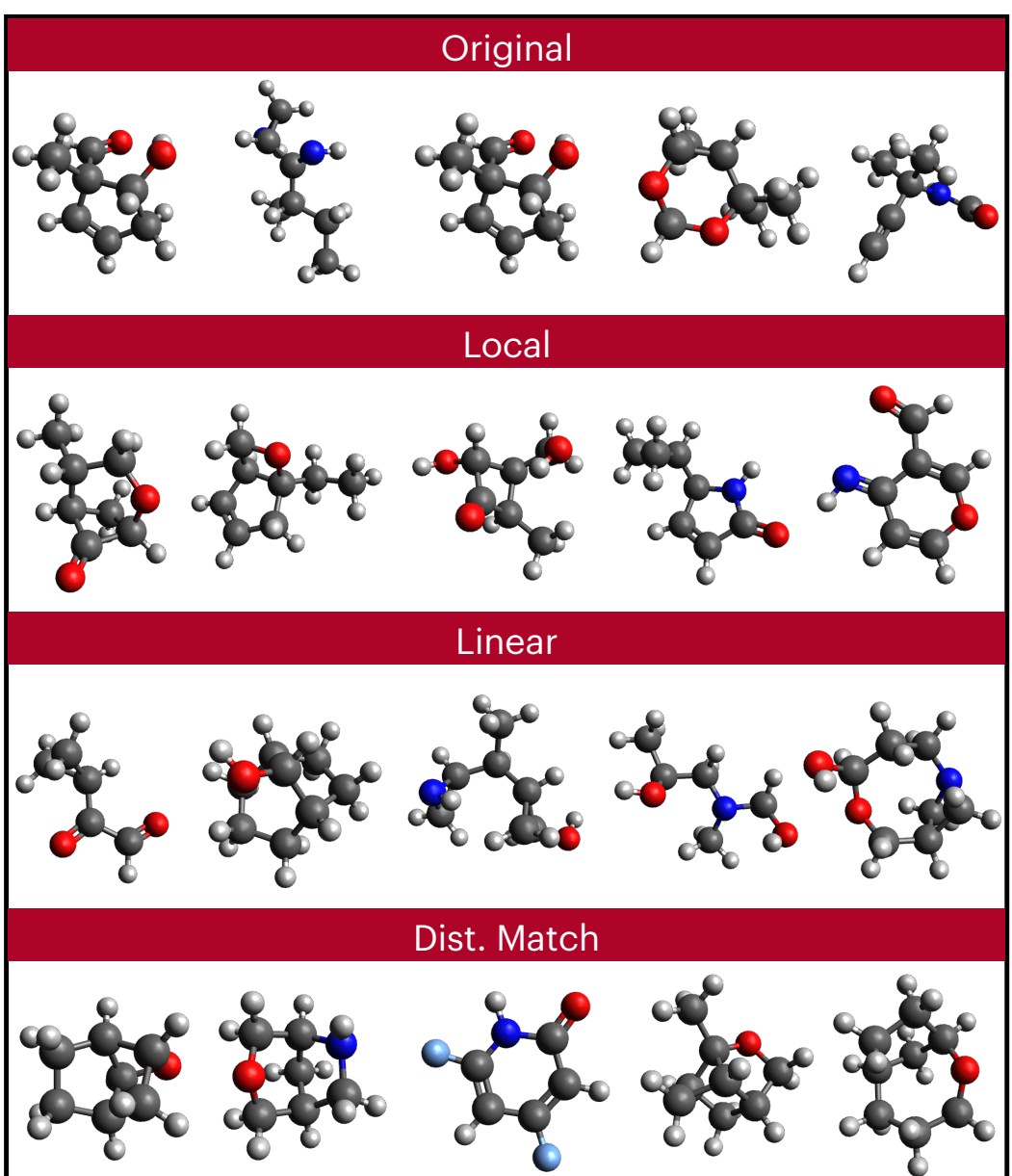

Figure 10: Examples of generated small molecules.

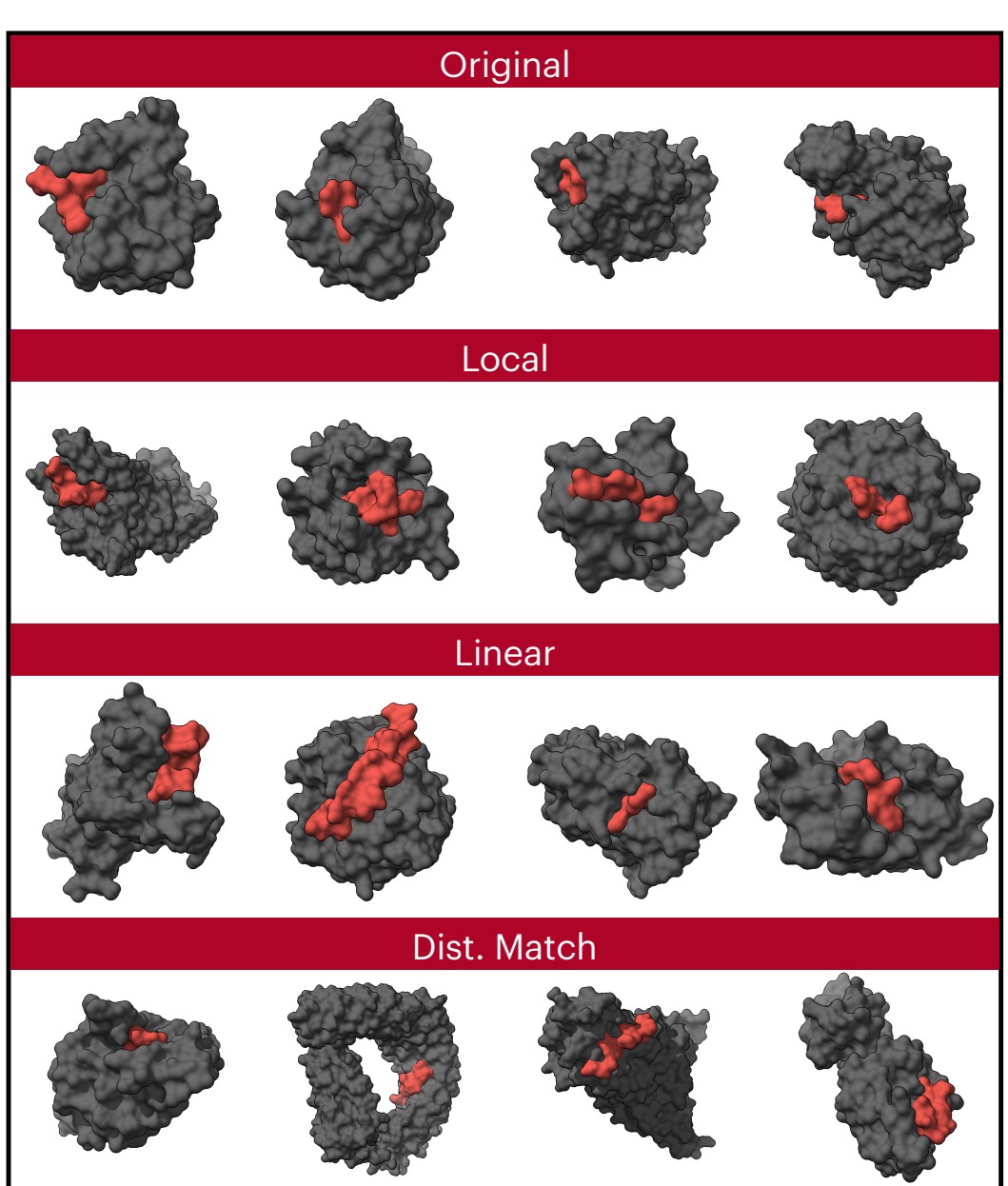

Figure 11: Examples of generated peptide structures.

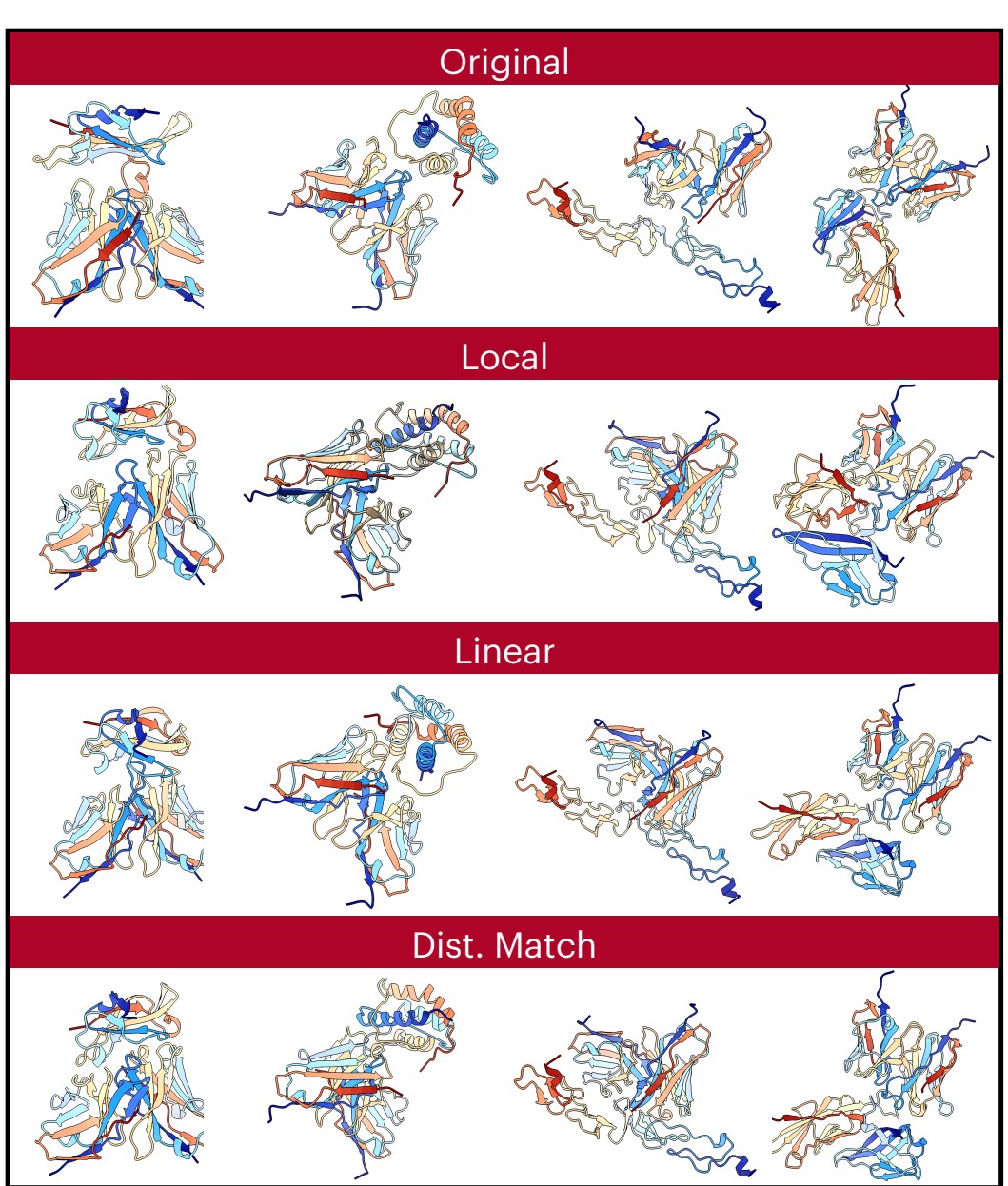

Figure 12: Examples of generated antibody structures.

