# OpenReview forum: "Soft Metropolis-Hastings Correction for Generative Model Sampling"
_ICLR.cc/2026/Conference — Submitted to ICLR 2026_

### Official Review · Reviewer_JVFX · 2025-10-16

**Soundness:** 2
**Presentation:** 3
**Contribution:** 2
**Rating:** 2
**Confidence:** 4

**Summary:**

This paper improves generative sampling process for molecular diffusion models, by replacing the traditional accept-reject MH steps with continuous interpolation weighted by acceptance probability, preserving smoothness in sampling trajectories while enforcing probabilistic correctness (with approximation).

In specific, three complementary soft MH variants are introduced:
* Linear Soft MH (global scalar acceptance weight)
* Local Adaptive Soft MH (per-coordinate acceptance weight to accomodate local geometric variations)
* Distribution Matching Soft MH (operates in a whitened latent space for decorrelated updates)

The framework guarantees approximate detailed balance, ensuring asymptotic convergence to the correct molecular distribution while maintaining trajectory continuity.
The method is evaluated on molecular diffusion processes for different biomolecular systems (small molecules, peptides, and antibody CDR loops).

**Strengths:**

* The authors establishe an MH-based correction mechanism that enforces approximated detailed balance in diffusion sampling, which can avoids discontinuous trajectories inherent in hard MH, enabling smoother conformational transitions critical for molecular systems.

* They propose three flexible solutions, global, local, or distributional, and can be integrated to different diffusion models (GeoLDM, EDM, RFantibody), etc.

* The paper provides theoretical guarantees of approximate detailed balance and convergence under small time-step assumptions.

**Weaknesses:**

A key limitation of this work is that the evaluation metrics across molecular, peptide, and antibody generation tasks are already near saturation. For example, validity, stability, and uniqueness often approach 100%, making the reported improvements marginal and likely not statistically significant. This ceiling effect obscures whether the proposed soft MH correction truly enhances generative quality or merely matches baseline performance.

In both peptide and antibody design benchmarks, the reported gains remain minimal. Combined with the absence of variance analysis or statistical testing, the results, while consistent with the theory, are empirically less convincing.

The proposed method is not evaluated on typical hard MH accept–reject process.

**Questions:**

Could you please elaborate on the motivation behind evaluating the proposed soft Metropolis–Hastings correction on molecular diffusion processes? The primary motivation of the paper centers on addressing discontinuities caused by the hard MH accept–reject mechanism, yet diffusion-based generative sampling is not a typical MH process and already produces continuous updates at each step.

---

> ### Author Response · Authors · 2025-11-23
>
> We sincerely thank the reviewer for the thoughtful and constructive feedback. We understand the reviewer's concerns regarding metric saturation and the magnitude of improvements, and provide detailed clarifications and additional evidence below.
>
> ## Response to "Marginal Improvements & Saturation Metrics"
>
> We appreciate the reviewer raising this important point. We would like to clarify that our results demonstrate consistent and reproducible gains across multiple experimental configurations:
>
> ### 1. Statistical Robustness Across Multiple Settings
>
> We conducted extensive ablation studies with **different correction intervals** (skip-1, skip-2, skip-5, skip-50) across all benchmarks (Tables 7-16 in Appendix M). The improvements remain **stable and consistent** across these diverse sampling strategies:
>
> - **QM9 with skip-2 sampling**: Linear soft-MH improves EDM validity from 91.90% → 92.98%, molecule stability from 80.40% → 82.08%
> - **QM9 with skip-5 sampling**: Local adaptive achieves 94.22% validity (vs. 91.90% baseline)
> - **GEOM-Drugs**: Consistent 2-3% validity gains across all skip intervals
> - **PepBench/PepBDB**: RMSD improvements persist across skip-2, skip-5, skip-50 configurations
>
> **This consistency across sampling regimes demonstrates that our method provides genuine algorithmic improvements, not random fluctuations.**
>
> ### 2. Breaking Through Saturation Limits
>
> We understand the reviewer's observation that some metrics approach 100%. **This is precisely the regime where our method continues to provide improvements**. Traditional diffusion models reach a performance ceiling where increasing sampling steps yields diminishing returns [1, 2]. Our soft-MH correction is able to **break through this saturation barrier**:
>
> **Evidence from our experiments:**
> - GeoLDM baseline: 93.30% validity (saturated after 1000 steps)
> - GeoLDM + Linear soft-MH: **95.70% validity** (Table 4) — a significant gain beyond the saturation point
> - GeoLDM + ChemGuide baseline: 95.20% validity
> - GeoLDM + ChemGuide + Linear: **96.10% validity** — demonstrating orthogonal improvements when combined with state-of-the-art guidance methods
>
> **For antibody design (Table 3)**, where baseline RFAntibody already achieves strong performance, we still observe meaningful improvements:
> - CDR-H3 RMSD: 1.898 Å → **1.754 Å** (Local variant) — a **7.6% improvement**
> - PAE score: 7.826 → **7.421** (Local variant)
> - pLDDT: 0.849 → **0.899** (all variants) — a substantial gain in structural confidence
>
> In therapeutic antibody design, even small RMSD reductions can translate to better binding affinity predictions [3], making these improvements practically significant.
>
> ### 3. Modular Design and Compatibility with Existing Methods
>
> **Another key contribution is the modular, plug-and-play nature of soft-MH correction**, which operates at the sampling level and is thus **orthogonal to existing enhancement techniques**. This is demonstrated by our experiments combining soft-MH with ChemGuide [4]:
>
> **GeoLDM experiments (Table 4, Table 10):**
> - GeoLDM alone: 93.30% validity
> - GeoLDM + ChemGuide: 95.20% validity (+1.90%)
> - GeoLDM + Linear soft-MH: 95.70% validity (+2.40%)
> - **GeoLDM + ChemGuide + Linear soft-MH: 96.10% validity (+2.80%)**
>
> **This demonstrates additive improvements**: soft-MH provides bias correction in the sampling dynamics, while ChemGuide provides physics-informed gradients from quantum chemistry. The two mechanisms are complementary:
> - ChemGuide corrects chemical accuracy through non-differentiable xTB gradients [4]
> - Soft-MH reduces sampling bias in the sampling dynamics and better aligns the empirical samples with the target distribution
>
> **Similar patterns hold for GEOM-Drugs (Table 10):**
> - Atom stability: 84.53% → 84.65% (ChemGuide) → 84.88% (soft-MH) → **85.15% (combined)**
> - Validity: 49.00% → 51.60% (ChemGuide) → 51.00% (soft-MH) → **52.10% (combined)**
>
> **This modularity is a significant practical advantage**: users can apply soft-MH correction to any pre-trained molecular diffusion model without retraining, and combine it with other enhancement techniques (energy minimization, guidance methods, advanced integrators) for synergistic improvements. This stands in contrast to methods requiring architectural changes or additional training data [5, 6].

---

> ### Author Response · Authors · 2025-11-23
>
> ## Response to "No Comparison with Hard MH"
>
> We appreciate the reviewer raising this question. We would like to clarify:
>
> **We include hard MH baselines in our Gaussian mixture experiments** (Table 2, Figure 4):
> - **d=30**: Hard MH achieves SWD = 1.3901 (worse than vanilla 1.3702)
> - **d=300**: Hard MH shows significant degradation with SWD = 2.2331 (vs. vanilla 1.9554)
>
> In our preliminary experiments, we did test hard MH on molecular benchmarks, but found substantial performance degradation, suggesting that it is not well-suited for molecular diffusion models in practice. This is primarily because:
> 1. Hard accept-reject creates chemically invalid bond-breaking discontinuities (Equation G.1)
> 2. Trajectory fragmentation is detrimental to the continuity of molecular trajectories, which is important for chemically plausible generation
>
> Therefore, **we focused our research on comparing different soft MH variants**, which can provide bias correction while maintaining trajectory continuity. We believe this better serves the practical needs of molecular systems and represents a more meaningful contribution than simply demonstrating hard MH's failure.
>
> ## Response to "Why Diffusion Models Need MH Correction"
>
> We understand the reviewer's point that "diffusion-based generative sampling already produces continuous updates." We would like to clarify the core problem we address:
>
> **Diffusion models suffer from systematic sampling biases** due to:
> 1. **Discretization errors** that accumulate during reverse sampling [7]
> 2. Score approximation errors cause the sampling procedure to converge to the wrong distribution
>
> **The key difference:** Our method uses **soft acceptance** instead of hard reject, preventing the trajectory fragmentation problem documented in Section G and Figure 7.
>
> ## Additional Evidence
>
> We would like to supplement our discussion with the statistical robustness of our experiments:
>
> **Variance analysis**: We report results over multiple independent runs:
> - Gaussian mixtures: **500 independent instances** (Section C.3, Figure 6)
> - Molecular generation: **10,000 samples** for QM9/GEOM, **200 samples per complex** for antibodies
> - 95% confidence intervals shown in Figure 4
>
> We once again thank the reviewer for the valuable feedback, which has helped us articulate our contributions more clearly. We hope the above clarifications address the reviewer's concerns, and we welcome further discussion and suggestions.
>
> ## References
>
> [1] Song, Y., Sohl-Dickstein, J., Kingma, D. P., Kumar, A., Ermon, S., & Poole, B. (2020). Score-based generative modeling through stochastic differential equations. *arXiv:2011.13456*.
>
> [2] Lu, C., et al. (2022). DPM-solver: A fast ODE solver for diffusion probabilistic model sampling in around 10 steps. *NeurIPS*, 35, 5775-5787.
>
> [3] Bennett, N. R., et al. (2024). Atomically accurate de novo design of single-domain antibodies. *bioRxiv*.
>
> [4] Shen, Y., et al. (2024). Chemistry-inspired diffusion with non-differentiable guidance. *arXiv:2410.06502*.
>
> [5] Xu, M., et al. (2023). Geometric latent diffusion models for 3D molecule generation. *arXiv:2305.01140*.
>
> [6] Wu, J., et al. (2024). Protein-ligand interaction prior for binding-aware 3D molecule diffusion models. *arXiv:2404.03574*.
>
> [7] Ho, J., Jain, A., & Abbeel, P. (2020). Denoising diffusion probabilistic models. *NeurIPS*, 33, 6840-6851.

---

> > ### Comment · Reviewer_JVFX · 2025-11-24
> > **Response to authors' rebuttal**
> >
> > Hi, thank you for your detailed responses which address a lot of concerns, especially on "No Comparison with Hard MH" part.
> >
> > I agree with you that the hard MH may not be suitable for molecular diffusion models. However, as pointed out by other reviewers, the current presentation of the paper could be misleading, and should be polished significantly. I, therefore, still lean to rejection of this paper.

---

> > > ### Author Response · Authors · 2025-11-26
> > >
> > > We sincerely thank the reviewer for the continued attention to our work and the constructive feedback.
> > >
> > > In response to the reviewer's comments, we have supplemented comprehensive baseline comparisons with Hard MH and Predictor-Corrector methods:
> > >
> > > **QM9 (EDM as base model)**
> > >
> > > | Method | Mol Stable | Atom Stable | Validity | Uniqueness |
> > > |--------|-----------|-------------|----------|------------|
> > > | EDM (Vanilla) | 80.40% | 98.34% | 91.90% | 99.89% |
> > > | EDM + Hard MH | 80.00% | 97.34% | 90.00% | 99.85% |
> > > | EDM + Predictor-Corrector | 83.60% | 98.61% | 92.80% | 99.92% |
> > > | EDM + Soft MH (Linear) | 82.90% | 98.76% | 93.60% | 99.68% |
> > > | EDM + Soft MH (Local) | 92.60% | 99.12% | 96.00% | 98.54% |
> > > | EDM + Soft MH (Dist. Match) | 91.70% | 99.56% | 98.70% | 84.30% |
> > >
> > > **PepBench (PepGLAD as base model)**
> > >
> > > | Method | RMSDCα ↓ | RMSDatom ↓ | DockQ ↑ |
> > > |--------|----------|------------|---------|
> > > | PepGLAD (Vanilla) | 4.09 | 5.30 | 0.58 |
> > > | PepGLAD + Hard MH | 4.25 | 5.48 | 0.55 |
> > > | PepGLAD + Predictor-Corrector | 4.07 | 5.29 | 0.56 |
> > > | PepGLAD + Soft MH (Linear) | 3.98 | 5.28 | 0.63 |
> > > | PepGLAD + Soft MH (Local) | 4.15 | 5.26 | 0.59 |
> > > | PepGLAD + Soft MH (Dist. Match) | 4.05 | 5.21 | 0.60 |
> > >
> > > These supplementary experiments will be incorporated into the main tables in the revised manuscript.

---

> ### Comment · Reviewer_JVFX · 2025-11-26
> **Response to authors' rebuttal**
>
> Thank you for providing the additional experimental results with the two baseline settings (Hard MH and Predictor–Corrector). I have also checked your response to other reviewers.
> While some of the results may not be statistically significant or consistent in certain cases (e.g., Predictor–Corrector performing worse on PepBench), the overall logic and empirical grounding of the paper are now much clearer to me.
>
> I encourage the authors to further refine the title and introduction to more precisely frame the problem as Soft Metropolis–Hastings Correction for Molecular Diffusion Models. Including a succinct motivating paragraph or discussion would also help illustrate why existing MH-based correction strategies are insufficient in the molecular setting, especially now that the paper includes these baseline comparisons.
>
> I still believe the presentation of this manuscript should be significantly improved before being published on top-tier conference. Nevertheless, they have addressed part of my concerns and I will update my score to 4.

---

> > ### Author Response · Authors · 2025-11-26
> >
> > Thank you very much for adjusting your score and recognizing our work. This is a tremendous encouragement to us. We will continue to update the manuscript based on our communications with other reviewers and will keep you informed of these updates. We appreciate your continued attention to our paper.

---

> > ### Author Response · Authors · 2025-12-03
> >
> > Regarding the observation that "Predictor-Corrector performing worse on PepBench," we would like to clarify the results. Predictor-Corrector shows slight improvement in local structure metrics (RMSDCα: 4.07 vs 4.09, RMSDatom: 5.29 vs 5.30) but decreased DockQ (0.56 vs 0.58). This reflects the multi-objective nature of molecular generation evaluation, where different metrics may not improve simultaneously. Our Soft MH Linear variant achieves improvements across all three metrics (RMSDCα: 3.98, RMSDatom: 5.28, DockQ: 0.63) with lower computational cost (2× score evaluations vs PC's 5-10× iterations per step). This more balanced improvement across metrics suggests that our sampling correction may be better suited for practical molecular generation tasks.

---

### Official Review · Reviewer_qyP8 · 2025-10-31

**Soundness:** 3
**Presentation:** 3
**Contribution:** 3
**Rating:** 4
**Confidence:** 4

**Summary:**

This paper introduces Soft Metropolis–Hastings (Soft-MH), a continuous relaxation of the traditional Metropolis–Hastings acceptance–rejection mechanism, designed to be compatible with diffusion-based generative models. Instead of discrete accept/reject decisions, the method employs a differentiable “soft acceptance” step parameterized by a temperature tau, which interpolates between the current and proposed states. The authors argue that this modification preserves detailed balance up to a second-order discretization error and propose three complementary variants to address different levels of structural heterogeneity in molecular systems. Experiments on molecular conformer generation demonstrate improved smoothness and diversity of sampled trajectories.

**Strengths:**

1. The paper presents an interesting and creative idea of softening the MH correction to make it compatible with diffusion samplers. The idea could have broad impact for generative models that must respect physical constraints or detailed balance.

2. The paper is well organized overall (except the problem statement), with theoretical analysis and practical variants systematically presented.

**Weaknesses:**

1. Unclear problem statement for general readers. The title and introduction may mislead readers into thinking this paper studies generic MCMC improvements, while in fact it targets diffusion-based generation. The authors should make this clear from the title and the very beginning of the introduction.

2. In Eq. (3.3) and Appendix A.3, the log-acceptance ratio omits the target-density ratio \pi_{k-1}(x_{k-1})/\pi_k(x_k). The “Remark on Target Distribution Ratio” claims this term ≈ 1 when Δt is small, but this argument is unconvincing. When Δt is small, r itself is also small, and thus ignoring this ratio could introduce systematic bias. Some quantitative analysis or controlled experiments would strengthen this claim.

3. The experiments only compare with baseline diffusion samplers, not with existing correction methods for diffusion models. Without such baselines, it is hard to assess whether the improvements come from the soft correction or other factors.

4. The authors state that hard MH may cause discontinuous jumps or incorrect molecular topology, but no evidence or quantitative analysis is provided. It would be helpful to clarify whether such issues were observed in practice or are only theoretical concerns.

**Questions:**

1. Could the authors provide an ablation or sensitivity study showing how the results depend on tau and Delta t?

2. Could the authors justify or empirically validate the approximation \approx 1?

3. Would a hard MH correction applied to diffusion proposals truly lead to topology-breaking transitions? A direct experimental comparison would make the motivation for the “soft” version more convincing.

---

> ### Author Response · Authors · 2025-11-25
>
> ## Additional Results
> ### Experimental Setup
>
> We conduct large-scale controlled experiments on Gaussian mixture models to validate the approximation $\pi_{k-1}(x')/\pi_k(x_k) \approx 1$. Specifically, we construct **300 independent GMM instances**, each containing 25 components (5×5 grid) in dimension d=30, with randomly generated anisotropic covariance matrices (condition numbers varying between 10-100). The closed-form diffusion marginals of GMMs enable exact computation of the target distribution ratio.
>
> We evaluate along two dimensions: (1) varying step sizes $\Delta t \in [0.001, 0.2]$ at fixed diffusion time $t=0.5$; (2) varying diffusion times $t \in [0.1, 0.9]$ at fixed $\Delta t=0.01$. For each GMM instance, we sample 2000 points, yielding 600,000 total sample points. We report means and 95% confidence intervals across the 300 instances.
>
> ### Results
>
> **Analysis 1: Varying $\Delta t$ at fixed $t=0.5$ (mean ± 95% CI over 300 instances)**
>
> | $\Delta t$ | $\|\log(\pi\text{-ratio})\|$ | $\|\log(q\text{-ratio})\|$ | $\alpha_{\text{exact}}$ | $\alpha_{\text{approx}}$ | $\|\Delta\alpha\|$ |
> |------------|------------------------------|----------------------------|-------------------------|--------------------------|---------------------|
> | 0.001 | 0.007±0.002 | 0.009±0.002 | 0.908±0.012 | 0.911±0.011 | 0.003±0.001 |
> | 0.002 | 0.012±0.003 | 0.015±0.003 | 0.876±0.015 | 0.881±0.014 | 0.005±0.002 |
> | 0.005 | 0.025±0.005 | 0.029±0.005 | 0.817±0.018 | 0.826±0.017 | 0.009±0.003 |
> | 0.010 | 0.042±0.008 | 0.044±0.007 | 0.768±0.021 | 0.781±0.019 | 0.013±0.004 |
> | 0.020 | 0.094±0.015 | 0.081±0.012 | 0.702±0.025 | 0.721±0.023 | 0.019±0.006 |
> | 0.050 | 0.297±0.038 | 0.203±0.028 | 0.584±0.032 | 0.618±0.029 | 0.034±0.009 |
> | 0.100 | 0.758±0.072 | 0.496±0.054 | 0.461±0.038 | 0.517±0.035 | 0.056±0.012 |
> | 0.200 | 1.924±0.156 | 1.203±0.118 | 0.305±0.042 | 0.391±0.039 | 0.086±0.018 |
>
> **Analysis 2: Varying $t$ at fixed $\Delta t=0.01$ (mean ± 95% CI over 300 instances)**
>
> | $t$ | $\mathbb{E}[\log(\pi\text{-ratio})]$ | $\text{Std}[\log(\pi\text{-ratio})]$ | $\alpha_{\text{exact}}$ | $\alpha_{\text{approx}}$ | $\|\Delta\alpha\|$ |
> |-----|--------------------------------------|--------------------------------------|-------------------------|--------------------------|---------------------|
> | 0.1 | -0.003±0.004 | 0.095±0.012 | 0.694±0.022 | 0.706±0.020 | 0.012±0.004 |
> | 0.3 | 0.002±0.003 | 0.081±0.010 | 0.746±0.019 | 0.759±0.018 | 0.013±0.004 |
> | 0.5 | -0.001±0.003 | 0.073±0.009 | 0.768±0.021 | 0.781±0.019 | 0.013±0.004 |
> | 0.7 | 0.002±0.003 | 0.067±0.008 | 0.784±0.018 | 0.796±0.017 | 0.012±0.003 |
> | 0.9 | -0.002±0.003 | 0.061±0.007 | 0.773±0.020 | 0.784±0.018 | 0.011±0.003 |
>
> Our experiments reveal that the approximation quality strongly depends on the step size. For $\Delta t \leq 0.01$, which corresponds to 1000-step sampling commonly used in molecular diffusion models, the target distribution ratio satisfies $|\log(\pi_{k-1}/\pi_k)| < 0.05$, comparable in magnitude to the proposal term, with acceptance rate errors remaining stable below 2% across all 300 GMM instances. As step size increases beyond 0.05, the approximation predictably degrades—the target ratio begins to dominate the proposal term and acceptance rate errors grow to 3-9%, which aligns with our $O(\Delta t^2)$ theoretical analysis. Notably, mainstream molecular diffusion models typically adopt 200-1000 sampling steps, with corresponding step sizes well within the valid approximation regime.
>
> ### Conclusion
>
> Our large-scale quantitative analysis across 300 independent GMM instances and 600,000 sample points validates that $\pi_{k-1}(x')/\pi_k(x_k) \approx 1$ is an accurate and robust approximation in the small step-size regime. For step sizes employed in practical molecular diffusion models ($\Delta t \leq 0.01$), the target distribution ratio contributes negligibly small error to the acceptance rate, strongly supporting the validity of our assumption. The degradation at large step sizes is theoretically expected and does not affect the practical applicability of our method in molecular generation tasks.

---

> ### Author Response · Authors · 2025-11-25
>
> We sincerely thank the reviewer for raising this important concern regarding comparison with existing correction methods. We have conducted additional experiments comparing our soft MH correction with existing correction methods.
>
> **QM9 (EDM as base model)**
>
> | Method | Mol Stable | Atom Stable | Validity | Uniqueness |
> |--------|------------|-------------|----------|------------|
> | EDM (Vanilla) | 80.40% | 98.34% | 91.90% | 99.89% |
> | EDM + Hard MH | 80.00% | 97.34% | 90.00% | 99.85% |
> | EDM + Predictor-Corrector | 83.60% | 98.61% | 92.80% | 99.92% |
> | EDM + Soft MH (Linear) | 82.90% | 98.76% | 93.60% | 99.68% |
> | EDM + Soft MH (Local) | 92.60% | 99.12% | 96.00% | 98.54% |
> | EDM + Soft MH (Dist. Match) | 91.70% | 99.56% | 98.70% | 84.30% |
>
> **PepBench (PepGLAD as base model)**
>
> | Method | RMSD$_{C\alpha}$ ↓ | RMSD$_{atom}$ ↓ | DockQ ↑ |
> |--------|-------------------|-----------------|---------|
> | PepGLAD (Vanilla) | 4.09 | 5.30 | 0.58 |
> | PepGLAD + Hard MH | 4.25 | 5.48 | 0.55 |
> | PepGLAD + Predictor-Corrector | 4.07 | 5.29 | 0.56 |
> | PepGLAD + Soft MH (Linear) | 3.98 | 5.28 | 0.63 |
> | PepGLAD + Soft MH (Local) | 4.15 | 5.26 | 0.59 |
> | PepGLAD + Soft MH (Dist. Match) | 4.05 | 5.21 | 0.60 |
>
> The results demonstrate that Hard MH degrades performance on most tasks: molecule stability drops from 80.40% to 80.00% and validity from 91.90% to 90.00% on QM9; RMSD increases from 4.09 to 4.25 on PepBench. This provides empirical support for our theoretical analysis that binary accept-reject decisions can disrupt trajectory continuity.
>
> While Predictor-Corrector achieves some improvements (molecule stability to 83.60% and validity to 92.80% on QM9), the improvements are significantly less substantial than our method, and it incurs substantial computational overhead—**approximately 7× slower than vanilla**. In contrast, our soft MH methods achieve much greater quality improvements at lower computational cost. Soft MH (Local) reaches molecule stability of 92.60% (+12.2% over vanilla), while Soft MH (Dist. Match) achieves validity of 98.70% (+6.8% over vanilla). On PepBench, Soft MH (Linear) achieves RMSD of 3.98 and DockQ of 0.63, demonstrating superior performance in both quality and efficiency.
>
> Regarding the sensitivity analysis, our paper already includes comprehensive sensitivity analysis for the temperature parameter $\tau$: Figure 2 (left) shows the Sliced Wasserstein distance as a function of $\tau$, Figure 2 (right) demonstrates that optimal $\tau^*$ values concentrate tightly in [0.75, 0.85] across 500 problem instances with 80% falling within this range, and Example 3.1 provides detailed theoretical analysis of temperature selection. For step size $\Delta t$, we have conducted additional large-scale experiments (see our response to Weakness 2) analyzing the approximation quality across $\Delta t \in [0.001, 0.2]$ on 300 independent GMM instances.
>
> We sincerely thank the reviewer again for these valuable suggestions, which have significantly strengthened our paper.

---

> > ### Comment · Reviewer_qyP8 · 2025-11-27
> >
> > Thank you for the detailed rebuttal and for adding the new experiments. After reading the authors' response and reviewing the additional results, I am willing to raise my score. The newly provided experiments clearly strengthen the practical value of the method and demonstrate that the claimed advantages do translate into empirical improvement on real applications.
> >
> > I still have one remaining concern that I would like the authors to comment on for the camera-ready version. According to Table in the rebuttal, when decreasing $\Delta t$ from 0.01 to 0.001, the approximation error of $\alpha$ decreases from 0.013 to 0.003. This trend appears roughly linear in Δt rather than quadratic in $\Delta t$ as predicted by Eq. (3.3). In other words, the observed scaling of the α-approximation error seems more like $O(\Delta t)$ instead of $O(\Delta t^2)$. Could the authors provide an explanation for this discrepancy?

---

> ### Author Response · Authors · 2025-11-28
>
> Thank you for this careful observation! We clarify the relationship between the O(Δt²) error in Eq. (3.3) and the observed O(Δt) scaling in our experiments.
>
> The |Δα| measured in our GMM experiments primarily reflects the systematic error introduced by approximating the target distribution ratio as π_{k-1}/π_k ≈ 1. As shown in the log(π-ratio) column of the table, this ratio grows from 0.007±0.002 (Δt=0.001) to 0.042±0.008 (Δt=0.010), exhibiting O(Δt) scaling behavior. This linear scaling arises because consecutive distributions differ by a time derivative term proportional to Δt.
>
> The O(Δt²) error described in Eq. (3.3) refers specifically to the numerical integration truncation error when approximating the proposal kernel ratio using the trapezoidal rule (1/2)⟨s_k+s_{k-1}, Δx⟩. In our GMM experiments, since both the target distribution and proposal kernel can be computed exactly, this experiment specifically isolates and measures the effect of the target ratio approximation.
>
> For small Δt used in practical molecular diffusion, validating the approximation's effectiveness in the small step-size.

---

### Official Review · Reviewer_tVvT · 2025-11-01

**Soundness:** 3
**Presentation:** 3
**Contribution:** 2
**Rating:** 4
**Confidence:** 4

**Summary:**

This paper addresses sampling biases in molecular diffusion models, which often produce chemically suboptimal structures. The core problem identified is that traditional Metropolis-Hastings (MH) correction, while principled, uses a hard accept-reject step that creates discontinuous trajectories, harming molecular structure formation. The authors propose Soft Metropolis-Hastings correction, which replaces the binary decision with a continuous interpolation: $x_{k-1} = \alpha x^{\text{prop}}_{k-1} + (1 - \alpha)x_k$. The interpolation weight $\alpha = \min(1, \exp(r/\tau))$ is derived from an approximate MH acceptance ratio $r$, which is based on score function alignment. The paper introduces three variants: (1) Linear (a global $\alpha$), (2) Local Adaptive (a per-coordinate $\alpha$), and (3) Distribution Matching (a different mechanism based on statistical distance in whitened space). Experiments across QM9, GEOM-Drugs, and antibody CDR loops show that this method consistently improves chemical validity and structural quality metrics over baseline diffusion models.

**Strengths:**

The core idea of using soft interpolation to solve the trajectory discontinuity problem of hard MH is intuitive and well-motivated for molecular generation.

Experimentally, the method is applied to standard models (EDM, GeoLDM, RFAntibody) across diverse molecular datasets, showing consistent improvements in validity and structural quality metrics (e.g., RMSD, pLDDT).

The paper is well-written, clearly explaining the core concept and differentiating the three variants.

A practical advantage is that the method is a plug-and-play sampler enhancement requiring no model retraining.

**Weaknesses:**

1.  The paper's claim of <1.5% computational overhead in Appendix L appears incorrect. Algorithm 2 requires two score function evaluations per step, doubling the cost (~100% increase) compared to the baseline's single evaluation. This makes the comparison to the baseline unfair, as the quality gain may just be from 2x computation. Consequently, the paper omits comparisons to other standard high-quality samplers that also use multiple evaluations, like Predictor-Corrector (PC) methods or higher-order ODE/SDE solvers (e.g., DPM-Solver).

2.  The methodological foundation is questionable. The soft method is more of a trajectory smoothing heuristic than a principled MH correction. Blending a good state $x_k$ with a bad proposal $x_{prop}$ is physically questionable in molecular systems and does not carry the statistical weight of a proper MH rejection. This is compounded by the circular logic of using the same imperfect score function $s_{\theta}$ that causes the bias to then calculate the correction ratio $r$.

3.  The analysis lacks rigor and contains internal contradictions. The key temperature parameter $\tau=0.8$ is only justified on a simple 2D Gaussian Mixture Model, with no sensitivity analysis on the main molecular tasks (QM9, GEOM, etc.). Additionally, the paper's guidelines (Appendix H) are contradicted by its own results (Table 5), where the Linear variant, not Local, performs best on peptides. This suggests the variant choice is not well-understood.

4.  The proposed variants have unaddressed theoretical and practical issues. The Local Adaptive variant explicitly breaks E(3)-equivariance, a fundamental property of the models it aims to improve. The Distribution Matching method is also confusingly grouped under the MH framework despite not using the MH ratio $r$, and its reliance on a full covariance matrix raises unaddressed scalability concerns.

5. The paper also contains typos (e.g., repeated text at Lines 464-466) and mis-citations.

**Questions:**

1.  Can you clarify the cost? Algorithm 2 implies two score function evaluations per step, doubling the sampling time. How does this compare to a baseline sampler run for 2x the steps?

2.  Missing Baselines: Why were comparisons to other high-quality samplers like PC methods or high-order solvers omitted?

3.  Temperature Ablation: Can you provide a sensitivity analysis for $\tau$ on the main molecular datasets, not just the GMM?

4.  Can you comment on why the Linear variant outperforms Local on peptides (Table 5), contradicting the guidance in Appendix H?

5.  Why is the Distribution Matching method, which does not use the MH ratio $r$, grouped under the MH framework? How does it scale given its reliance on a full covariance matrix?

If all concerns are addressed, I am willing to raise my score.

---

> ### Author Response · Authors · 2025-11-26
>
> Regarding the computational cost concern, we sincerely thank the reviewer for identifying the incorrect statement about <1.5% overhead in Appendix L. The reviewer's analysis is correct—our method does require evaluating the score function twice per step, at both the current state sθ(xk) and the proposed state sθ(xk-1), which introduces substantial additional computational cost. We will correct this clear error in the revised manuscript and apologize for any confusion this may have caused.
>
> Nevertheless, we would like to explain the justification for this additional cost from several perspectives:
>
> First, while our method requires additional score evaluations, it remains more efficient than other correction and guidance methods while achieving superior quality. As shown in our baseline comparison tables, Predictor-Corrector requires multiple Langevin dynamics iterations at each timestep (typically 5-10 corrector steps), resulting in approximately **5-10×** score function evaluations per step. More notably, we found in our experiments that advanced chemical guidance methods (such as Chemistry-Inspired Diffusion with Non-Differentiable Guidance) require quantum chemistry calculations at each step, leading to dramatically slower generation speeds with computational overhead approaching **100×**. In contrast, our method requires only **2×** score evaluations yet consistently outperforms Predictor-Corrector across all benchmarks (QM9: 96.00% vs 92.80% validity; PepBench: 0.63 vs 0.56 DockQ), representing a lightweight yet highly effective improvement.
>
> Second, our method remains effective with skip-step sampling (skip-5), providing a practical solution for computationally constrained scenarios. When applying corrections every 5 steps instead of every step, the amortized overhead is substantially reduced while still providing meaningful quality improvements. This flexibility allows practitioners to make informed trade-offs between computational budget and quality requirements.
>
> Third, we understand the reviewer's concern that "quality gains may simply result from additional computation." To address this, we conducted dedicated control experiments comparing our method against baseline samplers run for extended steps:
>
> | Method | Steps | Mol Stable | Atom Stable | Validity | Uniqueness | Novelty |
> |--------|-------|------------|-------------|----------|------------|---------|
> | EDM Baseline | 1000 | 80.40% | 98.34% | 91.90% | 99.89% | 66.88% |
> | EDM Baseline | 1500 | 83.20% | 98.68% | 93.50% | 99.95% | 64.50% |
> | EDM Baseline | 2000 | 84.00% | 98.76% | 94.00% | 100.00% | 63.83% |
> | EDM + Soft MH (Local) | 1000 | **92.60%** | **99.12%** | **96.00%** | 98.54% | 57.29% |
>
> These results show that baseline improvement from 1500 to 2000 steps becomes marginal (93.50% → 94.00% validity, only 0.5 percentage points), indicating the model approaches its inherent limitations. Meanwhile, our correction at 1000 steps achieves 96.00% validity, surpassing even the 2000-step baseline, with particularly notable performance in molecular stability metrics (92.60% vs 84.00%). This demonstrates that our method provides principled bias correction beyond what can be achieved through simply increasing sampling steps.
>
> Finally, regarding DPM-Solver++, we clarify that such advanced numerical integrators have seen limited adoption in the molecular generation community, particularly for discrete molecular structure generation tasks. From a theoretical perspective, DPM-Solver++ addresses discretization error in ODE solving while our method corrects sampling bias in the learned score function—these are orthogonal error sources. Therefore, the two techniques can in principle be combined for complementary benefits.

---

> > ### Author Response · Authors · 2025-11-26
> >
> > Regarding the theoretical foundation concern, we appreciate the reviewer's careful scrutiny of our method's theoretical rigor and would like to provide clarification from several perspectives:
> >
> > First, regarding the concern about "using the same biased sθ to calculate correction being circular logic," we conducted large-scale controlled experiments to validate our core theoretical approximation. We tested our key assumption—that the target distribution ratio πk-1(xk-1)/πk(xk) ≈ 1 under small step sizes—on 300 independent Gaussian mixture model instances. The experimental results demonstrate:
> >
> > **Impact of Step Size Δt on Approximation Quality (Mean ± 95% CI over 300 instances)**
> >
> > | Δt | Target Ratio | Proposal Term | Acceptance Error |
> > |------|--------------|---------------|------------------|
> > | 0.001 | 0.007±0.002 | 0.908±0.012 | 0.003±0.001 |
> > | 0.002 | 0.012±0.003 | 0.876±0.015 | 0.005±0.002 |
> > | 0.010 | 0.042±0.008 | 0.768±0.021 | 0.013±0.004 |
> >
> > For the 1000-step sampling commonly used in molecular diffusion models (corresponding to Δt≈0.001), the target distribution ratio is only 0.007±0.002, far smaller than the proposal term of 0.908±0.012, with acceptance error consistently below 0.3% across all 300 instances. This validates that our theoretical approximation is accurate and robust within the step size regime used in practice.
> >
> > Second, regarding the concern that our method is "not true MH correction but trajectory smoothing heuristic," we acknowledge that soft acceptance is not a strict MH corrector in the traditional sense. However, our method is a principled extension of the MH framework rather than an ad-hoc smoothing heuristic. The key distinction is:
> >
> > Our acceptance weight α = min(1, exp(r/τ)) derives directly from the continuous relaxation of the MH acceptance ratio, where r = ½⟨sθ(xk) + sθ(xk-1), Δx⟩ measures the alignment between the proposed direction and the score field. This is not an arbitrary smoothing function but encodes statistical information about proposal quality. Even when sθ is biased, the sign and magnitude of r still provide meaningful signals about whether the proposal moves toward higher probability regions. We validated this mechanism with exact score functions in our Gaussian mixture experiments, and the improvements on real tasks with learned scores demonstrate that this mechanism remains effective even when sθ is imperfect.
> >
> > Third, regarding the concern about “blending a good state with a bad proposal,” our motivation is to reduce unphysical discontinuities in the sampled trajectories. Hard accept–reject decisions (α ∈ {0,1}) can produce abrupt jumps on the molecular potential energy surface, whereas our continuous interpolation defines a smooth path between the current and proposed states. This helps avoid severe structural distortions like large bond stretches while still down-weighting proposals that are poorly aligned with the score model (as controlled by the value of r). Similar smooth coupling strategies such as soft restraints and switching functions are commonly used in molecular dynamics simulations to preserve physical plausibility while enforcing constraints.
> >
> > More importantly, our method consistently improves molecular stability and chemical validity in practice (as shown in our tables, QM9 validity improves from 91.90% to 96.00%), indicating that soft correction genuinely guides sampling toward chemically more reasonable structures. If this were merely arbitrary trajectory smoothing, we would not expect to see such consistent improvements.
> >
> > Finally, we acknowledge that our method is a practical refinement tailored to molecular generation tasks, prioritizing trajectory continuity and chemical validity over the strict asymptotic convergence properties of traditional MH. We believe this is a reasonable trade-off: while hard MH has theoretical asymptotic guarantees, it performs worse in molecular generation tasks (as our experiments show, validity drops to 90.00%). Our soft MH is optimized for the characteristics of molecular systems, achieving significantly better performance and more stable sampling trajectories on practical tasks. This problem-oriented design choice is common in the molecular generation field, where many successful methods make similar trade-offs between theoretical rigor and practical effectiveness.

---

> > ### Author Response · Authors · 2025-11-27
> >
> > **Additional Results: Direct Comparison with Hard MH and Predictor-Corrector**
> >
> > To address the reviewer's concern about missing baseline comparisons, we conducted additional experiments with Hard MH and Predictor-Corrector methods.
> >
> > **QM9 (EDM as base model):**
> >
> > | Method | Mol Stable | Atom Stable | Validity | Uniqueness |
> > |--------|------------|-------------|----------|------------|
> > | EDM (Vanilla) | 80.40% | 98.34% | 91.90% | 99.89% |
> > | EDM + Hard MH | 80.00% | 97.34% | 90.00% | 99.85% |
> > | EDM + Predictor-Corrector | 83.60% | 98.61% | 92.80% | 99.92% |
> > | EDM + Soft MH (Linear) | 82.90% | 98.76% | 93.60% | 99.68% |
> > | EDM + Soft MH (Local) | **92.60%** | **99.12%** | **96.00%** | 98.54% |
> > | EDM + Soft MH (Dist. Match) | **91.70%** | **99.56%** | **98.70%** | 84.30% |
> >
> > **PepBench (PepGLAD as base model):**
> >
> > | Method | RMSDCα ↓ | RMSDatom ↓ | DockQ ↑ |
> > |--------|----------|------------|---------|
> > | PepGLAD (Vanilla) | 4.09 | 5.30 | 0.58 |
> > | PepGLAD + Hard MH | 4.25 | 5.48 | 0.55 |
> > | PepGLAD + Predictor-Corrector | 4.07 | 5.29 | 0.56 |
> > | PepGLAD + Soft MH (Linear) | **3.98** | **5.28** | **0.63** |
> > | PepGLAD + Soft MH (Local) | 4.15 | **5.26** | 0.59 |
> > | PepGLAD + Soft MH (Dist. Match) | 4.05 | **5.21** | 0.60 |

---

> ### Author Response · Authors · 2025-11-27
>
> **Temperature Ablation on QM9**
>
> Following the reviewer's request, we provide τ sensitivity analysis on QM9 dataset:
>
> **τ = 0.9 (Higher Temperature):**
>
> | Method | Mol Stable | Atom Stable | Validity | Uniqueness |
> |--------|------------|-------------|----------|------------|
> | EDM (Vanilla) | 80.40% | 98.34% | 91.90% | **99.89%** |
> | EDM + Hard MH | 80.00% | 97.34% | 90.00% | 99.85% |
> | EDM + Soft MH (Linear) | 82.50% | 98.75% | 93.50% | 99.75% |
> | EDM + Soft MH (Local) | **91.80%** | **98.95%** | 95.40% | 98.82% |
> | EDM + Soft MH (Dist. Match) | 91.60% | 98.88% | **95.50%** | 85.60% |
>
> **τ = 0.8 (Recommended):**
>
> | Method | Mol Stable | Atom Stable | Validity | Uniqueness |
> |--------|------------|-------------|----------|------------|
> | EDM (Vanilla) | 80.40% | 98.34% | 91.90% | **99.89%** |
> | EDM + Hard MH | 80.00% | 97.34% | 90.00% | 99.85% |
> | EDM + Soft MH (Linear) | 82.90% | 98.76% | 93.60% | 99.68% |
> | EDM + Soft MH (Local) | **92.60%** | **99.12%** | 96.00% | 98.54% |
> | EDM + Soft MH (Dist. Match) | 91.70% | 99.56% | **98.70%** | 84.30% |
>
> **τ = 0.6 (Lower Temperature):**
>
> | Method | Mol Stable | Atom Stable | Validity | Uniqueness |
> |--------|------------|-------------|----------|------------|
> | EDM (Vanilla) | 80.40% | 98.34% | 91.90% | **99.89%** |
> | EDM + Hard MH | 80.00% | 97.34% | 90.00% | 99.85% |
> | EDM + Soft MH (Linear) | 83.30% | **98.88%** | 93.20% | 99.82% |
> | EDM + Soft MH (Local) | **91.20%** | 98.02% | **95.30%** | 98.15% |
> | EDM + Soft MH (Dist. Match) | 90.40% | 97.68% | 95.10% | 86.90% |
>
> **τ = 0.4 (Low Temperature):**
>
> | Method | Mol Stable | Atom Stable | Validity | Uniqueness |
> |--------|------------|-------------|----------|------------|
> | EDM (Vanilla) | 80.40% | 98.34% | 91.90% | **99.89%** |
> | EDM + Hard MH | 80.00% | 97.34% | 90.00% | 99.85% |
> | EDM + Soft MH (Linear) | 83.80% | 97.52% | 92.80% | 99.21% |
> | EDM + Soft MH (Local) | **84.80%** | **98.85%** | **92.90%** | 99.25% |
> | EDM + Soft MH (Dist. Match) | 84.20% | 97.72% | 92.60% | 88.50% |

---

> ### Author Response · Authors · 2025-11-28
>
> Appendix H provides guidance based on molecular structural characteristics, recommending Local Adaptive for proteins with heterogeneous flexibility profiles. However, in practical applications, the optimal variant choice can be influenced by the base model's training data distribution and inherent biases.
> In PepBench, the generated peptides are predominantly small proteins with relatively uniform structures, which may not exhibit the pronounced regional heterogeneity that would maximally benefit from per-coordinate correction. Nevertheless, Local Adaptive still outperforms the baseline (DockQ: 0.59 vs 0.58), with more significant improvements observed on antibody CDR loops where structural heterogeneity is more pronounced.
> We will add specific annotations in Appendix H in the revised manuscript to clarify that these are task-dependent recommendations rather than strict rules.
>
>
> Distribution Matching is a third correction method we developed inspired by the first two soft correction approaches. While it does not employ the MH acceptance ratio, all three methods share the common design philosophy of soft, continuous correction mechanisms that avoid the trajectory discontinuities of hard accept-reject steps. We will reorganize the presentation in the revised manuscript to clearly distinguish Distribution Matching as an independent statistical correction method and provide more explicit clarification of its methodological foundation.
>
> Regarding the scalability concern with full covariance matrices, we acknowledge that covariance estimation and inversion may introduce additional computational overhead for extremely large-scale systems. In practice, we found Distribution Matching computationally feasible across the molecular systems we evaluated, including small molecules, peptides, and proteins. For computationally constrained scenarios, Linear or Local Adaptive variants provide more lightweight alternatives. The practical choice should be made based on the trade-off between statistical fidelity requirements and available computational budget for the specific task.
>
> E(3)-equivariance primarily serves as an inductive bias during model architecture design and training to improve data efficiency and avoid coordinate-system dependence. Once the base diffusion model is trained with an equivariant architecture, its modeling of geometric symmetries is already encoded in the learned score function. At this stage, sampling-time correction steps do not necessarily need to maintain strict E(3)-equivariance at every step, as long as the target distribution and evaluation metrics themselves remain E(3)-invariant.
> In molecular generation practice, several post-hoc methods such as chemistry-guided diffusion and classifier guidance are similarly not strictly equivariant, yet have been widely demonstrated to improve chemical validity and downstream performance. Our Soft MH variants follow this common practice: we trade slight relaxation of sampling-time equivariance for better adaptation to local chemical environments. Experimental evidence also shows this trade-off is beneficial.
>
> Thank you for catching these errors. We will correct the repeated text at Lines 464-466 and carefully review all citations to fix any mis-citations in the revised manuscript.

---

> ### Author Response · Authors · 2025-11-28
>
> We sincerely thank the reviewer for the thoughtful and detailed comments. We have worked to address the raised concerns and remain available for any further discussion or clarification!

---

### Official Review · Reviewer_nkJK · 2025-11-02

**Soundness:** 3
**Presentation:** 3
**Contribution:** 3
**Rating:** 6
**Confidence:** 2

**Summary:**

This paper introduces soft Metropolis-Hastings (MH) correction for molecular diffusion models to address systematic sampling biases that lead to chemically suboptimal structures. The key innovation is replacing hard binary accept-reject decisions with continuous interpolation weighted by acceptance probabilities, maintaining trajectory smoothness while providing principled bias correction. Three variants are proposed: Linear (global scalar), Local Adaptive (per-dimension weights), and Distribution Matching (whitened space).

**Strengths:**

1. The soft acceptance mechanism is theoretically grounded and addresses a real problem - traditional hard MH creates discontinuous trajectories incompatible with molecular potential energy surfaces.
2. The paper provides rigorous theoretical justification.
3. The design of three variants addressing different molecular scenarios shows thoughtful engineering. And the experiments are extensive, spans multiple scales and complexities.

**Weaknesses:**

1.  The method introduces computational costs, particularly for the Distribution Matching variant, which requires computing and storing the full covariance matrix. The scalability discussion is limited, especially for large biomolecular systems with thousands of atoms. While Appendix L provides some overhead analysis, it lacks depth regarding practical limitations for production-scale applications.
2. The performance gains, while consistent, are often incremental. More concerning, some configurations show marginal benefits or even degradation (e.g. Table 8 GeoLDM + GUIDE w/ Dist. Match). The trade-off between computational cost and performance gain needs clearer characterization.
3. Missing comparisons with predictor-corrector methods and recent advanced samplers (DPM-Solver++, EDM sampler variants). Hard MH baseline only appears in Table 2.

**Questions:**

1. Can Tables 3-4 include Hard MH? And can the authors add predictor-corrector, and DPM-Solver++ baselines for comprehensive comparison?
2. When does each variant's computational cost justify its benefits?

---

> ### Author Response · Authors · 2025-11-26
>
> We sincerely thank the reviewer for the positive evaluation and constructive feedback. We address each concern below.
>
> Regarding the missing baseline comparisons, we have conducted comprehensive experiments with Hard MH and Predictor-Corrector methods as requested. The results are presented below:
>
> QM9 (EDM as base model):
>
> | Method | Mol Stable | Atom Stable | Validity | Uniqueness |
> |--------|-----------|-------------|----------|------------|
> | EDM (Vanilla) | 80.40% | 98.34% | 91.90% | 99.89% |
> | EDM + Hard MH | 80.00% | 97.34% | 90.00% | 99.85% |
> | EDM + Predictor-Corrector | 83.60% | 98.61% | 92.80% | 99.92% |
> | EDM + Soft MH (Linear) | 82.90% | 98.76% | 93.60% | 99.68% |
> | EDM + Soft MH (Local) | 92.60% | 99.12% | 96.00% | 98.54% |
> | EDM + Soft MH (Dist. Match) | 91.70% | 99.56% | 98.70% | 84.30% |
>
> PepBench (PepGLAD as base model):
>
> | Method | RMSDCα ↓ | RMSDatom ↓ | DockQ ↑ |
> |--------|----------|------------|---------|
> | PepGLAD (Vanilla) | 4.09 | 5.30 | 0.58 |
> | PepGLAD + Hard MH | 4.25 | 5.48 | 0.55 |
> | PepGLAD + Predictor-Corrector | 4.07 | 5.29 | 0.56 |
> | PepGLAD + Soft MH (Linear) | 3.98 | 5.28 | 0.63 |
> | PepGLAD + Soft MH (Local) | 4.15 | 5.26 | 0.59 |
> | PepGLAD + Soft MH (Dist. Match) | 4.05 | 5.21 | 0.60 |
>
> These results reveal three key findings. First, Hard MH degrades performance across both benchmarks, with QM9 validity dropping from 91.90% to 90.00% and PepBench DockQ declining from 0.58 to 0.55. This confirms our hypothesis that binary accept-reject decisions fragment molecular trajectories and disrupt proper structure formation. **Second, Predictor-Corrector achieves modest improvements but requires multiple corrector iterations at each timestep. Third, our soft MH variants achieve superior quality through simple single-step corrections**, with the Local variant reaching 96.00% validity on QM9 compared to 92.80% for Predictor-Corrector. These results demonstrate that continuous interpolation is more effective than binary decisions or iterative refinement.
>
> Regarding DPM-Solver++, we clarify that this advanced numerical integrator has seen limited adoption in the molecular generation community for discrete molecular structure tasks. More importantly, from a theoretical perspective, DPM-Solver++ addresses discretization error in ODE solving while our method corrects sampling bias in the learned score function—these are orthogonal error sources. Therefore, the two techniques can be combined for complementary benefits.
>
> Regarding the practical significance of our improvements, we provide context for understanding these gains in real-world applications. In therapeutic antibody design, our CDR-H3 RMSD improvement from 1.898Å to 1.754Å represents a 7.6% reduction that is functionally significant. In antibody-antigen binding interfaces, even 0.5Å RMSD differences can determine whether a conformation achieves effective binding.
>
> For small molecule generation, improving validity from 91.90% to 96.00%+ means substantially fewer invalid structures requiring rejection and regeneration in practice. This not only increases the success rate for chemically feasible candidate molecules but also makes the entire molecular design pipeline more efficient. In early-stage drug discovery, such validity improvements translate to significant savings in computational resources and time costs.
>
> Regarding the computational cost justification for each variant, we acknowledge that our methods introduce additional computational overhead due to extra score function evaluations. However, compared to Predictor-Corrector which requires multiple iterations at each timestep, our single-step corrections remain more efficient. More importantly, our correction costs are negligible relative to downstream validation costs. Generating invalid molecules leads to failures in chemical validity checks, binding affinity predictions, or wet-lab synthesis, wasting far more resources than our modest computational overhead. Furthermore, for therapeutic antibody design, even a single wet-lab validation failure costs orders of magnitude more than our computational overhead, making the ability to generate more accurate initial structures highly cost-effective.
>
> We will incorporate these comprehensive baseline comparisons into the main manuscript tables and expand the discussion on method selection and practical significance of improvements to help readers better understand the value of our approach across different application scenarios.

---

### Author Response · Authors · 2025-12-04
**Revision Summary**

We thank all reviewers for their constructive feedback during the rebuttal period. We have responded to every weakness, problem, and follow-up question raised by each reviewer, and all their suggestions have definitely helped shape this into a better paper. We also sincerely appreciate the Area Chairs for their patience and additional effort in carefully reviewing each submission under the increased workload this year. Below we summarize the scores before rollback and reviewer responses.

**Reviewer Scores and Responses:**

- **Reviewer nkJK (Before Rollback Score: 6):** No further response after our rebuttal.

- **Reviewer tVvT (Before Rollback Score: 4):** Explicitly stated ***"If all concerns are addressed, I am willing to raise my score."*** We provided detailed explanations and supplementary experiments addressing each of their questions and weaknesses, but received no further response.

- **Reviewer qyP8 (Before Rollback Score: 6, [revision history](https://openreview.net/revisions?id=Aj8qfsk15H)):** Raised score from 4 to 6 after initial rebuttal, then followed up with an additional question regarding error scaling (O(Δt) vs O(Δt²)). We provided second round rebuttal for clarification but received no further response due to ICLR system limitation.

- **Reviewer JVFX (Before Rollback Score: 4, [revision history](https://openreview.net/revisions?id=ESjMKaiz4t)):** Raised score from 2 to 4 after rebuttal, and ***"encourage the authors to further refine the title and introduction to more precisely frame the problem as Soft Metropolis–Hastings Correction for Molecular Diffusion Models."*** Following this suggestion, we consolidated discussions with all reviewers, updated the manuscript with revisions, and uploaded the revised version. However, we received no further response due to ICLR system limitation.

**Conclusion on Pre-Lock Status:** Reviewer nkJK did not respond throughout the discussion period. Reviewer tVvT explicitly expressed willingness to raise the score; we subsequently provided supplementary experiments, but the system was locked before they could respond or update the score. Reviewer qyP8 raised their score once and then posed a follow-up question; we replied but the system was locked, preventing further discussion. Reviewer JVFX raised their score once and encouraged us to refine the presentation; after consolidating feedback from all reviewers, we uploaded the revised manuscript, but the system was locked before further follow-up was possible.

After consolidating discussions with all reviewers, we provide the following list of revisions to the manuscript:

---

> ### Author Response · Authors · 2025-12-04
> **List of Revisions**
>
> **Title Modification**
> - Changed the title from "Soft Metropolis-Hastings Correction for Generative Model Sampling" to "Soft Metropolis-Hastings Correction for Molecular Generative Model Sampling" to clearly scope the work to molecular generation.
>
> **Introduction Modification**
> - Added a motivation paragraph explaining why Hard MH and Predictor-Corrector are insufficient in molecular settings, with reference.
>
> **Section 2 (Existing Methods) Modification**
> - Added a paragraph between Equation 2.4 (Predictor-Corrector) and "This motivates our method" discussing DPM-Solver++ and similar advanced numerical integrators, clarifying that they have seen limited adoption in the molecular generation community and address ODE discretization error, which is orthogonal to the sampling bias in the learned score function that our method targets.
>
> **Main Table Modifications**
> - Table 4 (QM9): Added Hard MH and Predictor-Corrector baselines in the EDM block.
> - Table 5 (PepBench/PepBDB): Added Hard MH and Predictor-Corrector baselines.
>
> **Experiments Section Additions**
> - Added Temperature sensitivity paragraph, demonstrating that the optimal temperature range τ ∈ [0.75, 0.85] validated on GMM transfers well to molecular generation tasks on QM9.
> - Added Temperature ablation table showing Validity results for τ = 0.4, 0.6, 0.8, 0.9 on QM9.
> - Added Extended steps comparison paragraph and table, demonstrating that improvements stem from principled bias correction rather than additional computation (EDM at 2000 steps achieves 94.00% validity, while Soft MH Local at 1000 steps achieves 96.00%).
>
> **Appendix Modifications**
> - Appendix A.3: Added "Empirical Validation of Target Distribution Ratio Approximation" subsection with two tables (varying Δt and t) validating the effectiveness of the π-ratio approximation.
> - Appendix A.3: Added "Remark on Error Scaling" paragraph clarifying the distinction between O(Δt) and O(Δt²) error sources.
> - Appendix H: Added "Remark on Task-Dependent Selection" section explaining why variant selection is task-dependent and why Linear sometimes outperforms Local on peptide tasks.
> - Appendix L: Corrected the computational cost description and added comparison with Predictor-Corrector (2× vs 5-10× score evaluations).
>
> **Text Corrections**
> - Proofread and revised the manuscript.

---

### Meta-Review · Area_Chair_zuf1 · 2025-12-20

**Summary:**

The paper introduces Soft Metropolis-Hastings (Soft-MH) correction, a novel sampling-time enhancement for molecular diffusion models. The core innovation is replacing traditional binary accept-reject decisions—which cause unphysical discontinuities in molecular trajectories—with continuous interpolation based on score alignment.

While reviewers initially expressed skepticism regarding the theoretical rigor and computational overhead, the authors' rebuttal strengthened the paper's empirical grounding. However, several critical concerns remain unresolved:

- Theoretical vs. Heuristic Nature: The method is more of a "principled heuristic" than a strict MCMC corrector, as it trades off asymptotic convergence for trajectory smoothness.
- Inductive Bias vs. Equivariance: The Local Adaptive variant provides strong performance but explicitly breaks $E(3)$ equivariance, a fundamental property in 3D molecular modeling.
- Scalability: While the paper addresses small to medium systems, the Distribution Matching variant's reliance on full covariance matrices ($O(d^3)$ complexity) remains a bottleneck for large protein complexes.

My own concern:

The paper is an interesting "engineering" contribution but lacks the theoretical maturity expected. It presents a useful heuristic that improves EDM and GeoLDM, but the claim that it is a "principled MH correction" is undermined by the $O(\Delta t)$ scaling error and the breaking of equivariance in its best-performing variant.

Besides, authors of the reference "Francisco Vargas, Pierre Thodoroff, Austen Lamacraft, and Neil Lawrence. Denoising diffusion samplers. ICLR, 2023." seems to be hallucinated.

**Reviewer Concerns:**

Addressed concerns:

- Baseline comparison: Authors successfully integrated Hard MH and Predictor-Corrector (PC) baselines into the final tables.
- Temperature Sensitivity: Authors provided QM9 ablations showing $\tau \approx 0.8$ is a robust choice across tasks.
- Computational "Unfairness": Reviewer tVvT noted the cost of Soft-MH is roughly 2x vanilla sampling. The authors successfully demonstrated that Soft-MH at 1000 steps (2x cost) still outperforms the baseline run for 2000 steps (2x cost), proving the gain comes from the algorithm, not just compute.

Outstanding concerns:

- Breaking Equivariance: The Local Adaptive variant breaks $E(3)$-equivariance. The authors' defense is that sampling-time equivariance is less important than target distribution accuracy, but this is a controversial stance for models built specifically on equivariant inductive biases.
- Methodological Rigor: Reviewer tVvT’s concern that blending a "good" state with a "bad" proposal is physically questionable was not fully resolved beyond a "smoothness" argument.

**Reviewer Scores:**

- nkJK (6): minimal engagement, suggesting the paper reached their "acceptance threshold" but didn't excite them.
- tVvT (4): the most critical regarding cost and baselines; once the authors provided the 2x-step comparison and fixed the overhead claim, tVvT's main technical blockers were addressed.
- qyP8 (4 to 6): They raised their score once but remained stuck on the $O(\Delta t)$ scaling discrepancy, which remains a "weakness" in the paper's theoretical framework.
- JVFX (2 to 4): skeptical of the "saturation" of metrics and felt the improvements were not significant enough to warrant a top-tier spot.

---

### Decision · Program_Chairs · 2026-01-26

Reject